# S²MAM: Semi-supervised Meta Additive Model for Robust Estimation and Variable Selection

## Abstract

Semi-supervised learning with manifold regularization is a classical family for learning from the labeled and unlabeled data jointly, where the key requirement is the support of unknown marginal distribution enjoys the geometric structure of a Riemannian manifold. Usually, the Laplace-Beltrami operator-based manifold regularization can be approximated empirically by the Laplacian regularization associated with the whole training data and its graph Laplacian matrix. However, the graph Laplacian matrix depends heavily on the pre-specifying similarity metric and may result in inappropriate penalties when facing redundant and noisy input variables. In order to address the above issues, this paper proposes a new *Semi-Supervised Meta Additive Model* (S²MAM) under a bilevel optimization scheme to automatically identify the informative variables, update the similarity matrix, and achieve the interpretable prediction simultaneously. Theoretical guarantees are provided for S²MAM including the computing convergence and the statistical generalization bound. Experimental assessments on synthetic and real-world datasets validate the robustness and interpretability of the proposed approach.

## 1 Introduction

Manifold regularization provides an elegant and effective framework to develop semi-supervised learning (SSL) models by utilizing a large amount of unlabeled data with limited labeled data jointly (Belkin & Niyogi, 2004; Belkin et al., 2005; 2006; Geng et al., 2012; Van Engelen & Hoos, 2020). The key assumption of manifold regularization is that the support of intrinsic marginal distribution has the geometric structure of a Riemannian manifold (Belkin & Niyogi, 2004; Belkin et al., 2006; Johnson & Zhang, 2007; 2008)). Usually, the Laplace-Beltrami operator-based manifold regularization can be approximated empirically by the Laplacian regularization associated with the whole training data and the corresponding similarity (adjacent) matrix (Belkin & Niyogi, 2004; Belkin et al., 2006; Roweis & Saul, 2000), where the similarity matrix is constructed by the principles of Gaussian fields and harmonic functions (Zhu et al., 2003b) or the local and global consistency (Zhou et al., 2003). Typical manifold regularization schemes include Laplacian regularized least squares (LapRLS) and Laplacian regularized support vector machine (LapSVM) (Belkin et al., 2006). Moreover, Nie et al. considered a flexible manifold embedding for semi-supervised dimension reduction (Nie et al., 2010), and Qiu et al. further developed an accelerated version (called fast flexible manifold embedding (f-FME)) by reconstructing a smaller adjacency matrix with low-rank and sparse constraints (Qiu et al., 2018).

Despite rapid progress, it is still scarce to validate the intrinsic manifold assumption (Belkin & Niyogi, 2004; Belkin et al., 2006; Johnson & Zhang, 2007; 2008) for different types of data, e.g., data with redundant or even noisy variables. Moreover, the investigation for the robustness and interpretability of manifold regularization is far below its empirical applications only concerning the prediction accuracy. The existing manifold regularization models require that the similarity matrices are pre-specified before the semi-supervised training procedures, where the adaptivity and robustness of manifold learning are unexplored. For real applications, there unavoidably involve some abundant irrelevant and even noisy variables, and the pre-specified similarity metric associated with the whole variables can not reflect the true adjacent relations properly. The uninformative and noisy variables often result in a large deviation in estimating manifold structure, and then seriously degrade the prediction capability of manifold regularization methods. As illustrated in Figure 1, the clean unlabeled data are beneficial to better fit the decision curve, while the randomly added noisy variables

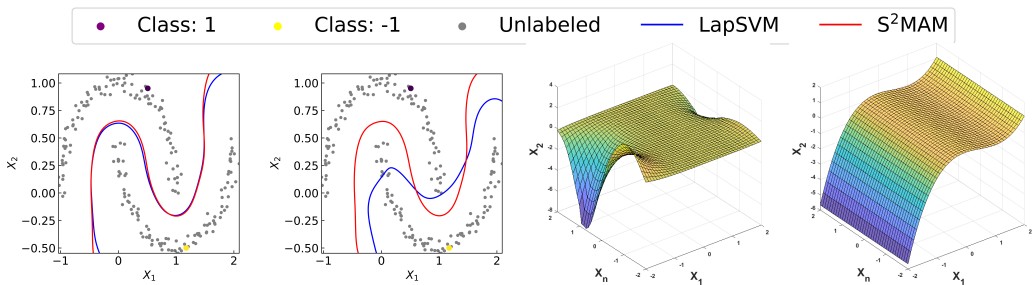

(a) Training on clean data  (b) Training on noisy data  (c) LapSVM on noisy data  (d) $S^2$MAM on noisy data

Figure 1: Toy examples on the impact of noisy variables in the moon dataset for LapSVM and our $S^2$MAM. (a) and (b) show the 2D prediction curves w.r.t the original input $X_1$ and $X_2$, where LapSVM is sensitive to feature corruptions $X_n$. (c) and (d) present the 3D decision surfaces on corrupted data, where $S^2$MAM is robust against the varying noisy variable $X_n$. Clean moon dataset contains inputs, $X_1$ and $X_2$. The corrupted data involves another noisy variable $X_n \in \mathcal{N}(100, 100)$. The used moon dataset contains 99 unlabeled points and only one labeled point for each class. Please refer to *Appendix B.8* for detailed descriptions.

obviously hurt the performance of LapSVM (See *Appendix B.8* for details). The inherent reason, resulting in the degraded performance, is the computation bias of the similarity matrix through the whole input variables directly (Nie et al., 2019; 2021). This motivates the following open questions:

"*How to alleviate the impact of redundant and even noisy variables on SSL models with manifold regularization? How to design a new manifold regularization scheme enjoying the robustness, interpretability, and prediction effectiveness simultaneously?*"

Intuitively, we can handle the above questions by a two-stage framework, i.e., selecting the informative variables firstly (e.g., via Lasso (Tibshirani, 1994), SpAM (Ravikumar et al., 2009)), and then implementing the manifold regularization approaches with the refined input variables. However, this variable selection strategy is independent of the intrinsic manifold structure and its accuracy is difficult to be guaranteed due to the scarcity of labeled data. Inspired by meta learning for coreset selection (Borsos et al., 2020; Zhou et al., 2022), this paper considers assigning all input variables with masks for both labeled and unlabeled data, where merely those truly informative variables are left for modeling and constructing the similarity matrix.

Nevertheless, there are several challenges along this way: 1) It is NP-hard to learn the discrete mask variables taking values in $\{0, 1\}$ directly. 2) The bilevel optimization usually needs the computation on Hessian and Jacobian matrices, which leads to a heavy computation burden. 3) Most kernel-based manifold regularization models construct the Gram matrix based on sample distance, which lacks the result's interpretability, e.g., screening the key variables associated with the response.

### 1.1 CONTRIBUTION

To address the aforementioned challenges, we inject the meta learning strategy and sparse additive models into manifold regularized SSL framework, and formulate a new *Semi-Supervised Meta Additive Model* ($S^2$MAM) to realize automatic variable masking and sparse approximation for high-dimensional inputs even with noisy variables.

The core technique is to update the decision function and similarity matrix simultaneously with proper masks on input variables, where the masks of $S^2$MAM are learned by a probabilistic meta strategy. Moreover, an efficient implementation is employed here to solve the bilevel optimization problem, which avoids the heavy computing burden on the implicit hypergradient calculation (Pedregosa, 2016), Neumann series and some variants with Hessian-vector or Jacobian-vector products (Lorraine et al., 2020; Ghadimi & Wang, 2018).

The main contributions of this paper are summarized below:

- *New statistical modeling*. To the best of our knowledge, our $S^2$MAM is the first meta learning method for manifold regularized additive models, where a novel bilevel optimization scheme is formulated for robust estimation and data-driven automatic variable selection

Table 1: Properties of our S$^2$MAM and related models ($\checkmark$ = enjoying the given information, and $\times$ = not available for the information).

| | SpAM | LapRLS | f-FME | AWSSL | PBCS | S$^2$MAM (ours) |
|---|---|---|---|---|---|---|
| Learning Task | Supervised | Semi-Supervised | Semi-Supervised | Semi-Supervised | Supervised | Semi-Supervised |
| Optimization Framework | Single-level | Single-level | Single-level | Single-level | Bilevel | Bilevel |
| Variable Selection | $\checkmark$ | $\times$ | $\checkmark$ | $\checkmark$ | $\checkmark$ | $\checkmark$ |
| Noisy Variable Robustness | $\times$ | $\times$ | $\times$ | $\checkmark$ | $\checkmark$ | $\checkmark$ |
| Convergence Analysis | $\times$ | $\times$ | $\times$ | $\times$ | $\checkmark$ | $\checkmark$ |
| Generalization Analysis | $\checkmark$ | $\times$ | $\times$ | $\times$ | $\times$ | $\checkmark$ |
| Computation Complexity Analysis | $\times$ | $\times$ | $\times$ | $\times$ | $\times$ | $\checkmark$ |

simultaneously. By assigning flexible masks on individual variables, the proposed S$^2$MAM is capable of reducing the impact of noisy variables on SSL tasks.

- *Computing and Theoretical Supports*. An efficient probabilistic bilevel optimization is developed to additionally learn the discrete masks, where the policy gradient estimation and the projection operation are employed. This computing algorithm reduces the computational burden of discrete bilevel optimization framework and enjoys theoretical guarantees of optimization convergence. Besides, we also establish the upper bounds of excess risk for the baseline model of S$^2$MAM, which implies the proposed approach can reach polynomial decay on generalization error.

- *Empirical competitiveness*. Empirical results on several synthetic and real-world benchmarks demonstrate that the proposed S$^2$MAM can identify the truly informative variables and realize robust prediction even facing redundant and noisy input variables.

## 1.2 COMPARISONS WITH THE RELATED WORKS

*Semi-supervised dimensionality reduction*. Recently, some efforts were made towards constructing a flexible similarity matrix against feature corruptions for SSL with manifold regularization (Chen et al., 2018; Nie et al., 2019). By rescaling the regression coefficients as variable weights, Chen et al. (Chen et al., 2018) developed an efficient SSL method to obtain important variables, which is called rescaled linear square regression. Another weighting approach in (Nie et al., 2019) is called auto-weighting semi-supervised learning (AWSSL), which adaptively assigns continuous weights on variables to update the similarity matrix. After the dimension reduction process, a specific classifier is employed for downstream tasks. A robust graph learning (RGL) method (Kang et al., 2020) combined label ranking regression and label propagation into a unified framework for weight graph construction and semi-supervised learning. Semi-supervised adaptive local embedding learning (SALE) (Nie et al., 2021) adaptively constructs two affinity graphs (based on labeled data and all embedding samples) separately to explore the local and global structures. Different from these works, this paper considers to automatically assign discrete masks 0/1 on input features (variables) for screening the truly active variables.

*Sparse additive models*. Additive models (Stone, 1985; Hastie & Tibshirani, 1990), as natural nonparametric extensions of linear models, have been burgeoning in high-dimensional data analysis due to their attractive properties, i.e., overcoming the curse of dimensionality, the flexibility of function approximation, and the ability of variable selection (Meier et al., 2009; Christmann & Hable, 2012; Yuan & Zhou, 2016; Chen et al., 2020). In recent years, many sparse additive models have been proposed from various theoretical or empirical motivations, see e.g., (Lv et al., 2018; Haris et al., 2022; Bouchiat et al., 2024; Duong et al., 2024). Naturally, the paradigm of additive models can be applied to semi-supervised learning settings. As far as we know, there are only three papers that touched on this topic (Culp & Michailidis, 2008; Culp et al., 2009; Culp, 2011). However, all of them don't consider the robustness on manifold learning against noisy variables, and ignore the data-driven variable structure discovery. These strong restrictions on the pre-defined similarity matrix and variable structure may result in degrading seriously of existing models under complex noise circumstances.

*Meta learning for sample/variable selection*. The meta-based masking policy was developed in (Borsos et al., 2020), where a bilevel neural network is designed for automatic supervised coreset selection. Furthermore, its improved version with probabilistic bilevel optimization is proposed for supervised classification (Zhou et al., 2022), especially for corrupted and imbalanced data. Indeed, Zhou et al. (Zhou et al., 2022) also provide an example of variable selection, while it is limited to

the supervised learning case and doesn't concern the impact of noisy variables. To the best of our knowledge, there has been no any endeavor before to explore the meta-based masking policy for semi-supervised additive models.

To better highlight the novelty of our S$^2$MAM, we summarize its properties in Table 1 compared with several related state-of-art models including sparse additive models (SpAM) (Ravikumar et al., 2009)), LapRLS (Belkin et al., 2006)), fast flexible manifold embedding (f-FME) (Qiu et al., 2018), auto-weighting semi-supervised learning (AWSSL) (Nie et al., 2019) and the probabilistic bilevel coreset selection (PBCS) (Zhou et al., 2022). Table 1 shows that the proposed S$^2$MAM enjoys nice properties, e.g., variable selection, robust estimation, and computing guarantees.

## 2 SEMI-SUPERVISED ADDITIVE MODELS

This section first introduces a manifold regularized semi-supervised additive model (Culp, 2011) as basic model, and then formulates the S$^2$MAM under the discrete bilevel optimization framework. Furthermore, a probabilistic bilevel scheme solves the NP-hard discrete optimization problem.

### 2.1 REVISITING MANIFOLD REGULARIZED SPARSE ADDITIVE MODEL

Let $\mathcal{X} = \{\mathcal{X}^{(1)}, \cdots, \mathcal{X}^{(p)}\} \in \mathbb{R}^p$ be a compact input space and the output space $\mathcal{Y} \in \mathbb{R}$. Denote $\rho$ as the jointed distribution on $\mathcal{X} \times \mathcal{Y}$, and $\rho_{\mathcal{X}}$ as the marginal distribution with respect to $\mathcal{X}$ induced by $\rho$. The training set $\mathbf{z} = \{\mathbf{z}_l, \mathbf{z}_u\}$ involves the labeled set $\mathbf{z}_l = \{(x_i, y_i)\}_{i=1}^l$ and the unlabeled set $\mathbf{z}_u = \{x_i\}_{i=l+1}^{l+u}$, where each input $x_i = (x_i^{(1)}, \cdots, x_i^{(p)})^T \in \mathbb{R}^p$ with $x_i^{(j)} \in \mathcal{X}^{(j)}$ and output $y_i \in \mathbb{R}$. The hypothesis space of additive models can be formulated as

$$\mathcal{F} = \{f : f(x) = \sum_{j=1}^p f^{(j)}(x^{(j)}), f^{(j)} \in \mathcal{F}^{(j)}\},$$

where $x^{(j)} \in \mathcal{X}^{(j)}$ and $\mathcal{F}^{(j)}$ is the component function space on $\mathcal{X}^{(j)}$ (Ravikumar et al., 2009). Typical candidates of additive hypothesis space include the basis expansion space (Meier et al., 2009; Ravikumar et al., 2009), the reproducing kernel Hilbert space (RKHS) (Raskutti et al., 2012; Christmann & Zhou, 2016), and the neural networks-based space (Agarwal et al., 2021; Yang et al., 2020).

This paper chooses $\mathcal{H}_{K^{(j)}}$ to form the additive hypothesis space, where $\mathcal{H}_{K^{(j)}}$ is the RKHS associated with Mercer kernel $K^{(j)}$ defined on $\mathcal{X}^{(j)} \times \mathcal{X}^{(j)}, j \in \{1, \ldots, p\}$. Equipped by component function $f^{(j)} : \mathcal{X}^{(j)} \to \mathbb{R}, j \in \{1, \ldots, p\}$, the additive hypothesis space can be further defined as

$$\mathcal{H} = \left\{f = \sum_{j=1}^p f^{(j)} : f^{(j)} \in \mathcal{H}_{K^{(j)}}, 1 \leq j \leq p\right\}$$

with $\|f\|_K^2 = \inf\left\{\sum_{j=1}^p \|f^{(j)}\|_{K^{(j)}}^2 : f = \sum_{j=1}^p f^{(j)}\right\}$. Indeed, $\mathcal{H}$ is an RKHS associated with kernel $K = \sum_{j=1}^p K^{(j)}$ (Christmann & Zhou, 2016). Due to the representer theorem of RKHS (Smola & Schölkopf, 1998), the prediction function of supervised additive models in RKHS often enjoys the parameter presentation

$$f(\cdot) = \sum_{j=1}^p \sum_{i=1}^l \alpha_i^{(j)} K_i^{(j)}(x_i^{(j)}, \cdot), \tag{1}$$

see e.g., (Yuan & Zhou, 2016; Christmann & Hable, 2012; Chen et al., 2020).

Given a predictor $f : \mathcal{X} \to \mathbb{R}$, denote $\mathbf{f} = (f(x_1), \ldots, f(x_{l+u}))^T$ as the prediction vector associated with the labeled data $\mathbf{z}_l$ and the unlabeled data $\mathbf{z}_u$. Let $\lambda_1, \lambda_2 > 0$ be the regularization coefficients and let $\tau_j$ be the positive weight to different input variables for $j = 1, \cdots, p$. Then the additive model for regularized Laplacian regression can be formulated as

$$f_{\mathbf{z}} = \arg\min_{f \in \mathcal{H}} \left\{\mathcal{E}_{\mathbf{z}}(f) + \lambda_1 \Omega_{\mathbf{z}}(f) + \frac{\lambda_2}{(l+u)^2} \mathbf{f}^T \mathbf{L} \mathbf{f}\right\}, \tag{2}$$

where the empirical risk

$$\mathcal{E}_{\mathbf{z}}(f) = \frac{1}{l} \sum_{i=1}^{l} (y_i - f(x_i))^2,$$

the sparse regularization

$$\Omega_{\mathbf{z}}(f) = \inf \Big\{ \sum_{j=1}^{p} \tau_j \|\boldsymbol{\alpha}^{(j)}\|_2 : f = \sum_{j=1}^{p} \sum_{i=1}^{l} \alpha_i^{(j)} K_i^{(j)}(x_i^{(j)}, \cdot) \Big\},$$

and the term $\mathbf{f}^T \boldsymbol{L} \mathbf{f}$ is the manifold regularization (Belkin & Niyogi, 2004; Culp, 2011). Here, $\boldsymbol{L} = \boldsymbol{D} - \boldsymbol{W}$ is the graph Laplacian, and diagonal matrix $\boldsymbol{D}$ satisfies $D_{ii} = \sum_{j=1}^{l+u} W_{ij}$ and $W_{ij}$ is the adjacent weight for inputs $x_i$ and $x_j$, e.g., $W_{ij} = \exp\{-\|x_i - x_j\|_2^2/\mu^2\}$ with bandwidth $\mu$.

**Remark 1.** *If the $j$-th variable is not truly informative, $\boldsymbol{\alpha}_{\mathbf{z}}^{(j)} = (\alpha_{\mathbf{z},1}^{(j)}, \ldots, \alpha_{\mathbf{z},l+u}^{(j)})^T \in \mathbb{R}^{l+u}$ is expected to satisfy $\|\boldsymbol{\alpha}_{\mathbf{z}}^{(j)}\|_2 = \sqrt{\sum_{i=1}^{l+u} \left| \alpha_{\mathbf{z},i}^{(j)} \right|^2} = 0$. Thus, $\ell_{2,1}$-regularizer is employed as the penalty. Obviously, noisy input variables may bring an inappropriate similarity matrix $\boldsymbol{W}$. Naturally, it is necessary to improve the robustness of (2) against noisy variables by replacing the pre-specified similarity measure (i.e., $\boldsymbol{W}, \boldsymbol{L}$) in manifold regularization with adaptive masking strategy.*

## 2.2 Discrete Bilevel Framework for S²MAM

To reduce the negative impact of noisy variables on Laplacian regularization in (2), we introduce a bilevel optimization framework for automatically learning the masks on variables. In particular, both the decision function $f$ and Laplacian matrix $\boldsymbol{L}$ are updated by the learned masks.

Denote $\ell(\cdot)$ as the loss function, $f(x; \boldsymbol{\alpha})$ as a decision function in RKHS $\mathcal{H}$ with spanning parameter $\boldsymbol{\alpha}$ and the mask $\boldsymbol{m} \in \{0,1\}^p$ as a binary vector, where $m_i = 1$ implies $i$-th variable is selected as the informative one and otherwise ignored. The bilevel framework for directly learning the discrete masks is formulated as follows.

**Upper Level:** Given the meta dataset $D_{meta} = \{(x_i, y_i)\}_{i=1}^{l}$, we formulate the discrete optimization

$$\min_{\boldsymbol{m} \in \tilde{\mathcal{C}}} \mathcal{L}(\boldsymbol{\alpha}^*(\boldsymbol{m})) = \frac{1}{l} \sum_{i=1}^{l} \ell\left(f(x_i; \boldsymbol{\alpha}^*(\boldsymbol{m})), y_i\right), \tag{3}$$

where the mask $\boldsymbol{m}$ is the learnable parameter in the upper level, $\boldsymbol{\alpha}$ is the parameter of the decision function in the lower level depending on $\boldsymbol{m}$, and $\tilde{\mathcal{C}} = \{\boldsymbol{m} : m_i \in \{0,1\}, \|\boldsymbol{m}\|_0 \leq C, i = 1, 2, \cdots, p\}$ is the feasible region of $\boldsymbol{m}$ with the size of selected variables $C$.

**Lower Level:** Based on the whole training set $D_{total}$ involving $D_{meta}$ and unlabeled samples $\{x_i\}_{i=l+1}^{l+u}$, the predictor of lower level optimization problem is

$$\hat{f}(x) = \sum_{j=1}^{p} \hat{f}^{(j)}(m_j x^{(j)}) = \sum_{j=1}^{p} \sum_{i=1}^{l} \alpha_i^{(j)} K_i^{(j)}(m_j x_i^{(j)}, m_j x^{(j)}), \tag{4}$$

where

$$\hat{\boldsymbol{\alpha}} = \underset{\boldsymbol{\alpha} \in \mathbb{R}^{(l+u) \times p}}{\arg\min} \mathcal{R}(\boldsymbol{\alpha}; \boldsymbol{m}; \boldsymbol{L}), \tag{5}$$

with

$$\mathcal{R}(\boldsymbol{\alpha}; \boldsymbol{m}; \boldsymbol{L}) = \frac{1}{l} \sum_{i=1}^{l} \ell(f(x_i \odot \boldsymbol{m}; \boldsymbol{\alpha}), y_i) + \lambda_1 \sum_{j=1}^{p} \tau_j \|\boldsymbol{\alpha}^{(j)}\|_2 + \frac{\lambda_2}{(l+u)^2} \mathbf{f}^T \boldsymbol{L} \mathbf{f}.$$

Different from (2), the Laplacian matrix $\boldsymbol{L}$ in (5) is computed based on the masked similarity matrix $\boldsymbol{W}$ with measure function $\mathcal{W}(\cdot, \cdot)$ and element $W_{ij} = \mathcal{W}(x_i \odot \boldsymbol{m}, x_j \odot \boldsymbol{m}), i, j \in \{1, 2, \cdots, l+u\}$.

Usually, it is intractable to directly solve the above discrete bilevel problem. Fortunately, we can formulate its continuous probabilistic form with the help of policy gradient estimation (Zhou et al., 2022), and develop an efficient gradient-based optimization algorithm in the following Section 2.3.

**Algorithm 1:** Procedure for S²MAM

**Input**: Labeled data $\mathbf{z}_l = \{(x_i, y_i)\}_{i=1}^{l}$, unlabeled data $\mathbf{z}_u = \{x_i\}_{i=l+1}^{l+u}$, step size $\eta^t$, core size $C$, $\mathbf{1} = (1, ..., 1) \in \mathbb{R}^p$.
**Initialization**: $\boldsymbol{\alpha}^0$, $\boldsymbol{s}^0 = \frac{C}{p} \cdot \mathbf{1}$, $\boldsymbol{m}^0$, $\boldsymbol{L}^0$.

  **for** $t = 1$ to $T$ **do**
    1) Update $\boldsymbol{\alpha}^t$ based on Step 1 with $\mathbf{z}_l$ & $\mathbf{z}_u$
    2) Update $\boldsymbol{s}^t$ based on Step 2 with $\mathbf{z}_l$
    3) Update $\boldsymbol{m}^t$ sampled from $p(\boldsymbol{m}|\boldsymbol{s}^t)$
    4) Update $\boldsymbol{L}^t$ based on Step 3 with $\mathbf{z}_l$ & $\mathbf{z}_u$
  **end for**
**Output**: Decision function $\hat{f}$.

**Algorithm 2:** Projection Operation $\mathcal{P}_{\mathcal{C}}(\boldsymbol{a})$

**Input**: Vector $\boldsymbol{a} \in \mathbb{R}^p$, core variables $C$, Domain $\mathcal{C} = \{\boldsymbol{s} : 0 \preceq \boldsymbol{s} \preceq 1, \|\boldsymbol{s}\|_1 \leq C\}$.

  1) Computing auxiliary variable $b$ satisfying:
    $\mathbf{1}^{\top} [\min(1, \max(0, \boldsymbol{a} - b \cdot \mathbf{1}))] - C = 0$
  2) Computing auxiliary variable $c$ satisfying:
    $c \leftarrow \max(0, b)$
  3) Update $\boldsymbol{a}$:
    $\boldsymbol{a}^* \leftarrow \min(1, \max(0, \boldsymbol{a} - c \cdot \mathbf{1}))$
**Output**: $\mathcal{P}_{\mathcal{C}}(\boldsymbol{a}) = \boldsymbol{a}^*$.

### 2.3 Probabilistic Bilevel Framework for S²MAM

It is popular to transform the discrete tunning parameter space into the continuous probability space for bilevel optimization (Zhao et al., 2023; Zhou et al., 2022). For simplicity, $m_i$ can be considered as a Bernoulli random variable $m_i \sim \text{Bern}(s_i)$, where $s_i \in [0, 1]$ represents the probability of $m_i = 1$. Denote the domain on probability variable $\boldsymbol{s} = (s_1, ..., s_p) \in \mathbb{R}^p$ as

$$\mathcal{C} = \{\boldsymbol{s} : 0 \preceq s_i \preceq 1, \|\boldsymbol{s}\|_1 \leq C, i = 1, 2, \cdots, p\}.$$

The discrete bilevel optimization in Section 2.2 can be relaxed into the following expected form

$$\min_{\boldsymbol{s} \in \mathcal{C}} \Phi(\boldsymbol{s}) = \mathbb{E}_{p(\boldsymbol{m}|\boldsymbol{s})} \mathcal{L}(\boldsymbol{\alpha}^*(\boldsymbol{m})), \text{ s.t. } \boldsymbol{\alpha}^*(\boldsymbol{m}) \in \operatorname*{arg\,min}_{\boldsymbol{\alpha} \in \mathbb{R}^{(l+u) \times p}} \mathcal{R}(\boldsymbol{\alpha}; \boldsymbol{m}; \boldsymbol{L}). \quad (6)$$

**Remark 2.** *Under the independent assumption on variable $m_i$, we can derive its distribution $p(\boldsymbol{m} \mid \boldsymbol{s}) = \Pi_{i=1}^{p} (s_i)^{m_i} (1 - s_i)^{(1-m_i)}$. Since $\mathbb{E}_{\boldsymbol{m} \sim p(\boldsymbol{m}|\boldsymbol{s})} \|\boldsymbol{m}\|_0 = \sum_{i=1}^{p} s_i$, the original domain $\tilde{\mathcal{C}} = \{\boldsymbol{m} : m_i \in \{0, 1\}, \|\boldsymbol{m}\|_0 \leq C, i = 1, 2, \cdots, p\}$ is transformed into $\mathcal{C}$ on probability $\boldsymbol{s}$.*

**Remark 3.** *A naive idea for continuing $\boldsymbol{m}$ is to directly consider it as a dynamic weighting vector varying in $[0, 1]$, which would bring expensive computation costs for the hypergradient estimation.*

### 2.4 Computing Algorithm of S²MAM

Initialize the decision parameter $\boldsymbol{\alpha}^0 = \mathbf{0}$, mask $\boldsymbol{m}^0 = \mathbf{1}$, probability $\boldsymbol{s}^0 = \frac{C}{p} \cdot \mathbf{1}$ and select Laplacian matrix associated with original $(x_1, \cdots, x_{l+u})$ as $\boldsymbol{L}^0$. Before each iteration, a sample batch $\mathcal{B}$ is selected from the whole training set. The computing steps of probabilistic S²MAM are summarized in Algorithm 1. The procedures for solving (6) at the $t$-th iteration contain:

**Step 1: Computing $\boldsymbol{\alpha}^t$ with $m^{t-1}$ and $\boldsymbol{L}^{t-1}$ by**

$$\boldsymbol{\alpha}^t = \operatorname*{arg\,min}_{\boldsymbol{\alpha} \in \mathbb{R}^{(l+u) \times p}} \mathcal{R}(\boldsymbol{\alpha}^{t-1}; \boldsymbol{m}^{t-1}; \boldsymbol{L}^{t-1}), \quad (7)$$

with $\mathcal{R}(\boldsymbol{\alpha}^{t-1}; \boldsymbol{m}^{t-1}; \boldsymbol{L}^{t-1})$ defined in (5). The computation algorithm for Step 1 based on the alternating direction method of multipliers is left in *Appendix E.4*.

**Step 2: Computing $\boldsymbol{s}^t$ and $m^t$ with $\boldsymbol{\alpha}^t$:**

From the probabilistic S²MAM in (6), the learning target changes from the discrete masks $\boldsymbol{m}$ into the continuous probability $\boldsymbol{s}$, which is updated by the policy gradient estimator (Zhou et al., 2022):

$$\nabla_{\boldsymbol{s}} \Phi(\boldsymbol{s}) = \mathbb{E}_{p(\boldsymbol{m}|\boldsymbol{s})} \mathcal{L}(\boldsymbol{\alpha}^*(\boldsymbol{m})) \nabla_{\boldsymbol{s}} \ln p(\boldsymbol{m} \mid \boldsymbol{s}).$$

This computing procedure is unbiased gradient estimation and without heavy computation burden on the inverse of the Hessian matrix or implicit differentiation.

Denote $\eta^t$ as the step size for updating the upper level parameter $\boldsymbol{s}$ at the $t$-th step. Given $\boldsymbol{\alpha}^t$, $\boldsymbol{s}$ can be updated by the projected stochastic gradient descent below:

$$\boldsymbol{s}^t \leftarrow \mathcal{P}_{\mathcal{C}} \left( \boldsymbol{s}^{t-1} - \eta^t \mathcal{L}(\boldsymbol{\alpha}^t) \nabla_{\boldsymbol{s}} \ln p(\boldsymbol{m}^{t-1} \mid \boldsymbol{s}^{t-1}) \right), \quad (8)$$

where the projection $\mathcal{P}_{\mathcal{C}}(\boldsymbol{s})$ from $\boldsymbol{s}$ to the domain $\mathcal{C}$ is summarized in Algorithm 2. Then, $\boldsymbol{m}^t = (m_1^t, \cdots, m_p^t) \in \mathbb{R}^p$ follows from the Bernoulli distribution, where $m_i^t \sim \text{Bern}(s_i^t)$. *Appendix E* states the theoretical validation of the closed-form solution in the projection computation (8).

**Step 3: Updating Laplacian matrix $\boldsymbol{L}^t$ with $\boldsymbol{m}^t$:**

$$\boldsymbol{L}^t = \boldsymbol{D}^t - \boldsymbol{W}^t, \tag{9}$$

where the diagonal matrix $\boldsymbol{D}^t \in \mathbb{R}^{(l+u) \times (l+u)}$ satisfies $D_{ii}^t = \sum_{j=1}^{l+u} W_{ij}$, and $W_{ij} = \exp\{-\|x_i \odot \boldsymbol{m}^t - x_j \odot \boldsymbol{m}^t\|_2^2 / \mu^2\}$ with the bandwidth parameter $\mu > 0$. The metric $W_{ij}$ evaluates the similarity between samples $x_i$ and $x_j$ with the shared mask $\boldsymbol{m}^t$. Finally, we obtain the decision function in (4) with coefficient $\boldsymbol{\alpha}$ and mask $\boldsymbol{m}$.

# 3 THEORETICAL ASSESSMENTS

For the proposed S$^2$MAM, this section states its computing convergence and generalization analysis for its basic model (2) in Section 2.1. All proofs are left in *Appendices C&D*.

## 3.1 COMPUTING CONVERGENCE ANALYSIS

Now we establish the theoretical guarantee of optimization convergence for the policy gradient estimation in equation 8. The following assumption has been used widely for characterizing the convergence behavior of projection operation algorithms (Pedregosa, 2016; Zhou et al., 2022) and bilevel optimization with sample batch (Shu et al., 2023).

**Assumption 1.** *Denote $\mathcal{L}_{\mathcal{B}}$ as the loss on selected sample batch $\mathcal{B}$. Assume that $\Phi(\boldsymbol{s})$ is $L$-smooth, constant $\sigma > 0$, there hold $\mathbb{E}[\mathcal{L}_{\mathcal{B}}(\boldsymbol{\alpha}^*(\boldsymbol{m})) \nabla_{\boldsymbol{s}} \ln p(\boldsymbol{m} \mid \boldsymbol{s}^t) - \nabla_{\boldsymbol{s}} \Phi(\boldsymbol{s}^t)] = 0$, and $\mathbb{E} \|\mathcal{L}_{\mathcal{B}}(\boldsymbol{\alpha}^*(\boldsymbol{m})) \nabla_{\boldsymbol{s}} \ln p(\boldsymbol{m} \mid \boldsymbol{s}^t) - \nabla_{\boldsymbol{s}} \Phi(\boldsymbol{s}^t)\|^2 \leq \sigma^2$.*

**Theorem 1.** *At the $t$-th iteration, let the step size $\eta^t = \frac{c}{\sqrt{t}} \leq \frac{1}{L}$ for some constant $c > 0$, and denote the gradient mapping $\mathcal{G}^t = \frac{1}{\eta^t}(s^t - \mathcal{P}_{\mathcal{C}}(s^t - \eta^t \nabla_s \Phi(s^t)))$. Under Assumption 1, there holds*

$$\min_{1 \leq t \leq T} \mathbb{E} \|\mathcal{G}^t\|^2 \lesssim \mathcal{O}\left(T^{-\frac{1}{2}}\right).$$

**Remark 4.** *Indeed, Zhou et al. (2022) demonstrates that the average gradient $\frac{1}{T} \sum_{t=1}^T \mathbb{E} \|\mathcal{G}^t\|^2$ of the policy gradient estimation converges to a small constant as $T \to \infty$. With the help of refined step size $\eta^t = \frac{c}{\sqrt{t}}$, our results in Theorem 1 shows better convergence property w.r.t. $T$. The empirical and theoretical analysis of algorithmic computation complexity is left in* Appendix B.7 & E.5.

## 3.2 GENERALIZATION ERROR ANALYSIS

The expected risk of $f : \mathcal{X} \to \mathcal{Y}$, w.r.t. $\mathcal{E}_{\mathbf{z}}(f)$ in (2), is measured by

$$\mathcal{E}(f) = \int_{\mathcal{X} \times \mathcal{Y}} (f(x) - y)^2 d\rho(x, y).$$

It is well known that

$$f_\rho = \int_{\mathcal{Y}} y \, d\rho(y|\cdot)$$

is the minimizer of $\mathcal{E}(f)$ over all measurable functions, where $\rho(y|x)$ denotes the conditional distribution of $y$ for given $x$. This work describes how fast $f_{\mathbf{z}}$ defined in (2) approximates $f_\rho$ as the number of samples increases. As far as we know, this is the first theoretical endeavor to analyze the generalization behavior of semi-supervised additive models.

Before presenting our results, we recall some necessary assumptions and definitions involved here, which have been widely used in bounding the excess risk for supervised learning algorithms (Shi et al., 2011; Shi, 2013; Christmann & Zhou, 2016; Wang et al., 2023; Deng et al., 2023) and SSL models (Belkin et al., 2006; Liu & Chen, 2018; Chen et al., 2018).

**Assumption 2.** *(Christmann & Zhou (2016)) For any $x \in \mathcal{X}$, there exists some $M \geq 0$ such that $\rho(\cdot \mid x)$ is almost everywhere supported on $[-M, M]$. Assume $f_\rho = \sum_{j=1}^{p} f_\rho^{(j)}$ with $0 < r \leq \frac{1}{2}$ and $f_\rho^{(j)} = L_{K^{(j)}}^r \left(g_j^*\right)$ with some $g_j^* \in L_2(\rho(\mathcal{X}^{(j)}))$ for any $j \in \{1, \ldots, p\}$, where $L_2(\rho(\mathcal{X}^{(j)}))$ is the square-integrable space on $\mathcal{X}^{(j)}$ and $L_{K^{(j)}}^r$ is $r$-power of integral operator $L_{K^{(j)}} : L_2(\rho(\mathcal{X}^{(j)})) \to L_2(\rho(\mathcal{X}^{(j)}))$ associated with kernel $K^{(j)}$.*

**Assumption 3.** *Each entry of similarity matrix $\boldsymbol{W}$ satisfies $0 \leq W_{ij} \leq w$ for positive constant $w$.*

**Assumption 4.** *Let $C^v$ be a $\nu$-times continuously differentiable function set. Assume that $K^{(j)} \in C^\nu \left(\mathcal{X}^{(j)} \times \mathcal{X}^{(j)}\right), j \in \{1, \ldots, p\}$.*

Define $\pi(f)(x) = \max\{\min\{f(x), M\}, -M\}, \forall f \in \mathcal{H}$. This truncated operator has been used extensively for error analysis of learning algorithms, see e.g., (Steinwart et al., 2009; Shi et al., 2019). Since $\mathcal{E}(\pi(f)) \leq \mathcal{E}(f)$ for any $f \in \mathcal{H}$, here we state the upper bound of $\mathcal{E}(\pi(f_{\mathbf{z}})) - \mathcal{E}(f_\rho)$ to get a tighter generalization characterization for the manifold regularized additive model in (2).

**Theorem 2.** *Let $\lambda_1 = (l+u)^{-\Delta}$, $\lambda_2 = \lambda_1^{1-r}$ for some $\Delta > 0$ and $0 < r \leq 1/2$. Under Assumptions 2-4, for any $0 < \delta < 1/2$, with confidence at least $1 - 2\delta$, there holds*

$$\mathcal{E}(\pi(f_{\mathbf{z}})) - \mathcal{E}(f_\rho) \lesssim \log(\frac{\delta}{8})\mathcal{O}\left(l^{-\Theta}\right),$$

*where $\Theta = \min\{\Delta r, 2/(2+\zeta), r + \Delta(r-1)\}$ with $\zeta = \begin{cases} \frac{2}{1+2v}, & v \in (0,1] \\ \frac{2}{1+v}, & v \in (1, 3/2] \\ \frac{1}{v}, & v \in (3/2, \infty) \end{cases}$ .*

**Remark 5.** *Theorem 2 guarantees the learning rate $\mathcal{O}(l^{-1/4})$ as setting $\Lambda = r = 1/2$ and $v \to \infty$. Besides the additional advantage of the interpretability of input variables, the basic model (2) of S$^2$MAM also achieves the polynomial decay rate of excess risk, which is comparable with SSL linear models (Chen et al., 2018).*

# 4 EXPERIMENTAL EVALUATIONS

This section validates the effectiveness of S$^2$MAM on simulated and real-world data. All experiments are implemented in Python. More results on images and sensitivity analysis are left in *Appendix B*.

## 4.1 BASELINES AND PARAMETER SELECTION

For the regression tasks, we compare the proposed S$^2$MAM with sparse supervised models (Lasso (Tibshirani, 1994) and SpAM (Ravikumar et al., 2009)), Deep Analytic Networks (DAN) (Dinh & Ho, 2020), LapRLS (Belkin et al., 2006), co-training regressor (COREG) (Zhou & Li, 2005) and deep SSL methods including the variational autoencoder (VAE) (Goodfellow et al., 2014) and the semi-supervised deep kernel learning (SSDKL) (Jean et al., 2018). For simplicity, the squared loss is selected as the loss function for SpAM and S$^2$MAM. The supervised methods are trained with merely labeled data. The mean squared error (MSE) and R-squared score with standard deviation information are used as the performance criterion.

For classification, the competitors include $\ell_1$-SVM (Zhu et al., 2003a), SpAM (with logistic loss) (Ravikumar et al., 2009), LapSVM (Belkin et al., 2006), f-FME (Qiu et al., 2018), AWSSL (Nie et al., 2019), RGL (Kang et al., 2020), SALE (Nie et al., 2021), Correntropy-based Sparse Additive Machine (CSAM) (Yuan et al., 2023), Tilted Sparse Additive Model (TSpAM) (Wang et al., 2023) and semi-supervised neural processes (SSNP) (Wang et al., 2022a). S$^2$MAM is equipped with the logistic loss. The 1-nearest neighbor classifier with Euclidean distance is employed in f-FME and AWSSL. Similarity measure $W_{ij} = \exp\{-\|x_i - x_j\|_2^2/\mu^2\}$ and accuracy criterion are exploited.

For fairness, the penalty coefficients $\lambda_1$ and $\lambda_2$ are tuned across $[10^{-4}, 10^{-3}, 10^{-2}, 10^{-1}]$, which are shared for all compared methods. Let $\tau_j = 1$ for all $j \in [1, 2, \cdots, p]$ for additive baselines (Wang et al., 2023). The bandwidth $\mu$ for similarity measure is selected within $[10^{-4}, 10^{-3}, 10^{-2}, 10^{-1}, 1]$. We repeat each experiment for 100 times and report the average accuracy as well as the standard deviation under different data settings. The numbers of selected variables $C$ and neighbors are shared for all SSL baselines on different data. The parameters for the other methods were set according to the corresponding references.

Table 2: Average MSE ± standard deviation on synthetic regression data with different label percentages ($r$) and noisy variable numbers ($p_n$). The upper and lower tables show the results on Friedman data and additive data.

| Model | r = 5%, $p_n$ = 0 | | r = 5%, $p_n$ = 10 | | r = 10%, $p_n$ = 0 | | r = 10%, $p_n$ = 10 | |
| --- | --- | --- | --- | --- | --- | --- | --- | --- |
| | Unlabeled | Test | Unlabeled | Test | Unlabeled | Test | Unlabeled | Test |
| Lasso | - | 15.579 ± 12.396 | - | 22.135 ± 14.442 | - | 8.684 ± 2.393 | - | 15.636 ± 7.785 |
| SpAM | - | 14.791 ± 11.595 | - | 21.055 ± 13.744 | - | 8.201 ± 2.464 | - | 14.706 ± 7.577 |
| DAN | - | 12.417 ± 7.947 | - | 23.350 ± 7.074 | - | 7.864 ± 2.017 | - | 17.392 ± 5.283 |
| LapRLS | 11.659 ± 5.024 | 11.678 ± 5.125 | 27.299 ± 8.549 | 27.588 ± 8.779 | 8.086 ± 2.000 | 8.103 ± 1.970 | 23.822 ± 4.498 | 23.918 ± 4.457 |
| VAE | 11.071 ± 7.011 | 11.499 ± 7.971 | 20.194 ± 9.477 | 20.860 ± 9.977 | 7.866 ± 3.752 | 7.950 ± 4.873 | 15.155 ± 4.950 | 15.809 ± 5.134 |
| COREG | 10.573 ± 6.855 | **10.730 ± 6.946** | 19.011 ± 7.644 | 19.644 ± 7.945 | 7.801 ± 3.011 | 7.820 ± 3.401 | 15.305 ± 4.117 | 15.914 ± 4.955 |
| SSDKL | **10.144 ± 6.917** | 10.744 ± 7.301 | 19.410 ± 7.809 | 19.655 ± 8.137 | **7.035 ± 7.155** | **7.195 ± 7.511** | 14.101 ± 4.055 | 14.731 ± 4.773 |
| S²MAM (ours) | 10.837 ± 4.355 | 11.350 ± 4.881 | **12.274 ± 5.101** | **12.941 ± 5.807** | 7.204 ± 2.591 | 7.430 ± 2.473 | **8.418 ± 3.140** | **8.701 ± 3.433** |
| Lasso | - | 1.193 ± 0.437 | - | 2.706 ± 3.174 | - | 1.079 ± 0.304 | - | 2.102 ± 0.705 |
| SpAM | - | 1.122 ± 0.422 | - | 2.597 ± 2.848 | - | 1.033 ± 0.301 | - | 1.955 ± 0.727 |
| DAN | - | 1.217 ± 0.346 | - | 2.133 ± 1.294 | - | 1.014 ± 0.232 | - | 1.792 ± 0.538 |
| LapRLS | 1.025 ± 0.121 | 1.073 ± 0.182 | 3.571 ± 0.138 | 3.592 ± 0.171 | 0.986 ± 0.136 | 1.055 ± 0.181 | 3.101 ± 0.104 | 3.122 ± 0.166 |
| VAE | 1.117 ± 0.569 | 1.126 ± 0.590 | 1.433 ± 0.622 | 1.573 ± 0.662 | 0.991 ± 0.233 | 1.103 ± 0.247 | 1.341 ± 0.305 | 1.379 ± 0.337 |
| COREG | **0.959 ± 0.237** | **0.974 ± 0.295** | 1.137 ± 0.306 | 1.255 ± 0.411 | **0.937 ± 0.209** | **0.961 ± 0.104** | 1.059 ± 0.287 | 1.141 ± 0.388 |
| SSDKL | 0.992 ± 0.221 | 1.046 ± 0.269 | 1.312 ± 0.411 | 1.344 ± 0.462 | 0.959 ± 0.210 | 0.983 ± 0.233 | 1.247 ± 0.359 | 1.287 ± 0.394 |
| S²MAM (ours) | 0.982 ± 0.117 | 1.027 ± 0.162 | **1.093 ± 0.210** | **1.178 ± 0.281** | 0.944 ± 0.106 | 0.970 ± 0.146 | **0.979 ± 0.147** | **1.094 ± 0.240** |

Table 3: Average Accuracy ± standard deviation (%) on synthetic classification data with fixed label percentages in each class ($r = 5\%$), uninformative variable ($p_u$) and noisy variable numbers ($p_n$). Upper and lower tables show the results of moon data and additive data.

| Model | r = 5%, $p_u = p_n = 0$ | | r = 5%, $p_u = 10, p_n = 0$ | | r = 5%, $p_u = 0, p_n = 10$ | | r = 5%, $p_u = p_n = 10$ | |
| --- | --- | --- | --- | --- | --- | --- | --- | --- |
| | Unlabeled | Test | Unlabeled | Test | Unlabeled | Test | Unlabeled | Test |
| $\ell_1$-SVM | - | 83.917 ± 1.949 | - | 78.631 ± 6.737 | - | 60.183 ± 10.243 | - | 55.872 ± 8.377 |
| SpAM | - | 84.122 ± 1.626 | - | 76.021 ± 5.434 | - | 62.307 ± 9.590 | - | 54.481 ± 7.808 |
| CSAM | - | 85.309 ± 1.216 | - | 77.611 ± 4.790 | - | 65.698 ± 7.139 | - | 64.714 ± 7.211 |
| TSpAM | - | 85.729 ± 1.436 | - | 79.183 ± 4.260 | - | 67.064 ± 6.833 | - | 65.592 ± 7.148 |
| LapSVM | 88.635 ± 3.307 | 86.395 ± 2.825 | 69.261 ± 6.064 | 69.670 ± 5.941 | 50.083 ± 4.989 | 51.011 ± 5.001 | 49.026 ± 1.150 | 50.000 ± 0.000 |
| f-FME | 89.201 ± 1.955 | 87.370 ± 2.070 | 71.631 ± 5.255 | 72.314 ± 5.061 | 53.083 ± 5.109 | 54.171 ± 5.411 | 51.026 ± 6.598 | 51.231 ± 6.919 |
| AWSSL | **93.171 ± 1.801** | 92.395 ± 1.977 | 87.549 ± 2.701 | 87.616 ± 2.844 | 79.810 ± 3.577 | 79.901 ± 3.650 | 77.301 ± 3.944 | 77.368 ± 4.050 |
| RGL | 91.127 ± 2.497 | 90.804 ± 2.781 | 88.311 ± 3.030 | 87.914 ± 3.152 | 81.706 ± 3.951 | 81.254 ± 4.077 | 79.176 ± 4.511 | 78.679 ± 4.989 |
| SALE | 91.104 ± 2.060 | 90.799 ± 2.135 | 88.915 ± 2.944 | 88.193 ± 3.029 | 82.791 ± 3.464 | 82.199 ± 3.891 | 80.988 ± 5.066 | 80.489 ± 5.066 |
| SSNP | 92.720 ± 2.184 | **92.437 ± 2.237** | 88.642 ± 2.847 | **88.306 ± 3.195** | 81.244 ± 4.230 | 80.859 ± 4.406 | 79.287 ± 5.026 | 79.310 ± 5.211 |
| S²MAM (ours) | 91.195 ± 1.919 | 91.877 ± 2.207 | **89.704 ± 2.414** | 88.255 ± 2.873 | **83.013 ± 4.097** | **83.454 ± 4.388** | **81.636 ± 4.240** | **81.950 ± 4.713** |
| $\ell_1$-SVM | - | 83.914 ± 6.410 | - | 62.713 ± 6.098 | - | 62.261 ± 6.550 | - | 54.791 ± 6.951 |
| SpAM | - | 84.150 ± 6.104 | - | 65.091 ± 5.917 | - | 64.814 ± 6.039 | - | 54.413 ± 6.295 |
| CSAM | - | 86.597 ± 5.424 | - | 69.717 ± 5.101 | - | 65.178 ± 5.255 | - | 61.980 ± 5.701 |
| TSpAM | - | 86.993 ± 5.340 | - | 71.044 ± 5.079 | - | 67.340 ± 4.959 | - | 63.145 ± 5.130 |
| LapSVM | 88.814 ± 5.398 | 88.850 ± 5.269 | 59.992 ± 5.259 | 60.325 ± 5.184 | 55.630 ± 8.213 | 55.957 ± 8.292 | 55.137 ± 8.414 | 55.203 ± 8.496 |
| f-FME | 89.141 ± 3.172 | 89.305 ± 3.359 | 64.495 ± 4.033 | 64.611 ± 4.208 | 59.671 ± 6.473 | 59.801 ± 6.655 | 59.311 ± 6.602 | 59.407 ± 6.659 |
| AWSSL | 91.259 ± 2.871 | 90.211 ± 3.077 | 83.691 ± 3.423 | 83.950 ± 3.519 | 73.701 ± 4.105 | 73.859 ± 4.322 | 72.255 ± 4.211 | 72.370 ± 4.428 |
| RGL | 90.422 ± 2.909 | 90.026 ± 3.477 | 84.065 ± 4.501 | 84.879 ± 4.711 | 77.726 ± 4.591 | 78.041 ± 4.510 | 75.155 ± 4.965 | 75.413 ± 4.708 |
| SALE | 89.717 ± 2.811 | 90.149 ± 2.665 | 85.742 ± 4.132 | 85.971 ± 4.018 | 79.071 ± 4.709 | 79.844 ± 4.277 | 77.201 ± 4.697 | 77.891 ± 4.431 |
| SSNP | **90.492 ± 3.059** | 89.871 ± 3.218 | **86.130 ± 3.922** | 85.908 ± 4.105 | 78.250 ± 4.294 | 78.062 ± 4.133 | 77.462 ± 4.412 | 77.601 ± 5.513 |
| S²MAM (ours) | 89.979 ± 3.255 | **90.309 ± 3.409** | 85.517 ± 3.481 | **86.015 ± 3.575** | **81.702 ± 3.897** | **81.855 ± 4.055** | **80.012 ± 4.177** | **80.112 ± 4.370** |

## 4.2 EXPERIMENTS ON SYNTHETIC DATA

**Semi-supervised Regression:** The Friedman dataset (Friedman, 1991) owns $p^* = 5$ informative variables, and is generated by $y = 10\sin(\pi x^{(1)}x^{(2)}) + 20(x^{(3)} - 0.5)^2 + 10x^{(4)} + 5x^{(5)} + \epsilon$, where each $x^{(j)} \sim U(0, 1)$ and $\epsilon \sim \mathcal{N}(0, 1)$.

The additive data (Ravikumar et al., 2009; Chen et al., 2020; Wang et al., 2023) is generated from $y = \sum_{j=1}^{8} f^{(j)}(x^{(j)}) + \epsilon$, where $f^{(1)}(u) = -2\sin(2u)$, $f^{(2)}(u) = 8u^2$, $f^{(3)}(u) = \frac{7\sin u}{2 - \sin u}$, $f^{(4)}(u) = 6e^{-u}$, $f^{(5)}(u) = u^3 + \frac{3}{2}(u-1)^2$, $f^{(6)}(u) = 5u$, $f^{(7)}(u) = 10\sin(e^{-u/2})$, $f^{(8)}(u) = -10\widetilde{\phi}(u, \frac{1}{2}, \frac{4}{5})$. Here $\widetilde{\phi}$ stands for the normal cumulative distribution with mean of $\frac{1}{2}$ and the standard deviation of $\frac{4}{5}$. We generate $n = 200$ samples with $p^* = 8$ ($p^* = 5$) informative variables and $p_u = 92$ ($p_u = 95$) uninformative variables following $\mathcal{N}(0, 1)$ for the additive data (the Friedman data). To illustrate the impact of noisy variables, additional $p_n = 10$ variables are designed as noisy variables following $\mathcal{N}(100, 100)$ for simplicity. The whole dataset is then equally split into training and testing sets, where merely 10% or 20% samples still keep their labels in the training set.

As shown in Table 2, S²MAM enjoys competitive or even the best performance over the baselines. Under clean scenarios without corruption, some deep SSL baselines may perform slightly better, which is understandable due to their strong approximation ability and reliance on high-quality training data. Especially under the variable corruptions, our model owns the smallest MSE as well as standard deviation, which implies S²MAM can identify most of the truly active variables by assigning the right mask. As validation in *Appendix B.5*, these supervised baselines require larger labeled counterparts.

Table 4: Average R-squared score $\pm$ standard deviation on UCI data ("Buzz-Regression", "Boston House", "Ozone" and "SkillCraft") with 10% labeled training samples and 10 noisy variables for regression.

| Model | Buzz-Regression | | Boston House | | Ozone | | SkillCraft | |
|---|---|---|---|---|---|---|---|---|
| | Unlabeled | Test | Unlabeled | Test | Unlabeled | Test | Unlabeled | Test |
| Lasso | - | $0.773 \pm 0.433$ | - | $0.526 \pm 0.571$ | - | $-1.025 \pm 3.630$ | - | $0.515 \pm 0.149$ |
| SpAM | - | $0.747 \pm 0.542$ | - | $0.530 \pm 0.672$ | - | $0.324 \pm 3.395$ | - | $0.522 \pm 0.191$ |
| DAN | - | $0.781 \pm 0.370$ | - | $0.516 \pm 0.503$ | - | $0.511 \pm 1.926$ | - | $0.519 \pm 0.134$ |
| LapRLS | $0.711 \pm 0.377$ | $0.702 \pm 0.392$ | $0.522 \pm 0.193$ | $0.510 \pm 0.217$ | $0.574 \pm 0.278$ | $0.563 \pm 0.304$ | $0.504 \pm 0.127$ | $0.498 \pm 0.132$ |
| VAE | $0.742 \pm 2.871$ | $0.736 \pm 2.951$ | $0.546 \pm 3.720$ | $0.541 \pm 2.807$ | $0.591 \pm 2.041$ | $0.584 \pm 2.259$ | $0.529 \pm 0.511$ | $0.522 \pm 0.519$ |
| COREG | $0.771 \pm 2.142$ | $0.761 \pm 2.216$ | $0.565 \pm 1.836$ | $0.561 \pm 1.862$ | $0.595 \pm 1.320$ | $0.589 \pm 1.452$ | $0.538 \pm 0.431$ | $0.530 \pm 0.438$ |
| SSDKL | $0.764 \pm 3.104$ | $0.749 \pm 3.277$ | $0.537 \pm 2.541$ | $0.522 \pm 2.679$ | $0.602 \pm 1.655$ | $0.590 \pm 1.712$ | $0.546 \pm 0.831$ | $0.541 \pm 0.840$ |
| $S^2$MAM (ours) | $\mathbf{0.812 \pm 1.255}$ | $\mathbf{0.804 \pm 1.278}$ | $\mathbf{0.621 \pm 0.866}$ | $\mathbf{0.610 \pm 0.879}$ | $\mathbf{0.644 \pm 0.386}$ | $\mathbf{0.631 \pm 0.397}$ | $\mathbf{0.558 \pm 0.265}$ | $\mathbf{0.551 \pm 0.271}$ |

Table 5: Average Accuracy $\pm$ standard deviation (%) on UCI data ("Buzz-Classification", "Breast Cancer", "Phishing Websites" and "Statlog Heart") with 10% labeled samples and 10 noisy variables for classification.

| Model | Buzz-Classification | | Breast Cancer | | Phishing Websites | | Statlog Heart | |
|---|---|---|---|---|---|---|---|---|
| | Unlabeled | Test | Unlabeled | Test | Unlabeled | Test | Unlabeled | Test |
| $\ell$-1 SVM | - | $72.882 \pm 9.734$ | - | $74.994 \pm 8.531$ | - | $55.918 \pm 5.575$ | - | $67.251 \pm 9.143$ |
| SpAM | - | $75.068 \pm 7.455$ | - | $79.943 \pm 6.824$ | - | $57.701 \pm 5.311$ | - | $69.989 \pm 9.744$ |
| CSAM | - | $77.213 \pm 5.622$ | - | $81.408 \pm 5.134$ | - | $60.097 \pm 4.201$ | - | $73.319 \pm 8.202$ |
| TSpAM | - | $79.225 \pm 5.412$ | - | $82.260 \pm 5.042$ | - | $60.471 \pm 4.030$ | - | $74.471 \pm 7.207$ |
| LapSVM | $70.864 \pm 12.250$ | $70.214 \pm 12.738$ | $61.553 \pm 9.502$ | $61.114 \pm 9.810$ | $51.700 \pm 5.306$ | $51.342 \pm 5.395$ | $58.025 \pm 5.427$ | $57.984 \pm 5.470$ |
| f-FME | $82.759 \pm 5.692$ | $82.302 \pm 5.741$ | $75.261 \pm 6.740$ | $75.204 \pm 6.862$ | $76.623 \pm 3.695$ | $76.594 \pm 3.710$ | $74.998 \pm 4.217$ | $74.903 \pm 4.236$ |
| AWSSL | $89.672 \pm 5.310$ | $89.155 \pm 5.412$ | $77.197 \pm 6.025$ | $77.120 \pm 6.136$ | $78.025 \pm 4.257$ | $77.989 \pm 4.303$ | $76.622 \pm 4.773$ | $76.595 \pm 4.914$ |
| RGL | $90.219 \pm 4.916$ | $90.020 \pm 5.173$ | $86.302 \pm 5.894$ | $86.044 \pm 6.013$ | $78.103 \pm 4.271$ | $78.011 \pm 4.630$ | $78.230 \pm 4.206$ | $78.088 \pm 4.317$ |
| SALE | $\mathbf{91.064 \pm 4.617}$ | $90.671 \pm 4.832$ | $86.252 \pm 4.904$ | $86.030 \pm 5.088$ | $80.130 \pm 3.977$ | $79.878 \pm 4.121$ | $77.971 \pm 4.062$ | $77.807 \pm 4.217$ |
| SSNP | $90.040 \pm 4.107$ | $89.312 \pm 4.383$ | $84.195 \pm 5.251$ | $82.836 \pm 5.301$ | $80.672 \pm 3.472$ | $80.183 \pm 3.711$ | $76.595 \pm 5.650$ | $75.722 \pm 4.315$ |
| $S^2$MAM (ours) | $92.618 \pm 4.377$ | $\mathbf{92.431 \pm 4.526}$ | $\mathbf{88.053 \pm 4.935}$ | $\mathbf{87.995 \pm 4.947}$ | $\mathbf{81.992 \pm 2.514}$ | $\mathbf{81.894 \pm 2.527}$ | $\mathbf{79.498 \pm 4.119}$ | $\mathbf{79.277 \pm 4.171}$ |

**Semi-supervised Classification:** Following the experimental design in (Chen et al., 2020; Wang et al., 2023), we consider the additive discriminant function $f^*(x_i) = (x_i^{(1)} - 0.5)^2 + (x_i^{(2)} - 0.5)^2 - 0.08$, where $x_i^{(j)} = (W_{ij} + U_i)/2$. $W_{ij}$ and $U_i$ are independently from $U(0,1)$ for $i = 1, \cdots, 200$, $j = 1, \cdots, 100$. The label satisfies $y_i = 0$ when $f(x_i) \leq 0$ and 1 otherwise.

To evaluate the robustness of $S^2$MAM, $p_n$ irrelevant variables are designed as noisy variables following $\mathcal{N}(100, 100)$. After equally dividing the whole data into the training and testing sets, 5% or 10% samples for each class from the training set are randomly selected as the labeled set. As shown in Table 3, our method often enjoys better performance than the other baselines, especially in the case with noisy variables.

### 4.3 EXPERIMENTS ON REAL-WORLD DATA

This subsection states the empirical evaluations of $S^2$MAM on eight real-world datasets from UCI repository (Asuncion & Newman, 2007), which have been widely used in recent SSL works (Jean et al., 2018; Nie et al., 2019). Tables 4 for regression demonstrates that $S^2$MAM enjoys competitive performance and even stronger robustness against variable corruptions compared to the other baselines, e.g., average 0.088 higher R-squared score on corrupted Boston House. Even with corrupted training data in Table 5 for classification, $S^2$MAM still owns better prediction accuracy and stronger stability with the smallest variance than those supervised or semi-supervised competitors.

We state the detailed descriptions of employed data and competitors, ablation and sensitivity analysis, and empirical results on other empirical settings in *Appendixes B.1-B.4*, due to the space limitation. Interpretability visualization results and high-dimensional applications (e.g., images) with time cost analysis are present in *Appendixes B.6 & B.7*, respectively.

## 5 CONCLUSION

This paper proposes a semi-supervised meta additive model, called $S^2$MAM, to improve the robustness and interpretability of manifold regularization (Belkin et al., 2006) under the redundant and noisy input variable settings. Compared with the existing SSL with manifold regularization (Belkin et al., 2006; Nie et al., 2019), the proposed approach is capable of realizing variable selection, interpretable and robust estimation simultaneously. Theoretical and empirical evaluations verify its superiority over some state-of-the-art learning models.

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

# Appendix

## A  NOTATIONS

Some used notations are summarized in Table 6.

Table 6: Notations

| Notations | Descriptions |
|---|---|
| $p$ | the dimension of the input |
| $\mathcal{X}, \mathcal{Y}$ | the input space $\mathcal{X} = \{\mathcal{X}^{(1)}, \cdots, \mathcal{X}^{(p)}\} \in \mathbb{R}^p$ and the output space $\mathcal{Y} \subset \mathbb{R}$, respectively |
| $\rho$ | the jointed distribution on $\mathcal{X} \times \mathcal{Y}$ |
| $\rho_{\mathcal{X}}$ | the marginal distribution with respect to $\mathcal{X}$ induced by $\rho$ |
| $l/u$ | the number of labeled / unlabeled samples |
| $x_i; y_i$ | input $x_i = (x_i^{(1)}, \cdots, x_i^{(p)})^T \in \mathbb{R}^p$ with $x_i^{(j)} \in \mathcal{X}^{(j)}$; output $y_i \in \mathcal{Y}$ |
| $\mathbf{z}_l; \mathbf{z}_u$ | the labeled dataset $\mathbf{z}_l = \{(x_i, y_i)\}_{i=1}^l$; the unlabeled dataset $\mathbf{z}_u = \{x_i\}_{i=l+1}^{l+u}$ |
| $\mathcal{H}$ | the hypothesis space $\mathcal{H} = \left\{ f = \sum_{j=1}^p f^{(j)} : f^{(j)} \in \mathcal{H}_{K^{(j)}}, 1 \le j \le p \right\}$ |
| $\mathcal{H}_{K^{(j)}}$ | the RKHS associated with Mercer kernel $K^{(j)}$ defined on $\mathcal{X}^{(j)} \times \mathcal{X}^{(j)}, j \in \{1, \ldots, p\}$ |
| $L_{K^{(j)}}$ | integral operator $L_{K^{(j)}} : L_2(\rho(\mathcal{X}^{(j)})) \to L_2(\rho(\mathcal{X}^{(j)}))$ based on the square-integrable space $L_2$ |
| $L_{K^{(j)}}^r$ | the $r$-power of $L_{K^{(j)}}$ associated with feature $\mathcal{X}^{(j)}$ and kernel $K^{(j)}$ |
| $f(\cdot)$ | the prediction function of supervised additive models in RKHS where $f(\cdot) = \sum_{j=1}^p \sum_{i=1}^l \alpha_i^{(j)} K_i^{(j)}(x_i^{(j)}, \cdot)$ |
| $f^*$ | the ground truth function |
| $\mathbf{f}$ | the prediction vector $\mathbf{f} = (f(x_1), \ldots, f(x_{l+u}))^T$, associated with $\mathbf{z}_l$ and $\mathbf{z}_u$ |
| $f_{\mathbf{z}}$ | the empirical decision function of manifold regularized additive model |
| $\tau_j$ | the weight of $j$-th variable |
| $\alpha$ | the coefficient of the lower level additive model |
| $\mathbf{W}$ | the similarity matrix for SSL tasks |
| $\mathbf{D}; \mathbf{L}$ | the diagonal matrix $D_{ii} = \sum_{j=1}^{l+u} W_{ij}$; the graph Laplacian $\mathbf{L} = \mathbf{D} - \mathbf{W}$ |
| $\mathbf{m}$ | the variable mask vector $\mathbf{m} \in \{0, 1\}^p$ |
| $\mathbf{s}$ | the vector $\mathbf{s} = (s_1, \cdots, s_p)$ where $s_i$ stands for the probability of $m_i = 1$ |

## B  DESCRIPTIONS FOR BENCHMARKS AND BASELINES & ADDITIONAL EXPERIMENTAL RESULTS

In this paper, 4 synthetic data and 9 real-world data are selected in our experiments. Indeed, these datasets have been widely used for validating additive models (Ravikumar et al., 2009; Lahiri et al., 2016; Chen et al., 2020; Wang et al., 2023) or semi-supervised learning models (Jean et al., 2018; Qiu et al., 2018; Nie et al., 2019; 2021; Bao et al., 2024). We briefly summarize these used datasets and some learning methods for baselines as follows.

### B.1  DATA DESCRIPTION

Denote $N$ and $p$ ($p = p^* + p_u + p_n$) as the total number of samples and the dimensions in each individual dataset, where the training set involves $l$ labeled data and $u$ unlabeled data, and the remained samples are left for testing. We generate $p_u$ uninformative variables and $p_n$ noisy variables, which are added into the truly informative variables $p^*$ from all samples within the dataset (including the training and testing sets).

The datasets used in this paper include:

- Friedman data for regression. The corresponding generation function is provided in the experiment section, which involves 200 samples, $p^* = 5$ true informative features, and

$p_u = 95$ uninformative features. And $p_n = 10$ noisy features are also considered to better highlight the robustness. Denote $\epsilon$ as the Gaussian noise $\mathcal{N}(0, 1)$, the output $y$ is generated by

$$f(X) = 10\sin\left(\pi X^{(1)}X^{(2)}\right) + 20\left(X^{(3)} - 0.5\right)^2 + 10X^{(4)} + 5X^{(5)} + \epsilon.$$

- Synthetic additive data for regression. It involves $N = 200$ samples, $p^* = 8$ true informative features, and $p_u = 92$ uninformative features. We also consider adding $p_n = 10$ noisy features following $\mathcal{N}(100, 100)$ into the whole dataset,

$$Y = f^*(X) + \epsilon = \sum_{j=1}^{8} f^{(j)}(X^{(j)}) + \epsilon, \tag{10}$$

where $f^{(1)}(u) = -2\sin(2u)$, $f^{(2)}(u) = 8u^2$, $f^{(3)}(u) = \frac{7\sin u}{2 - \sin u}$, $f^{(4)}(u) = 6e^{-u}$, $f^{(5)}(u) = u^3 + \frac{3}{2}(u - 1)^2$, $f^{(6)}(u) = 5u$, $f^{(7)}(u) = 10\sin(e^{-u/2})$, $f^{(8)}(u) = -10\tilde{\phi}(u, \frac{1}{2}, \frac{4}{5}^2)$. Notably, in order to validate the additive models on testing sets, the Gram matrices or new splined features for testing sets are required to be generated.

- Synthetic additive data for classification. It involves $N = 200$ samples, $p^* = 2$ informative features, $p_u = 98$ uninformative redundant features following $\mathcal{N}(0, 1)$ and $p_n = 10$ noisy features following $\mathcal{N}(100, 100)$, and the output

$$f^*(x_i) = (x_i^{(1)} - 0.5)^2 + (x_i^{(2)} - 0.5)^2 - 0.08,$$

where $x_i^{(j)} = (W_{ij} + U_i)/2$. $W_{ij}$ and $U_i$ are independently from $U(0, 1)$ for $i = 1, \cdots, 200$, $j = 1, \cdots, 100$. The label satisfies $y_i = 0$ when $f(x_i) \le 0$ and 1 otherwise. This synthetic data for classification has been widely used in some existing research for evaluating the performance of additive models (Chen et al., 2020; Wang et al., 2023)

- Synthetic Moon data for classification. It involves two classes with totally 200 samples, $p^* = 2$ informative features, $p_u =$ uninformative redundant features, and $p_n =$ additional noisy features. This data has been widely used for estimating the model's capability for correctly identifying different categories (Qiu et al., 2018; Nie et al., 2019; 2021).

- Four datasets from the UCI repository for regression.

  1) Buzz prediction on Twitter dataset for regression. It involves totally 38393 samples, $p^* = 77$ original features, and additional $p_n = 10$ noisy features. This dataset helps to predict the mean number of active discussions.

  2) Boston Housing Price dataset for regression. It involves merely 506 samples, $p^* = 13$ original features, and additional $p_n = 10$ noisy features. This dataset has been widely used for estimating the performance of regression models.

  3) Ozone Level Detection dataset for regression. It includes $N = 2536$ instances with $p^* = 73$ attributes, which aims to forecast the ground ozone pollution using the given features. We also add $p_n = 10$ noisy features into the original dataset.

  4) SkillCraft Master dataset for regression. The dataset is made of $N = 3395$ observations and $p^* = 19$ input variables. And $p_n = 10$ noisy features are further added to the original dataset.

- Four datasets from the UCI repository for classification.

  1) Predicting Buzz Magnitude in Social Media dataset for classification. It involves $N = 38393$ instances with $p^* = 77$ original features. We further add $p_n = 10$ noisy features into the original datasets for comparing the robustness of these baselines.

  2) Breast Cancer Wisconsin dataset for classification. There are 569 instances and $p^* = 29$ original input features. $p_n = 10$ noisy features following $\mathcal{N}(100, 100)$ are further added into the original dataset.

  3) Phishing Websites dataset for classification. It contains 31 columns, with 30 features and 1 target. The dataset has 2456 observations.

  4) Statlog (Heart) dataset for classification. It involves $N = 270$ instances with $p^* = 13$ input features. Noisy features are further added for comparison.

- The image data from the COIL20 image library, which originally contains 20 objects, for classification. For simplicity, the 12th and 13th digits are selected, where there are $N = 72$ instances for each digit and $p^* = 16384$ original features (gray images with the size of 128*128). This dataset has been used for evaluating the prediction performance of semi-supervised learning models on feature reduction (Nie et al., 2019; 2021).

Above real-world datasets have undergone preliminary data cleaning, where those entries with empty values are filled with mean values, or even removed when major features are missing (ratio of missing features $\geq 20\%$).

## B.2 BASELINES & PARAMETER SETTINGS

### B.2.1 REGRESSION TASKS

The baselines for regression tasks include:

- Lasso (Tibshirani, 1994), is a type of supervised linear regression model that is used for variable selection with sparsity-induced regularization. The regularization parameter $\lambda$ is tuned across $[10^{-4}, 10^{-3}, 10^{-2}, 10^{-1}, 1]$.

- SpAM (Ravikumar et al., 2009), is an additive supervised nonparametric model for high-dimensional nonparametric regression and classification tasks. The regularization parameter $\lambda$ is tuned across $[10^{-4}, 10^{-3}, 10^{-2}, 10^{-1}, 1]$.

- DAN (Dinh & Ho, 2020) is designed to identify a subset of relevant features in deep learning models. The core technology involves the use of the adaptive group Lasso selection procedure with group Lasso as the base estimator, which is proven to be selection-consistent for a wide class of networks.

- LapRLS (Belkin et al., 2006), learns a semi-supervised linear model using the labeled data by minimizing a regularized least squares objective function. The regularization term incorporates the graph Laplacian matrix, which captures the smoothness assumption that similar points should have similar labels. The regularization parameters $\lambda_1$ and $\lambda_2$ are both tuned across $[10^{-4}, 10^{-3}, 10^{-2}, 10^{-1}, 1]$.

- Variational autoencoder (VAE) (Goodfellow et al., 2014), is designed as a semi-supervised generative model by first learning an unsupervised embedding of the data and then using the embeddings as input to a supervised multilayer perceptron.

- Co-training regressor (COREG) (Zhou & Li, 2005), is a co-training algorithm for regression tasks that uses two $k$-NN regressors with different distance metrics. During the training process, each regressor generates labels for each other.

- Semi-supervised deep kernel learning (SSDKL) (Jean et al., 2018), is a semi-supervised regression model based on minimizing predictive variance in the posterior regularization framework. It combines the hierarchical learning of networks with the probabilistic modeling capabilities of Gaussian processes.

For fairness, a network with a $[d - 100 - 50 - 50 - 2]$ structure is employed here for the downstream regression task. Following (Jean et al., 2018), the same base network is shared for all deep semi-supervised models including VAE and SSDKL. The learning rates for neural network and Gaussian process are $10^{-3}$ and $10^{-1}$, respectively. The training process of VAE, COREG, and SSDKL follows the settings in (Jean et al., 2018). Besides, the bandwidth $\mu$ for the Gaussian similarity function ($W_{ij} = \exp\{-\|x_i - x_j\|_2^2/\mu^2\}$) is also tuned across $[10^{-4}, 10^{-3}, 10^{-2}, 10^{-1}, 1]$ for all SSL methods for computing the similarity and Laplacian matrices. Notice that the similarity matrix for S$^2$MAM is calculated by $W_{ij} = \exp\{-\|x_i \odot \boldsymbol{m} - x_j \odot \boldsymbol{m}\|_2^2/\mu^2\}$ with learned mask $\boldsymbol{m}$, $i, j \in \{1, 2, \cdots, l+u\}$. In practice, the proportion of labeled points in a single batch is consistent with the settings in the whole training set to avoid empty labeled sets or inconsistency among each batch.

### B.2.2 CLASSIFICATION TASKS

The baselines for classification tasks include:

- $\ell_1$-SVM (Zhu et al., 2003a), is a supervised classification model with $\ell_1$ sparse regularization based on the classical SVM. The regularization parameter $\lambda$ is tuned across $[10^{-4}, 10^{-3}, 10^{-2}, 10^{-1}, 1]$.

- SpAM (induced by logistic loss) (Ravikumar et al., 2009), is equipped with logistic loss for classification, which has been introduced above. Its regularization parameter $\lambda$ is tuned across $[10^{-4}, 10^{-3}, 10^{-2}, 10^{-1}, 1]$.

- LapSVM (Belkin et al., 2006), utilizes the concept of graph Laplacian, which captures the underlying manifold structure of the data. The objective of LapSVM is to find a decision boundary that not only separates the labeled data accurately but also respects the smoothness assumption captured by the graph Laplacian. The regularization parameters $\lambda_1$ and $\lambda_2$ are both tuned across $[10^{-4}, 10^{-3}, 10^{-2}, 10^{-1}, 1]$

- f-FME (Qiu et al., 2018), is an improved version of classical flexible manifold embedding (FME) by employing additional anchor graphs to reduce the time cost and computation burden of FME.

- AWSSL (Nie et al., 2019), is a semi-supervised learning model which constructs an adaptive graph for propagating label information and using special strategies for ranking the importance of variables. An auto-weighting matrix is learned to select informative variables from both labeled and unlabeled data.

- RGL (Kang et al., 2020) constructs a graph from the pristine data derived from restored technology, subsequently utilizing this resilient graph to improve the performance of semi-supervised classification tasks.

- SALE (Nie et al., 2021) merges the processes of adaptive graph formation and label dissemination into a singular optimization framework, simultaneously developing an automatic weighting matrix that discerns and emphasizes significant variables across the entire dataset.

- CSAM (Yuan et al., 2023) exploits the robust error metric based on statistical correntropy measure, which forms a robust additive model for classification with noisy labels.

- TSpAM (Wang et al., 2023) builds a robust additive model with the tilted empirical risk. It's capable of robust estimation and imbalanced classification. Notably, an efficient random Fourier features approach is used to accelerate the kernel-based computation.

- SSNP (Wang et al., 2022a) integrates neural processes with semi-supervised learning for image classification tasks. The innovation lies in adapting NPs, a probabilistic model that approximates Gaussian Processes, to the SSL framework. The CNN structure is slightly modified to satisfy 1D value-based inputs.

For simplicity, the parameter $\tau_j = 1$ for all $j \in \{1, 2, \cdots, p\}$. The regularization parameters for regularized models are all tuned across $[10^{-4}, 10^{-3}, 10^{-2}, 10^{-1}, 1]$. As introduced in (Qiu et al., 2018; Nie et al., 2021; Bao et al., 2024), the 1-nearest neighbor (1NN) classifier with Euclidean distance is suggested to evaluate classification accuracy after dimension reduction. The number of selected variables $C$ is shared for S$^2$MAM and those baselines for dimension reduction.

Inspired by (Qiu et al., 2018; Nie et al., 2021), the PCA method is used to preserve $95\%$ of the information for each dataset. To avoid singular solutions or unfair comparisons, each experiment has been repeated 20 times and the similarity (weight) graph is constructed following (Nie et al., 2019; 2021; Bao et al., 2024) for those baselines with Laplacian matrix. Each dataset is divided into training and testing sets with a ratio of $1 : 1$. Then we select $l$ samples from each class as the labeled set, and the left training samples are considered the unlabeled set. The optimal parameters are selected by the leave-one-out cross-validation, due to the rarity of labeled samples. The parameters for the other methods were set according to their corresponding references (Jean et al., 2018).

## B.3 ABLATION ANALYSIS

To better show the effects of the manifold regularization term, the probabilistic bilevel optimization method, and the additive modeling strategy, we first illustrate the relationship between the three models in Figure 2:

- Manifold Regularized Sparse Additive Model in Section 2.1,

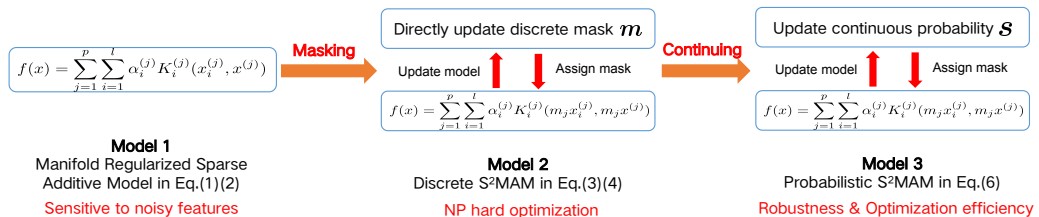

Figure 2: Connections among three models introduced in Section 2.

- Discrete Bilevel Framework for $S^2$MAM in Section 2.2,
- Probabilistic Bilevel Framework for $S^2$MAM in Section 2.3.

We've further conducted extended ablation experiments by:

- removing the manifold regularization term ($\mathbf{f}^T \boldsymbol{L} \mathbf{f}$), named Supervised Meta Additive Model (SMAM);
- removing the upper-level problem (bilevel optimization), called Semi-supervised Additive Model ($S^2$AM);
- removing the additive strategy, named Semi-supervised Meta-based Model ($S^2$MM).

The experiments on the synthetic Friedman data are shown below:

Table 7: Extended ablation experiments by 1) removing the manifold regularization term; 2) removing the upper-level problem (bilevel optimization); 3) removing the additive strategy.

| Models | $r = 10\%$ and $p_n = 0$ | $r = 10\%$ and $p_n = 10$ |
|---|---|---|
| 1) SMAM | 8.319±2.740 | 10.291±3.511 |
| 2) $S^2$AM | 8.041±1.862 | 21.328±4.108 |
| 3) $S^2$MM | 7.861±2.611 | 8.913±3.811 |
| $S^2$MAM | 7.820±2.473 | 8.701±3.433 |

From the results in the above table, one can see that 1) SMAM has the worst performance with few labeled samples and even noisy variables. 2) Without feature corruptions, SSAM has similar performance to $S^2$MAM. Otherwise, SSAM breaks down. 3) Both SSMM and $S^2$MAM are robust to feature corruptions. And $S^2$MAM performs slightly better than SSMM.

It implies that 1) the manifold regularization helps to use the unlabeled samples to learn better prediction functions; 2) the employed bilevel scheme for automatically assigning variable masks is vital to deal with noisy variables; 3) the additive strategy can improve the non-linear approximation ability. And SSMM fails to illustrate the prediction curve of each input variable, since the additive model is important for improving interpretability.

**Remark 6.** *The above results also suggested that, after filtering out effective features using S2MAM, the extracted data can be applied to downstream tasks under an adaptive bandwidth strategy, which can adapt to complex data distributions like imbalanced categories.*

### B.4 EMPIRICAL VALIDATION ON SENSITIVITY & CONVERGENCE

#### B.4.1 IMPACT OF THE NUMBER OF LABELED SAMPLES

Based on the synthetic additive regression data, we first give the sensitivity analysis for the proposed $S^2$MAM on the size of the training set $n$ involving $l$ labeled samples and $u$ unlabeled ones.

As shown in Figures 3, we find that larger size of labeled training data helps to improve the performance of semi-supervised model, which is consistent with our theoretical findings on the generalization error bounds, as well as some existing conclusions of statistical learning theory for supervised

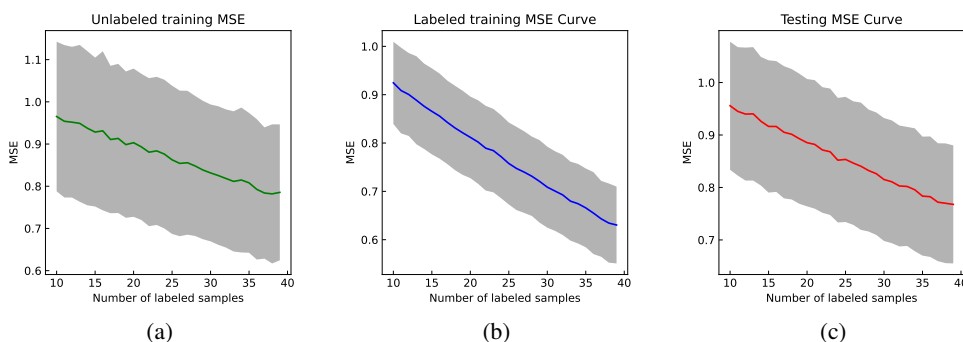

(a)                                          (b)                                          (c)

Figure 3: Average prediction MSE with standard deviation with different numbers of labeled samples. (a), (b) and (c) represent the results of the unlabeled training set, labeled training set as well as the testing set, respectively.

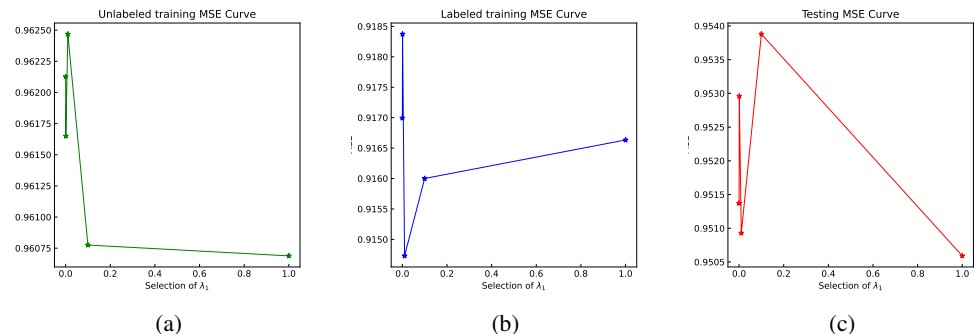

(a)                                          (b)                                          (c)

Figure 4: Average prediction MSE with different settings of $\lambda_1$. (a), (b) and (c) represent the results of the unlabeled training set, labeled training set as well as the testing set, respectively.

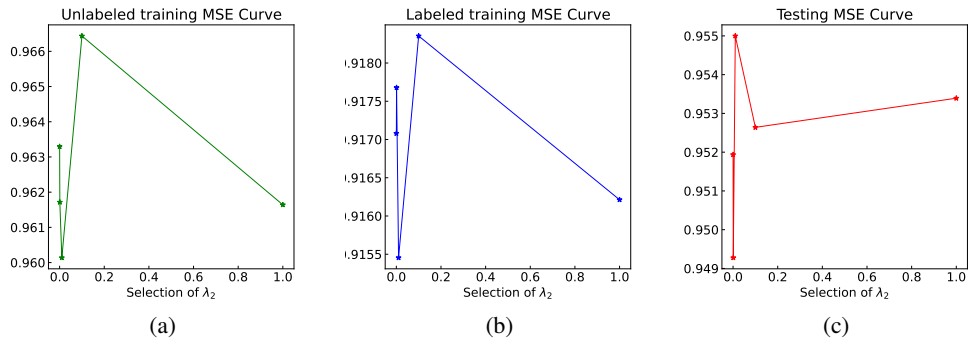

(a)                                          (b)                                          (c)

Figure 5: Average prediction MSE with different settings of $\lambda_2$. (a), (b) and (c) represent the results of the unlabeled training set, labeled training set as well as the testing set, respectively.

learning (Christmann & Zhou, 2016; Chen et al., 2020) and semi-supervised learning (Liu & Chen, 2018).

### B.4.2 IMPACT OF REGULARIZATION COEFFICIENTS AND GAUSSIAN KERNEL BANDWIDTH

Here we focus on the impact of regularization coefficients $\lambda_1, \lambda_2$ as well as the Gaussian kernel bandwidth on the prediction performance.

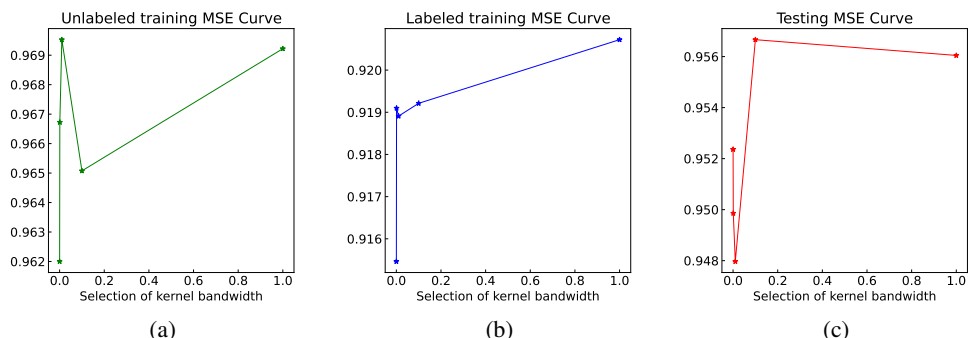

Figure 6: Average prediction MSE with different settings of Gaussian kernel bandwidth for computing similarity matrix. (a), (b) and (c) represent the results of the unlabeled training set, labeled training set as well as the testing set, respectively.

Initially, we set $\lambda_1 = \lambda_2 = 10^{-3}$ as default. By changing merely a single parameter and fixing the left one, we draw the sensitive curves in Figures 4, 5, and 6. From practical experiments, we find that too large $\lambda_1$ may introduce too much sparsity, where truly informative variables could also be assigned with quite small weights. And $\lambda_2$ directly determines the bias degree of the model towards unlabeled samples. And the kernel bandwidth controls the similarity matrix, where too small or too large ones can hinder the presentation of similarity between labeled and unlabeled samples. Properly selected parameters guide the model to better investigate information from unlabeled data.

### B.4.3 IMPACT OF SELECTED CORE SIZE $C$

Now we start to analyze the sensitivity of core size C on the performance. Following similar settings as in the last subsection, the sensitive curves on varying C with the Friedman regression data and synthetic additive regression data are plotted in Figure 7. The labeled rate is 5% in the training set. The average MSE and standard deviation after 20 repeated experiments are reported.

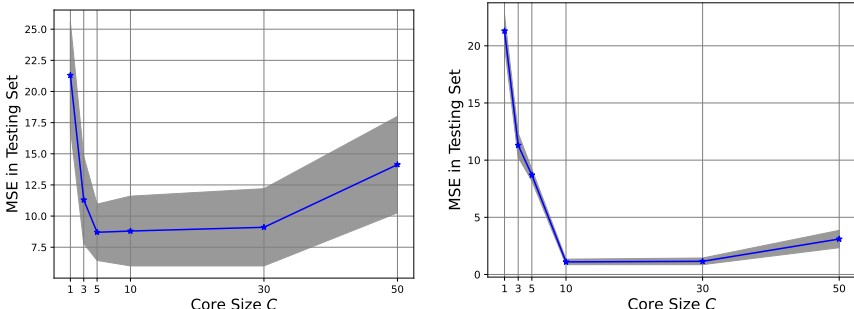

Figure 7: Average prediction MSE with different settings of parameter $C$. The left and right panels present the results on Friedman data (with 5/95/10 informative/redundant/noisy features) and synthetic additive regression data (with 8/92/10 informative/redundant/noisy features), respectively.

The empirical results show that, the size of core variables $C$ is also a crucial parameter of S$^2$MAM to assign proper masks on informative variables. In some high-dimensional real-world data without prior knowledge of truly useful variables, the binary (half-interval) searching method is suggested for setting $C$. Moreover, developing another level of problem to automatically search the proper $C$ is also an interesting and meaningful direction, while the computation cost might be also increasing. Empirically, the coreset size $C$ for useful variables could be set slightly larger than ground truth due to the sparsity constraint with $\ell$-1 regularization. Besides, too large $C$ may introduce more useless variables or even noisy variables, which could degrade the prediction performance.

When it comes to determining the value of $C$ within the confines of the constraint set $\mathcal{C}_s$, which is defined by:

$$\mathcal{C}_s = \left\{ s : 0 \preceq s_i \preceq 1, \|s\|_1 \leq C, i = 1, 2, \cdots, p \right\},$$

we take the overall dimension $d$ as the starting point, setting $C$ equal to $d$. To streamline the process, in the initial stage, we identify the most suitable value for $C$, denoted as $\hat{C}$, by examining a sequence that starts at $d$ and decreases by factors of two down to 1, i.e., $[d, d/2, d/4, \ldots, 2, 1]$. Fortunately, our practical tests have shown that $S^2$MAM is capable of pinpointing the correct dimensions with high accuracy right from the outset, thereby significantly easing the burden of manually identifying key features.

### B.4.4 CONVERGENCE OF UPPER LEVEL PROBLEM

Then we analyze the convergence performance of the mask learner in the upper level by drawing the curve of the upper-level objective function value with respect to the iteration $t$ in Figure 8.

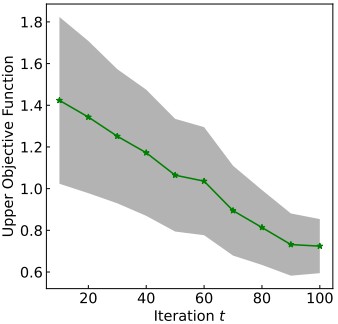

Figure 8: Convergence curve of the upper level problem of $S^2$MAM.

The synthetic additive regression data with noisy feature corruptions is used here. With less than 100 iterations, our method almost realizes convergence. However, compared to some existing SSL methods, the proposed $S^2$MAM may introduce more computation and space complexity due to the additional computation on the masks.

### B.5 ADDITIONAL SEMI-SUPERVISED REGRESSION & CLASSIFICATION ON UCI DATASET

Here we further present the additional empirical results of some baselines and $S^2$MAM on SSL learning problems. Following similar strategies for hyperparameter selection, we conduct more experiments on these eight UCI datasets by assigning a few data with true labels as well as some samples without labels, and regarding the remaining points as testing sets. To better highlight the robustness of $S^2$MAM against noisy variables, the original input $X$ is corrupted by 10 noisy variables following $\mathcal{N}(100, 100)$.

Tables 8 and 9 illustrate the experimental results on UCI data sets by changing the number of labeled training samples $l$, unlabeled training samples $u$, and noisy variables $p_n$. Due to the fact that the data sizes of different classes could be different, we fixed the size of training samples and merely changed the labeled data size. The remaining samples are the unlabeled data sets. Because some datasets are extremely large, we repeat each method 100 times on each dataset, and list the average results as well as the standard deviation information.

Besides, one can see that these algorithms almost perform better with the increasing number of labeled samples. Instead of the MSE and accuracy results, we further consider the R-squared score as the criterion to measure the performance of these methods on complex real-world data (involving a few labeled samples and unknown noises). Moreover, our proposed $S^2$MAM enjoys competitive or even better performance than these supervised or semi-supervised baselines, especially when the data is additionally corrupted by noisy variables.

Table 8: Average R-squared score $\pm$ standard deviation on UCI data. The four tables from top to bottom represent the regression results under settings of $\{l = 50/20/10/50, u = 450/180/40/450, p_n = 0\}$, $\{l = 50/20/10/50, u = 450/180/40/450, p_n = 10\}$, $\{l = 100/40/20/100, u = 400/160/30/400, p_n = 0\}$ and $\{l = 100/40/20/100, u = 400/160/30/400, p_n = 10\}$, respectively.

| Model | Buzz-Regression | | Boston House | | Ozone | | SkillCraft | |
|---|---|---|---|---|---|---|---|---|
| | Unlabeled | Test | Unlabeled | Test | Unlabeled | Test | Unlabeled | Test |
| Lasso | - | -0.146 ± 12.345 | - | 0.045 ± 3.135 | - | 0.324 ± 0.822 | - | 0.467 ± 0.220 |
| SpAM | - | 0.559 ± 1.969 | - | 0.322 ± 3.693 | - | 0.340 ± 0.278 | - | 0.504 ± 0.173 |
| LapRLS | 0.631 ± 0.236 | 0.632 ± 0.240 | 0.513 ± 0.196 | 0.482 ± 0.219 | 0.557 ± 0.178 | 0.550 ± 0.192 | 0.509 ± 0.125 | 0.506 ± 0.141 |
| VAE | 0.659 ± 2.406 | 0.641 ± 2.711 | 0.525 ± 1.213 | 0.519 ± 1.301 | 0.562 ± 1.043 | 0.557 ± 1.260 | 0.512 ± 0.460 | 0.504 ± 0.475 |
| COREG | 0.691 ± 1.733 | 0.684 ± 1.851 | **0.565 ± 0.981** | 0.557 ± 1.020 | **0.573 ± 0.958** | **0.566 ± 1.030** | 0.540 ± 0.376 | 0.532 ± 0.386 |
| SSDKL | **0.717 ± 2.307** | **0.709 ± 2.434** | 0.534 ± 2.107 | 0.527 ± 2.195 | 0.569 ± 1.424 | 0.562 ± 1.472 | 0.524 ± 0.560 | 0.512 ± 0.581 |
| S²MAM (ours) | 0.712 ± 1.055 | 0.704 ± 1.240 | 0.563 ± 0.737 | **0.559 ± 0.802** | 0.568 ± 0.194 | 0.563 ± 0.207 | **0.542 ± 0.217** | **0.535 ± 0.240** |
| Lasso | - | -3.364 ± 137.251 | - | -0.358 ± 3.329 | - | -0.719 ± 4.627 | - | 0.322 ± 0.564 |
| SpAM | - | 0.364 ± 2.596 | - | -0.023 ± 0.370 | - | -0.028 ± 0.078 | - | 0.375 ± 0.438 |
| LapRLS | 0.581 ± 0.244 | 0.574 ± 0.251 | 0.473 ± 0.223 | 0.461 ± 0.247 | 0.362 ± 0.347 | 0.357 ± 0.378 | 0.485 ± 0.138 | 0.477 ± 0.146 |
| VAE | 0.573 ± 3.107 | 0.566 ± 3.211 | 0.492 ± 4.683 | 0.487 ± 4.820 | 0.485 ± 2.177 | 0.463 ± 2.305 | 0.503 ± 0.870 | 0.494 ± 0.891 |
| COREG | 0.595 ± 2.422 | 0.581 ± 2.507 | 0.511 ± 3.328 | 0.509 ± 3.511 | 0.492 ± 1.560 | 0.481 ± 1.633 | 0.517 ± 0.644 | 0.512 ± 0.671 |
| SSDKL | 0.517 ± 3.924 | 0.504 ± 3.955 | 0.502 ± 3.730 | 0.501 ± 3.795 | 0.483 ± 1.866 | 0.475 ± 1.947 | 0.511 ± 1.104 | 0.506 ± 1.193 |
| S²MAM (ours) | **0.687 ± 1.401** | **0.673 ± 1.534** | **0.549 ± 0.947** | **0.541 ± 0.982** | **0.529 ± 0.471** | **0.517 ± 0.492** | **0.523 ± 0.424** | **0.520 ± 0.439** |
| Lasso | - | 0.817 ± 0.115 | - | 0.552 ± 0.309 | - | 0.619 ± 0.331 | - | 0.524 ± 0.141 |
| SpAM | - | 0.804 ± 0.177 | - | 0.554 ± 0.335 | - | 0.631 ± 0.314 | - | 0.529 ± 0.102 |
| LapRLS | 0.841 ± 0.149 | 0.822 ± 0.205 | 0.612 ± 0.161 | 0.607 ± 0.170 | 0.650 ± 1.273 | 0.642 ± 1.311 | 0.536 ± 0.102 | 0.531 ± 0.125 |
| VAE | 0.817 ± 0.346 | 0.812 ± 0.355 | 0.631 ± 0.971 | 0.627 ± 0.990 | 0.664 ± 0.913 | 0.657 ± 0.930 | 0.542 ± 0.310 | 0.538 ± 0.318 |
| COREG | 0.881 ± 0.311 | 0.869 ± 0.320 | 0.646 ± 0.730 | **0.642 ± 0.762** | 0.673 ± 0.731 | 0.662 ± 0.760 | 0.548 ± 0.261 | 0.541 ± 0.275 |
| SSDKL | **0.911 ± 0.395** | **0.905 ± 0.418** | 0.634 ± 1.625 | 0.627 ± 1.692 | **0.679 ± 1.105** | 0.670 ± 1.231 | **0.569 ± 0.462** | **0.560 ± 0.471** |
| S²MAM (ours) | 0.901 ± 0.211 | 0.891 ± 0.180 | **0.650 ± 0.510** | 0.641 ± 0.522 | 0.677 ± 0.143 | **0.672 ± 0.159** | 0.563 ± 0.135 | 0.558 ± 0.146 |
| Lasso | - | 0.773 ± 0.433 | - | 0.526 ± 0.571 | - | -1.025 ± 3.630 | - | 0.515 ± 0.149 |
| SpAM | - | 0.747 ± 0.542 | - | 0.530 ± 0.672 | - | 0.324 ± 3.395 | - | 0.522 ± 0.191 |
| LapRLS | 0.711 ± 0.377 | 0.702 ± 0.392 | 0.522 ± 0.193 | 0.510 ± 0.217 | 0.574 ± 0.278 | 0.563 ± 0.304 | 0.504 ± 0.127 | 0.498 ± 0.132 |
| VAE | 0.742 ± 2.871 | 0.736 ± 2.951 | 0.546 ± 3.720 | 0.541 ± 2.807 | 0.591 ± 2.041 | 0.584 ± 2.259 | 0.529 ± 0.511 | 0.522 ± 0.519 |
| COREG | 0.771 ± 2.142 | 0.761 ± 2.216 | 0.565 ± 1.836 | 0.561 ± 1.862 | 0.595 ± 1.320 | 0.589 ± 1.452 | 0.538 ± 0.431 | 0.530 ± 0.438 |
| SSDKL | 0.764 ± 3.104 | 0.749 ± 3.277 | 0.537 ± 2.541 | 0.522 ± 2.679 | 0.602 ± 1.655 | 0.590 ± 1.712 | 0.546 ± 0.831 | 0.541 ± 0.840 |
| S²MAM (ours) | **0.812 ± 1.255** | **0.804 ± 1.278** | **0.621 ± 0.866** | **0.610 ± 0.879** | **0.644 ± 0.386** | **0.631 ± 0.397** | **0.558 ± 0.265** | **0.551 ± 0.271** |

Table 9: The average Accuracy $\pm$ standard deviation (%) on UCI data. The upper and lower tables represent the results under $\{l = 50/50/50/20, u = 450/250/250/130, p_n = 0\}$ and $\{l = 50/100/100/50, u = 450/200/200/100, p_n = 10\}$, respectively.

| Model | Buzz-classification | | Breast Cancer | | Phishing Websites | | Statlog Heart | |
|---|---|---|---|---|---|---|---|---|
| | Unlabeled | Test | Unlabeled | Test | Unlabeled | Test | Unlabeled | Test |
| $\ell$-1 SVM | - | 90.792 ± 4.287 | - | 91.957 ± 2.966 | - | 73.874 ± 4.527 | - | 82.127 ± 7.906 |
| SpAM | - | 91.021 ± 3.022 | - | 92.358 ± 2.962 | - | 76.637 ± 4.204 | - | 82.143 ± 8.439 |
| LapSVM | 92.171 ± 2.957 | 92.019 ± 3.031 | 93.229 ± 2.415 | 93.102 ± 2.493 | 82.268 ± 3.481 | 82.144 ± 3.546 | 84.736 ± 4.622 | 84.622 ± 4.640 |
| f-FME | 96.387 ± 2.254 | 96.149 ± 2.293 | 94.903 ± 2.281 | 94.622 ± 2.341 | 87.530 ± 4.503 | 87.492 ± 4.670 | 85.903 ± 3.379 | 85.811 ± 3.401 |
| AWSSL | 96.507 ± 3.513 | 96.540 ± 3.562 | 94.942 ± 1.955 | 94.903 ± 1.986 | 85.297 ± 2.248 | 85.166 ± 2.317 | **86.120 ± 3.213** | **86.089 ± 3.266** |
| S²MAM (ours) | **96.784 ± 2.908** | **96.713 ± 2.930** | **95.007 ± 1.748** | **94.916 ± 1.803** | **88.343 ± 3.840** | **88.286 ± 3.867** | 86.095 ± 4.376 | 86.011 ± 4.409 |
| $\ell$-1 SVM | - | 72.882 ± 9.734 | - | 74.994 ± 8.531 | - | 55.918 ± 5.575 | - | 67.251 ± 9.143 |
| SpAM | - | 75.068 ± 7.455 | - | 79.943 ± 6.824 | - | 57.701 ± 5.311 | - | 69.989 ± 9.744 |
| LapSVM | 70.864 ± 12.250 | 70.214 ± 12.738 | 61.553 ± 9.502 | 61.114 ± 9.810 | 51.700 ± 5.306 | 51.342 ± 5.395 | 58.025 ± 5.427 | 57.984 ± 5.470 |
| f-FME | 82.759 ± 5.692 | 82.302 ± 5.741 | 75.261 ± 6.740 | 75.204 ± 6.862 | 76.623 ± 3.695 | 76.594 ± 3.710 | 74.998 ± 4.217 | 74.903 ± 4.236 |
| AWSSL | 89.672 ± 5.310 | 89.155 ± 5.412 | 77.197 ± 6.025 | 77.120 ± 6.136 | 78.025 ± 4.257 | 77.989 ± 4.303 | 76.622 ± 4.773 | 76.595 ± 4.914 |
| S²MAM (ours) | **92.618 ± 4.377** | **92.431 ± 4.526** | **88.053 ± 4.935** | **87.995 ± 4.947** | **81.992 ± 2.514** | **81.894 ± 2.527** | **79.498 ± 4.119** | **79.277 ± 4.171** |

As shown in Tables 2 and 9, S²MAM realizes the competitive or even best performance under most settings, especially with corrupted features. However, when the synthetic data is clean (without noisy variables), some deep SSL methods (COREG and SSDKL) may perform better than S²MAM.

This is understandable, as the proposed S²MAM is built on kernels and deep neural networks usually have stronger fitting ability under clean data (Ghorbani et al., 2020; Agarwal et al., 2021; Yang et al., 2020). These deep SSL methods and the well-trained S²MAM use all the informative input variables. While still enjoying competitive prediction accuracy w.r.t. Deep SSL methods, S2MAM further provides explainable predictions; please refer to Fig.7 with visual examples on Page 23, where there may exist a tradeoff between interpretability and accuracy (Rudin, 2019).

We further consider more settings of noisy variables, e.g., $\mathcal{N}(0, 100)$, $\mathcal{N}(50, 100)$, Student T distribution (with freedom of 2/5/10) and Chi-square noise (with freedom of 2/5/10), where the results are analogous to the setting ($X_n \in \mathcal{N}(100, 100)$). Thus the extremely large random noise following $\mathcal{N}(100, 100)$ is employed throughout the whole paper for simplicity and consistency.

In order to make a comprehensive comparison, we further consider the data settings of 5%/50% labeled samples and $p_n = 0/100$ noisy features on the synthetic additive data. The results are summarized in Table 10. The empirical results show that:

Table 10: Average Accuracy $\pm$ standard deviation (%) on synthetic additive data with label percentages in each class ($r = 5\%/50\%$) and noisy variable numbers ($p_n = 0/100$).

| Model | r = 5%, $p_n = 0$ | | r = 5%, $p_n = 100$ | | r = 50%, $p_n = 0$ | | r = 50%, $p_n = 100$ | |
|---|---|---|---|---|---|---|---|---|
| | Unlabeled | Test | Unlabeled | Test | Unlabeled | Test | Unlabeled | Test |
| $\ell_1$-SVM | - | $83.914 \pm 6.410$ | - | $53.471 \pm 8.427$ | - | $93.644 \pm 5.171$ | - | $88.474 \pm 6.209$ |
| SpAM | - | $84.150 \pm 6.104$ | - | $51.308 \pm 7.242$ | - | $94.020 \pm 4.255$ | - | $90.201 \pm 5.330$ |
| CSAM | - | $86.597 \pm 5.424$ | - | $56.410 \pm 8.781$ | - | $94.973 \pm 4.955$ | - | $91.210 \pm 5.237$ |
| TSpAM | - | $86.993 \pm 5.340$ | - | $56.811 \pm 7.570$ | - | $95.031 \pm 4.601$ | - | $\mathbf{91.244 \pm 5.197}$ |
| LapSVM | $88.814 \pm 5.398$ | $88.850 \pm 5.269$ | $37.174 \pm 10.244$ | $38.208 \pm 10.959$ | $93.899 \pm 4.860$ | $94.101 \pm 4.571$ | $41.177 \pm 9.814$ | $41.490 \pm 9.202$ |
| f-FME | $89.141 \pm 3.172$ | $89.305 \pm 3.359$ | $60.276 \pm 8.427$ | $59.771 \pm 8.610$ | $94.505 \pm 2.871$ | $94.893 \pm 2.747$ | $71.038 \pm 7.979$ | $70.875 \pm 8.201$ |
| AWSSL | $\mathbf{91.259 \pm 2.871}$ | $90.211 \pm 3.077$ | $62.707 \pm 8.660$ | $62.842 \pm 8.290$ | $95.410 \pm 3.229$ | $95.601 \pm 3.073$ | $69.071 \pm 7.759$ | $69.368 \pm 7.831$ |
| RGL | $90.422 \pm 2.909$ | $90.026 \pm 3.477$ | $64.371 \pm 8.391$ | $65.011 \pm 8.140$ | $\mathbf{95.973 \pm 2.417}$ | $96.027 \pm 2.289$ | $71.462 \pm 7.141$ | $71.511 \pm 7.062$ |
| SALE | $89.717 \pm 2.811$ | $90.149 \pm 2.665$ | $65.805 \pm 8.106$ | $65.887 \pm 8.010$ | $95.402 \pm 2.311$ | $95.427 \pm 2.268$ | $71.855 \pm 6.947$ | $71.913 \pm 6.850$ |
| $S^2$MAM (ours) | $89.979 \pm 3.255$ | $\mathbf{90.309 \pm 3.409}$ | $\mathbf{73.420 \pm 6.177}$ | $\mathbf{73.641 \pm 6.020}$ | $95.941 \pm 2.031$ | $\mathbf{96.147 \pm 1.954}$ | $\mathbf{76.518 \pm 5.326}$ | $76.560 \pm 5.244$ |

- At a 5% labeling rate, $S^2$MAM is capable of assigning suitable masks, effectively utilizing the input from 95% unlabeled data to boost the model's predictive accuracy.

- At a 50% labeling rate, these supervised baselines usually maintain better sparse regression estimators than $S^2$MAM. The empirical observations are natural since the labeled data under this setting is often enough to find the predictor, and supervised methods should be suggested.

### B.6 INTERPRETABILITY AND VISUALIZATION

Notably, additive models, including our proposed $S^2$MAM, own strong interpretability, where the component function of each input variable can be explicitly formulated and directly visualized. Here we also give an example with our synthetic additive regression data, where the ground truth function is merely relevant to the first eight input variables:

$$Y = f^*(X) + \epsilon = \sum_{j=1}^{8} f^{(j)*}(X^{(j)}) + \epsilon, \tag{11}$$

where $f^{(1)*}(u) = -2\sin(2u)$, $f^{(2)*}(u) = 8u^2$, $f^{(3)*}(u) = \frac{7\sin u}{2 - \sin u}$, $f^{(4)*}(u) = 6e^{-u}$, $f^{(5)*}(u) = u^3 + \frac{3}{2}(u-1)^2$, $f^{(6)*}(u) = 5u$, $f^{(7)*}(u) = 10\sin(e^{-u/2})$, $f^{(8)*}(u) = -10\widetilde{\phi}(u, \frac{1}{2}, \frac{4}{5}^2)$.

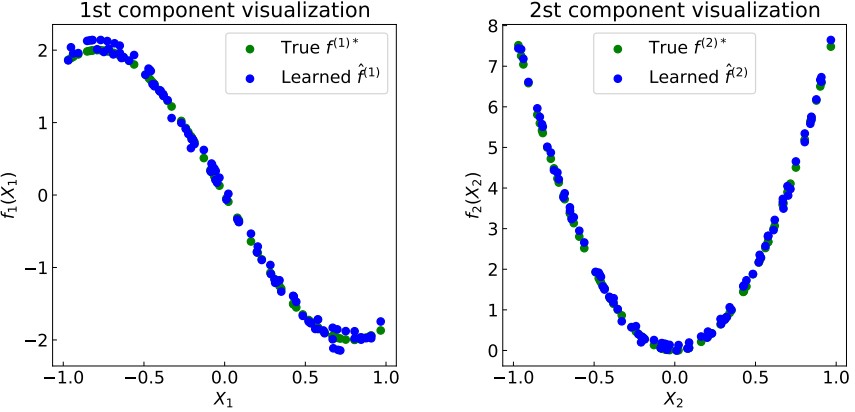

Figure 9: Visualization of the first two components. $f^*$ : ground truth; $\hat{f}$ : results predicted by $S^2$MAM.

For simplicity, we present the prediction components of $\hat{f}^{(1)}$ and $\hat{f}^{(2)}$ as well as their ground truth $f^{(1)*}$ and $f^{(2)*}$ in Figure 9. We generate the input uniformly among $[-1, 1]$, which is further transformed into the Gram matrix of the corresponding component ($\mathbf{K}^{(1)}$ and $\mathbf{K}^{(2)}$). By multiplying with the model coefficients $\alpha^{(1)}$ and $\alpha^{(2)}$, one can directly obtain the outputs. As shown in Figure 9, the prediction results of $S^2$MAM for each input variable are close to the ground truth, which better

validates the effectiveness. And the other components can also be formulated or visualized, we omit it here.

**Remark 7.** *In some relevant works, the high-dimensional observations can be regarded as the mixture of hidden information from an unknown manifold and ambient noise (Yao et al., 2024). In many realistic settings, including redundant useless, or noisy variables, the real-world data can be also corrupted by some noisy labels. In order to achieve robustness against such corruptions, a commonly considered approach is to replace the loss function with a robust one (e.g., the widely used robust Huber loss function (Wang et al., 2022b) for regression tasks). Simple modifications may help to improve the models' robustness against noisy labels. Extensions of $S^2MAM$ from other perspectives are interesting directions in the future study.*

### B.7    EXTENSION TO IMAGE DATA

Inspired by some supervised (Su et al., 2023) and semi-supervised works (Qiu et al., 2018; Nie et al., 2019; Kang et al., 2020; Nie et al., 2021), an interesting approach for dealing with high-dimensional data like images is to extract the variable vectors first.

Following (Bao et al., 2024), we first use a CNN to learn the vectors with 32 features for each image, which realizes rough dimensional reduction. However, this step may not remove those irrelevant or even noisy variables. Thus, it's still necessary to employ robust methods before building semi-supervised models. Similar preprocessing methods for dimensional reduction also apply to larger (image) datasets. The extended experimental results on classifying the 12-th and 13-th objects in COIL20 image data (download from https://www.cs.columbia.edu/CAVE/software/softlib/coil-20.php) after dimensional reduction are present as follows.

Firstly, we directly conduct experiments on the clean process COIL20 feature matrix. The results are present in Table 11. Secondly, following the settings in (Bao et al., 2024), to simulate pixel-level corruption in images, we manually add 5 noisy variables following $\mathcal{N}(100, 100)$ to the processed 32 dimensions, where the results are left in Table 12.

For the following classification task, the supervised competitors include linear $\ell_1$-SVM (Zhu et al., 2003a), SpAM (Ravikumar et al., 2009), CSAM (Yuan et al., 2023) and TSpAM (Wang et al., 2023). And the semi-supervised baselines include LapSVM (Belkin et al., 2006), f-FME (Qiu et al., 2018), AWSSL (Nie et al., 2019), RGL (Kang et al., 2020) and SALE (Nie et al., 2021).

Table 11: Extended experiments with average accuracy, standard deviation (SD), and training time cost (minutes) on image data. Merely 30% samples are labeled. Both $\ell_1$-SVM and LapSVM adopt the gradient optimization.

|            | $\ell_1$-SVM | SpAM   | CSAM   | TSpAM  | LapSVM | f-FME  |
|------------|--------------|--------|--------|--------|--------|--------|
| Accuracy   | 67.329       | 69.917 | 73.577 | 72.230 | 81.092 | 85.518 |
| SD         | 0.583        | 0.709  | 0.622  | 0.616  | 0.417  | 0.408  |
| Time Cost  | 0.2          | 0.9    | 2.3    | 2.5    | 0.6    | 1.5    |

|            | AWSSL  | RGL    | SALE   | SSNP   | $S^2MAM$ |  |
|------------|--------|--------|--------|--------|----------|--|
| Accuracy   | 86.821 | 83.416 | 87.235 | 83.370 | 86.833   |  |
| SD         | 0.430  | 0.527  | 0.616  | 0.429  | 0.501    |  |
| Time Cost  | 2.7    | 3.1    | 2.2    | 4.1    | 9.6      |  |

Table 12: Extended experiments with average accuracy $\pm$ standard deviation on (the 12-th and 13-th objects of) the corrupted COIL20 image data, which involves 5 noisy variables. For simplicity, the competitors used here are all designed for SSL.

| LapSVM | f-FME | AWSSL | RGL | SALE | SSNP | $S^2MAM$ |
|--------|-------|-------|-----|------|------|----------|
| $57.026 \pm 7.192$ | $76.464 \pm 4.106$ | $74.034 \pm 3.226$ | $74.217 \pm 3.011$ | $75.109 \pm 4.049$ | $77.629 \pm 4.310$ | $78.917 \pm 3.601$ |

From the above results in Tables 11 and 12, our proposed $S^2MAM$ provides competitive and robust prediction performance under clean or corrupted data. However, $S^2MAM$ brings more computation cost. This is mainly caused by:

1) The bilevel optimization requires more iterations to learn the additional masks;

2) The additive scheme expands the data dimensions to provide interpretable feature-wise contributions.

In order to reduce the computation burden of bilevel optimization, this paper adopts the optimization from (Zhou et al., 2022) with the probabilistic formulation and policy gradient estimation.

To further accelerate the computation process, the random Fourier acceleration technique (Rahimi & Recht, 2007) can be exploited to approximate the additive kernel (Gram) matrix, which has been previously validated to be effective for additive models (Wang et al., 2023).

### B.8  EXPLANATION FOR TOY EXAMPLE IN FIGURE 1

To better illustrate the negative impact of noisy variables on SSL models, we conduct semi-supervised binary classification experiments on moon data (Nie et al., 2019). For simplicity, here we generate totally 200 samples involving 99 unlabeled points and 1 labeled point for each class. The original moon data involves two inputs ($X$ and $y$) and a single label ($-1$ or $1$). In order to highlight the robustness, we further add a noisy input variable ($X_n \in \mathcal{N}(100, 100)$). Thus the corrupted sample involves three inputs and a single output, where the $i$-th sample includes input variables $x_i = (X_i, y_i, (X_n)_i)$ and true label -1 or 1.

As shown in Figure 1, both LapSVM and our proposal S$^2$MAM perform well on the clean moon data without corruptions in Figure 1 (a). In the 2D plot in Figure 1 (b) and 3D plot in Figure 1 (d), the noisy variable directly causes negative impact on the Laplacian matrix $\mathbf{W}$, whose calculation relies on all input variables $W_{ij} = \exp\{-\|x_i - x_j\|/2\mu^2\}$ with bandwidth $\mu$.

And as present in Figure 1 (d), our proposed S$^2$MAM, with learned mask $\boldsymbol{m} = (1, 1, 0)$ assigned on inputs $(X, y, X_n)$, is robust with masked similarity $W_{ij} = \exp\{-\|\boldsymbol{m} \odot x_i - \boldsymbol{m} \odot x_j\|/\mu^2\}$, since noisy variable $X_n$ is suppressed with mask 0.

### B.9  VISUALIZED LEARNING PROCESS OF S$^2$MAM

Here we further present the visualization for the learning process of S$^2$MAM, which shows the importance of assigning proper masks for (high-dimensional) semi-supervised modeling.

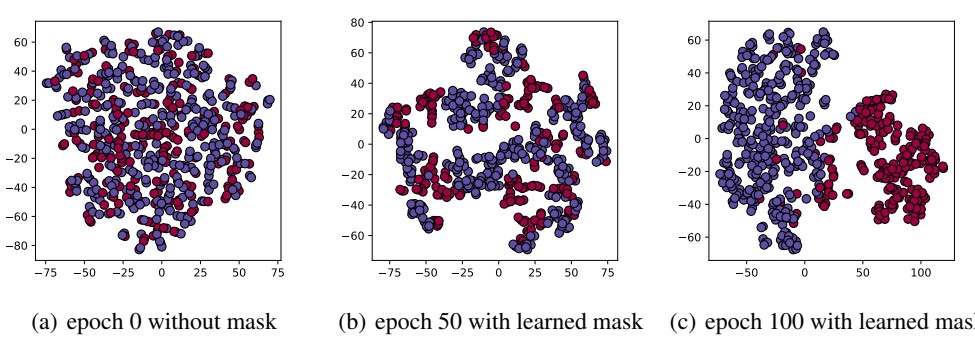

(a) epoch 0 without mask      (b) epoch 50 with learned mask      (c) epoch 100 with learned mask

Figure 10: 2d tSNE visualization for masked Breast Cancer data corrupted by 10 noisy features during the training process of S$^2$MAM at epoch 0, 50 and 100, respectively. Dots with different colors represent different classes.

In Figure 10, we present the visualization of masked Breast Cancer data based on the tSNE technique (Van der Maaten & Hinton, 2008), where the masks are updated gradually and almost could reach the ground truth after 100 epochs.

## C  GENERALIZATION ERROR ANALYSIS (PROOF OF THEOREM 2)

To better illustrate the proof process, we summarize the major steps and lemmas in the following Figure 11.

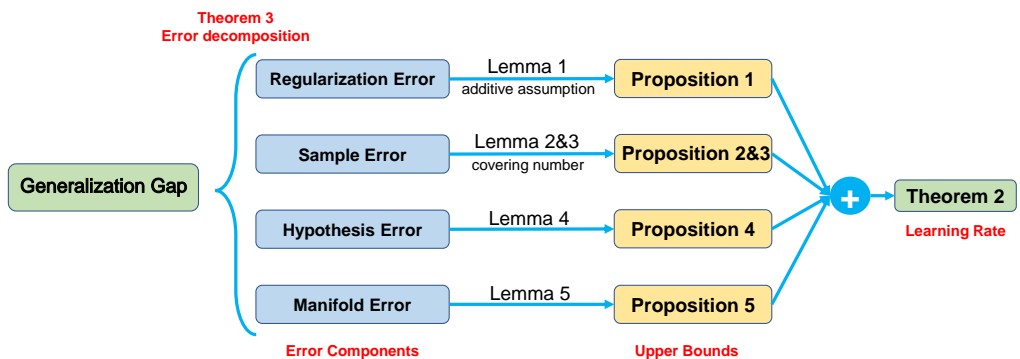

Figure 11: Sketch of the theoretical proofs for generalization bound.

## C.1 ERROR DECOMPOSITION

Now we are in the position to recall the semi-supervised algorithm with $\ell_2$ regularizer in the additive hypothesis space

$$f_{\mathbf{z}} = \arg\min_{f \in \mathcal{H}} \left\{ \mathcal{E}_{\mathbf{z}}(f) + \lambda_1 \Omega_{\mathbf{z}}(f) + \frac{\lambda_2}{(l+u)^2} \mathbf{f}^T L \mathbf{f} \right\}. \tag{12}$$

For simplicity, the semi-supervised regression task with squared loss under a kernel-based framework is considered here. Denote $\mathbf{z} = \{\mathbf{z}_l, \mathbf{z}_u\}$ as the labeled data $\mathbf{z}_l = \{x_i, y_i\}_{i=1}^l$ and unlabeled data $\mathbf{z}_u = \{x_i\}_{i=l+1}^{l+u}$ together. Denote $\mathbf{f} = (f(x_1), \dots, f(x_{l+u}))^T$, which involves the prediction of both the labeled and unlabeled data. $\lambda_1 > 0$ and $\lambda_2 > 0$ are regularization parameters. Series $\{\tau_j\}_{j=1}^p$ are weights to different input variables. For feasibility, define the Gram matrix $\mathbf{K}_i = \left( \mathbf{K}_i^{(1)}, \dots, \mathbf{K}_i^{(p)} \right)^T \in \mathbb{R}^{(l+u) \times p}$, $\mathbf{K}^{(j)} = \left( \mathbf{K}_1^{(j)}, \dots, \mathbf{K}_{l+u}^{(j)} \right)^T \in \mathbb{R}^{(l+u) \times (l+u)}$ with $\mathbf{K}_i^{(j)} = \left( K^{(j)}(x_1^{(j)}, x_i^{(j)}), \dots, K^{(j)}(x_{l+u}^{(j)}, x_i^{(j)}) \right)^T \in \mathbb{R}^{l+u}$ and the coefficient $\boldsymbol{\alpha} = \left( \boldsymbol{\alpha}^{(1)}, \dots, \boldsymbol{\alpha}^{(p)} \right)^T \in \mathbb{R}^{(l+u) \times p}$ with $\boldsymbol{\alpha}^{(j)} = \left( \alpha_1^{(j)}, \dots, \alpha_{l+u}^{(j)} \right)^T \in \mathbb{R}^{l+u}$.

The manifold regularized additive model in Eq.(12) can be formulated as

$$f_{\mathbf{z}} = \arg\min_{f = \sum_{j=1}^p f^{(j)} \in \mathcal{H}} \left\{ \mathcal{E}_{\mathbf{z}}(f) + \lambda_1 \Omega_{\mathbf{z}}(f) + \frac{\lambda_2}{(l+u)^2} \mathbf{f}^T L \mathbf{f} \right\}, \tag{13}$$

where

$$\mathcal{E}_{\mathbf{z}}(f) = \frac{1}{l} \sum_{i=1}^l (f(x_i) - y_i)^2 = \frac{1}{l} \sum_{i=1}^l \left( \sum_{j=1}^p (\mathbf{K}_i^{(j)})^T \alpha^{(j)} - y_i \right)^2. \tag{14}$$

If the $j$-th variable is not truly informative, we expect that $\hat{\alpha}_{\mathbf{z}}^{(j)} = \left( \hat{\alpha}_{\mathbf{z},1}^{(j)}, \dots, \hat{\alpha}_{\mathbf{z},l+u}^{(j)} \right)^T \in \mathbb{R}^{l+u}$ satisfies $\left\| \hat{\alpha}_{\mathbf{z}}^{(j)} \right\|_2 = \left( \sum_{i=1}^{l+u} \left| \hat{\alpha}_{\mathbf{z},i}^{(j)} \right|^2 \right)^{(1/2)} = 0$. Inspired by this, we introduce the $\ell_{2,1}$-regularizer

$$\Omega_{\mathbf{z}}(f) = \inf \left\{ \sum_{j=1}^p \tau_j \left\| \alpha^{(j)} \right\|_2 : f = \sum_{j=1}^p \sum_{i=1}^{l+u} \alpha^{(j)} K^{(j)} \left( x_i^{(j)}, \cdot \right), \alpha^{(j)} \in \mathbb{R}^{l+u} \right\} \tag{15}$$

as the penalty to address the sparsity of the output functions.

Suppose that $\rho$ is a fixed (but unknown) probability distribution on $Z := X \times Y$. Define $f^{(j)} = (\mathbf{K}^{(j)})^T \alpha^{(j)}$. Similarly, now we introduce a regularizing function as

$$f_\lambda = \arg\min_{f = \sum_{j=1}^p f^{(j)} \in \mathcal{H}} \left\{ \mathcal{E}(f) + \lambda_1 \Omega(f) + \lambda_2 \langle f, L_\omega f \rangle_2 \right\}, \tag{16}$$

where

$$\mathcal{E}(f) = \int_{\mathbf{z}} (f(x) - y)^2 d\rho, \tag{17}$$

and

$$\Omega(f) = \sum_{j=1}^{p} \tau_j \|f^{(j)}\|_{K^{(j)}}^2. \tag{18}$$

Before presenting the error analysis, we give some basic definitions throughout this paper.

**Definition 1.** *Define* $\kappa = \sup_{j,u} \left( K^{(j)}(u,u) \right)^{1/2} < \infty$. *For* $f_{\mathbf{z}}$ *defined above, there holds*

$$\|f_{\mathbf{z}}\|_K \leq \kappa \sum_{j=1}^{p} \sum_{i=1}^{l+u} \left| \alpha_{\mathbf{z},i}^{(j)} \right| \leq \kappa \sum_{j=1}^{p} \left( \sum_{i=1}^{l+u} 1^{1-\frac{1}{q}} \right)^{1-\frac{1}{q}} \left( \sum_{i=1}^{l+u} \left| \alpha_{\mathbf{z},i}^{(j)} \right|^q \right)^{\frac{1}{q}} \leq \kappa \sqrt{l+u} \sum_{j=1}^{p} \left\| \alpha_{\mathbf{z}}^{(j)} \right\|_2, \tag{19}$$

*where the last inequality is obtained from the Hölder inequality with positive constant* $q = 2$.

**Remark 8.** *Based on the definition of* $\kappa$ *and* $\Omega_{\mathbf{z}}(f)$, *we can further obtain* $\|f\|_\infty \leq \kappa \|f\|_K$ *for any* $f \in \mathcal{H}_K$ *(Mukherjee et al., 2006; Chen et al., 2018).*

**Definition 2.** *Define an operator* $L_\omega : L_{\rho_X}^2 \to L_{\rho_X}^2$ *by* $(L_\omega f)(x) = f(x)p(x) - \int_X K(x,x') f(x') d\rho_X(x')$, *with* $p(x) = \int_X K(x,x') d\rho_X(x')$. *Then we have*

$$\langle f, L_\omega f \rangle_2 = \frac{1}{2} \iint (f(x) - f(x'))^2 W(x,x') d\rho_X(x) d\rho_X(x').$$

**Definition 3.** *For any measurable function* $f : X \to \mathbb{R}$, *define the following clipping function:*

$$\pi(f) = \left\{ \begin{array}{ll} M & f(x) > M \\ -M & f(x) < -M \\ f(x) & otherwise \end{array} \right. . \tag{20}$$

**Theorem 3.** *Let* $f_{\mathbf{z}}$ *be defined by (12) and* $\pi(f)$ *defined in (20). Then for* $\lambda > 0$, *we have*

$$\mathcal{E}(\pi(f_{\mathbf{z}})) - \mathcal{E}(f_\rho) \leq \mathcal{D}(\lambda) + \mathcal{S}(\mathbf{s},\lambda) + \mathcal{H}(\mathbf{s},\lambda) + \mathcal{M}(\mathbf{s},\lambda), \tag{21}$$

*where the regularization error, sample error, hypothesis error, and manifold error can be defined respectively as*

$$\mathcal{D}(\lambda) = \mathcal{E}(f_\lambda) - \mathcal{E}(f_\rho) + \lambda_1 \sum_{j=1}^{p} \tau_j \left\| f_\lambda^{(j)} \right\|_{K^{(j)}}^2 + \lambda_2 \sum_{j=1}^{p} \left\langle f_\lambda^{(j)}, L_\omega f_\lambda^{(j)} \right\rangle_2,$$

$$\mathcal{S}(\mathbf{z},\lambda) = \mathcal{E}(\pi(f_{\mathbf{z}})) - \mathcal{E}_{\mathbf{z}}(\pi(f_{\mathbf{z}})) + \mathcal{E}_{\mathbf{z}}(f_\lambda) - \mathcal{E}(f_\lambda),$$

$$\mathcal{H}(\mathbf{z},\lambda) = \mathcal{E}_{\mathbf{z}}(\pi(f_{\mathbf{z}})) + \lambda_1 \Omega(f_{\mathbf{z}}) + \frac{\lambda_2}{(l+u)^2} \sum_{j=1}^{p} (\mathbf{f}_{\mathbf{z}}^{(j)})^T L_j \mathbf{f}_{\mathbf{z}}^{(j)}$$

$$- \left\{ \mathcal{E}_{\mathbf{z}}(f_\lambda) + \lambda_1 \sum_{j=1}^{p} \tau_j \|f_\lambda^{(j)}\|_{K^{(j)}}^2 + \frac{\lambda_2}{(l+u)^2} \sum_{j=1}^{p} (\mathbf{f}_\lambda^{(j)})^T L_j \mathbf{f}_\lambda^{(j)} \right\}, \tag{22}$$

$$\mathcal{M}(\mathbf{z},\lambda) = \frac{\lambda_2}{(l+u)^2} \sum_{j=1}^{p} (\mathbf{f}_{\mathbf{z}}^{(j)})^T L_j \mathbf{f}_{\mathbf{z}}^{(j)} - \lambda_2 \sum_{j=1}^{p} \left\langle f_\lambda^{(j)}, L_\omega f_\lambda^{(j)} \right\rangle_2.$$

*Proof.* Based on the definition of $f_{\mathbf{z}}$ and $\pi(f)$, we have

$$\mathcal{E}\left(\pi\left(f_{\mathbf{z}}\right)\right) - \mathcal{E}\left(f_\rho\right)$$

$$\leq \mathcal{E}\left(\pi\left(f_{\mathbf{z}}\right)\right) - \mathcal{E}\left(f_\rho\right) + \lambda_1\Omega(f_{\mathbf{z}}) + \frac{\lambda_2}{(l+u)^2}\sum_{j=1}^{p}(\mathbf{f}_{\mathbf{z}}^{(j)})^T L_j \mathbf{f}_{\mathbf{z}}^{(j)}$$

$$\leq \mathcal{E}\left(\pi\left(f_{\mathbf{z}}\right)\right) - \mathcal{E}_{\mathbf{z}}(\pi\left(f_{\mathbf{z}}\right)) + \mathcal{E}_{\mathbf{z}}(\pi\left(f_{\mathbf{z}}\right)) + \lambda_1\Omega((f_{\mathbf{z}})) + \frac{\lambda_2}{(l+u)^2}\sum_{j=1}^{p}(\mathbf{f}_{\mathbf{z}}^{(j)})^T L_j \mathbf{f}_{\mathbf{z}}^{(j)}$$

$$- \left\{\mathcal{E}_{\mathbf{z}}\left(f_\lambda\right) + \lambda_1\sum_{j=1}^{p}\tau_j\|f_\lambda^{(j)}\|_{K^{(j)}}^2 + \lambda_2\sum_{j=1}^{p}\left\langle f_\lambda^{(j)}, L_\omega f_\lambda^{(j)}\right\rangle_2\right\}$$

$$+ \left\{\mathcal{E}_{\mathbf{z}}\left(f_\lambda\right) + \lambda_1\sum_{j=1}^{p}\tau_j\|f_\lambda^{(j)}\|_{K^{(j)}}^2 + \lambda_2\sum_{j=1}^{p}\left\langle f_\lambda^{(j)}, L_\omega f_\lambda^{(j)}\right\rangle_2\right\}$$

$$- \mathcal{E}\left(f_\lambda\right) + \mathcal{E}\left(f_\lambda\right) - \mathcal{E}\left(f_\rho\right) + \frac{\lambda_2}{(l+u)^2}\sum_{j=1}^{p}(\mathbf{f}_\lambda^{(j)})^T L_j \mathbf{f}_\lambda^{(j)} - \frac{\lambda_2}{(l+u)^2}\sum_{j=1}^{p}(\mathbf{f}_\lambda^{(j)})^T L_j \mathbf{f}_\lambda^{(j)}$$

$$\leq \underbrace{\mathcal{E}\left(f_\lambda\right) - \mathcal{E}\left(f_\rho\right) + \lambda_1\sum_{j=1}^{p}\tau_j\left\|f_\lambda^{(j)}\right\|_{K^{(j)}}^2 + \lambda_2\sum_{j=1}^{p}\left\langle f_\lambda^{(j)}, L_\omega f_\lambda^{(j)}\right\rangle_2}_{\mathcal{D}(\lambda)}$$

$$+ \underbrace{\mathcal{E}\left(\pi\left(f_{\mathbf{z}}\right)\right) - \mathcal{E}_{\mathbf{z}}\left(\pi\left(f_{\mathbf{z}}\right)\right) + \mathcal{E}_{\mathbf{z}}\left(f_\lambda\right) - \mathcal{E}\left(f_\lambda\right)}_{\mathcal{S}(\mathbf{z},\lambda)}$$

$$+ \underbrace{\mathcal{E}_{\mathbf{z}}\left(\pi\left(f_{\mathbf{z}}\right)\right) + \lambda_1\Omega\left(f_{\mathbf{z}}\right) + \frac{\lambda_2}{(l+u)^2}\sum_{j=1}^{p}(\mathbf{f}_{\mathbf{z}}^{(j)})^T L_j \mathbf{f}_{\mathbf{z}}^{(j)} - \left\{\mathcal{E}_{\mathbf{z}}\left(f_\lambda\right) + \lambda_1\sum_{j=1}^{p}\tau_j\|f_\lambda^{(j)}\|_{K^{(j)}}^2 + \frac{\lambda_2}{(l+u)^2}\sum_{j=1}^{p}(\mathbf{f}_\lambda^{(j)})^T L_j \mathbf{f}_\lambda^{(j)}\right\}}_{\mathcal{H}(\mathbf{z},\lambda)}$$

$$+ \underbrace{\frac{\lambda_2}{(l+u)^2}\sum_{j=1}^{p}(\mathbf{f}_\lambda^{(j)})^T L_j \mathbf{f}_\lambda^{(j)} - \lambda_2\sum_{j=1}^{p}\left\langle f_\lambda^{(j)}, L_\omega f_\lambda^{(j)}\right\rangle_2}_{\mathcal{M}(\mathbf{z},\lambda)},$$

where $\mathcal{D}(\lambda)$, $\mathcal{S}(\mathbf{z},\lambda)$, $\mathcal{H}(\mathbf{z},\lambda)$ and $\mathcal{M}(\mathbf{z},\lambda)$ stand for the regularization error, sample error, hypothesis error, and manifold error, respectively. The proof is completed. $\qquad\square$

## C.2 BOUNDING REGULARIZATION ERROR $\mathcal{D}(\lambda)$

In this section, we give the theoretical results under specific assumptions on $f_\rho$ for bounding the regularization error of manifold regularized additive models. Inspired by the supervised work (Christmann & Zhou, 2016), we give some necessary assumptions and lemmas before deriving the bound under the additive space.

As defined in Section 2, we denote $\rho_{\mathcal{X}}$ as the marginal distribution with respect to $\mathcal{X}$. Here we further introduce $\rho_{\mathcal{X}^{(j)}}$ for $\mathcal{X}^{(j)}$, which is the $j$-th component of $\mathcal{X}$ (Christmann & Zhou, 2016; Chen et al., 2020). For completeness, we restate the settings in Assumption 2.

**Assumption 5.** *Assume $f_\rho \in L_\infty\left(\rho_{\mathcal{X}}\right)$ and $f_\rho = f_\rho^{(1)} + f_\rho^{(2)} + \ldots + f_\rho^{(p)}$ where for some $0 < r \leq \frac{1}{2}$ and for each $j \in \{1, \ldots, p\}$, the $j$-th component function $f_\rho^{(j)} : \mathcal{X}^{(j)} \to \mathbb{R}$ is a mapping: $f_\rho^{(j)} = L_{K^{(j)}}^r\left(g_j^*\right)$ with some $g_j^* \in L_2\left(\rho_{\mathcal{X}^{(j)}}\right)$.*

The case $r = \frac{1}{2}$ of Assumption 5 means each $f_\rho^{(j)}$ lies in the RKHS $K^{(j)}$. Here the operator $L_K$ is defined by

$$L_K(f)\left(X^{(1)}, \ldots, X^{(p)}\right)$$

$$= \int_{\mathcal{X}}\left(\sum_{j=1}^{p}K^{(j)}\left(X^{(j)}, X^{(j)\prime}\right)\right) f\left(X^{(1)\prime}, \ldots, X^{(p)\prime}\right) d\rho_{\mathcal{X}}\left(X^{(1)\prime}, \ldots, X^{(p)\prime}\right).$$

**Lemma 1.** *(Christmann & Zhou, 2016) Let $j \in \{1, \ldots, p\}$ and $0 < r \leq \frac{1}{2}$. Assume the $j$-th component function $f_\rho^{(j)} = L_{K^{(j)}}^r \left( g_j^* \right)$ for some $g_j^* \in L_2 \left( \rho_{\mathcal{X}^{(j)}} \right)$. Define an intermediate function $f_\lambda^{(j)}$ on $\mathcal{X}^{(j)}$ by*

$$f_\lambda^{(j)} = (L_{K^{(j)}} + \lambda I)^{-1} L_{K^{(j)}} \left( f_\rho^{(j)} \right).$$

*Then we have*

$$\left\| f_\lambda^{(j)} - f_\rho^{(j)} \right\|_{L_2\left(\rho_{X^{(j)}}\right)}^2 + \lambda \left\| f_\lambda^{(j)} \right\|_{K^{(j)}}^2 \leq \lambda^{2r} \left\| g_j^* \right\|_{L_2\left(\rho_{X^{(j)}}\right)}^2.$$

**Proposition 1.** *Under Assumption 5 and $\lambda_2 = \lambda_1^{1-r}$ where $0 < r \leq 1/2$, we have*

$$\mathcal{D}(\lambda) \leq C\lambda_1^r \quad \forall 0 < \lambda_1 \leq 1,$$

*where $C$ is the constant given by*

$$C = \sum_{j=1}^p \left( L \left\| g_j^* \right\|_{L_2\left(\rho_{\mathcal{X}^{(j)}}\right)} + \left( 2\omega\kappa^2 + \max_j\{\tau_j\} \right) \left\| g_j^* \right\|_{L_2\left(\rho_{\mathcal{X}^{(j)}}\right)}^2 \right).$$

*Proof.* Observe that $f_\lambda^{(j)} \in H_{K^{(j)}}$ and $\sum_j^p f_\lambda^{(j)} \in H_K$. By the definition of the regularization error, we have

$$\mathcal{D}(\lambda) = \mathcal{E}\left(f_\lambda\right) - \mathcal{E}\left(f_\rho\right) + \lambda_1 \sum_{j=1}^p \tau_j \left\| f_\lambda^{(j)} \right\|_{K^{(j)}}^2 + \lambda_2 \sum_{j=1}^p \left\langle f_\lambda^{(j)}, L_\omega f_\lambda^{(j)} \right\rangle_2$$

Denote

$$\mathcal{D}_1(\lambda) = \mathcal{E}\left(f_\lambda\right) - \mathcal{E}\left(f_\rho\right) + \lambda_1 \sum_{j=1}^p \tau_j \left\| f_\lambda^{(j)} \right\|_{K^{(j)}}^2.$$

By Theorem 1 of (Christmann & Zhou, 2016), based on the additive hypothesis with $p$ components in Assumption 1 and the $L$-Lipschitz property, we can rewrite

$$\mathcal{E}\left(f_\lambda\right) - \mathcal{E}\left(f_\rho\right) = \mathcal{E}\left(f_\lambda^{(1)} + \cdots + f_\lambda^{(p)}\right) - \mathcal{E}\left(f_\rho^{(1)} + \cdots + f_\rho^{(p)}\right)$$

$$\leq L \sum_{j=1}^p \int_{\mathcal{X}^{(j)}} \left| f_\lambda^{(j)} \left( X^{(j)} \right) - f_\rho^{(j)} \left( X^{(j)} \right) \right| d\rho_{\mathcal{X}^{(j)}} \left( X^{(j)} \right)$$

$$\leq L \left\| f_\lambda^{(j)} - f_\rho^{(j)} \right\|_{L_2\left(\rho_{\mathcal{X}^{(j)}}\right)}.$$

With Lemma 1, we can further derive that

$$\left\| f_\lambda^{(j)} - f_\rho^{(j)} \right\|_{L_2\left(\rho_{\mathcal{X}^{(j)}}\right)}^2 \leq \lambda_1^{2r} \left\| g_j^* \right\|_{L_2\left(\rho_{\mathcal{X}^{(j)}}\right)}^2,$$

and

$$\lambda_1 \left\| f_\lambda^{(j)} \right\|_{K^{(j)}}^2 \leq \lambda_1^{2r} \left\| g_j^* \right\|_{L_2\left(\rho_{\mathcal{X}^{(j)}}\right)}^2.$$

Thus we have

$$\mathcal{D}(\lambda) \leq \mathcal{D}_1(\lambda) + \lambda_2 \sum_{j=1}^p \left\langle f_\lambda^{(j)}, L_\omega f_\lambda^{(j)} \right\rangle_2,$$

where $0 \leq \lambda_1 \leq 1, 0 < r \leq 1/2$ and

$$\mathcal{D}_1(\lambda) \leq \sum_{j=1}^p \left( L\lambda_1^r \left\| g_j^* \right\|_{L_2\left(\rho_{\mathcal{X}^{(j)}}\right)} + \lambda_1^{2r} \max_j\{\tau_j\} \left\| g_j^* \right\|_{L_2\left(\rho_{\mathcal{X}^{(j)}}\right)}^2 \right)$$

$$\leq \lambda_1^r \sum_{j=1}^p \left( L \left\| g_j^* \right\|_{L_2\left(\rho_{\mathcal{X}^{(j)}}\right)} + \max_j\{\tau_j\} \left\| g_j^* \right\|_{L_2\left(\rho_{\mathcal{X}^{(j)}}\right)}^2 \right).$$

From the fact that $(f_\lambda(x) - f_\lambda(x'))^2 W(x,x') \le 4\omega \|f_\lambda\|_\infty^2$ and $\|f_\lambda\|_\infty \le \kappa \|f_\lambda\|_K$. With the definition of $\langle f, L_\omega f \rangle_2 = \frac{1}{2} \iint (f(x) - f(x'))^2 W(x,x') d\rho_X(x) d\rho_X(x')$ and the inequalities above, we have

$$\|f_\lambda\|_K^2 \le \sum_{j=1}^p \left\| f_\lambda^{(j)} \right\|_{K^{(j)}}^2 \le \lambda_1^{2r-1} \sum_{j=1}^p \left\| g_j^* \right\|_{L_2\left(\rho_{\mathcal{X}^{(j)}}\right)}^2 .$$

By setting $\lambda_2 = \lambda_1^{1-r}$ where $0 < r \le 1/2$, we can derive

$$\lambda_2 \langle f_{\lambda_1}, L_\omega f_{\lambda_1} \rangle_2 \le 2\omega \kappa^2 \lambda_2 \lambda_1^{2r-1} \sum_{j=1}^p \left\| g_j^* \right\|_{L_2\left(\rho_{\mathcal{X}^{(j)}}\right)}^2 \le 2\omega \kappa^2 \lambda_1^r \sum_{j=1}^p \left\| g_j^* \right\|_{L_2\left(\rho_{\mathcal{X}^{(j)}}\right)}^2 .$$

Combining the above inequalities, then the desired bound is derived. $\qquad\square$

### C.3    Bounding Sample Error $\mathcal{S}(\mathbf{z}, \lambda)$

In this section, we aim to bound the sample error term, which could be written as

$$\mathcal{S}(\mathbf{z}, \lambda) = \mathcal{S}_1(\mathbf{z}, \lambda) + \mathcal{S}_2(\mathbf{z}, \lambda),$$

where

$$\mathcal{S}_1(\mathbf{z}, \lambda) = \{\mathcal{E}(\pi(f_{\mathbf{z}})) - \mathcal{E}(f_\rho)\} - \{\mathcal{E}_{\mathbf{z}}(\pi(f_{\mathbf{z}})) - \mathcal{E}_{\mathbf{z}}(f_\rho)\} \tag{23}$$

and

$$\mathcal{S}_2(\mathbf{z}, \lambda) = \{\mathcal{E}_{\mathbf{z}}(f_\lambda) - \mathcal{E}_{\mathbf{z}}(f_\rho)\} - \{\mathcal{E}(f_\lambda) - \mathcal{E}(f_\rho)\} . \tag{24}$$

Before bounding above $\mathcal{S}_1(\mathbf{z}, \lambda)$ and $\mathcal{S}_2(\mathbf{z}, \lambda)$, we introduce the following definitions and lemmas.

**Definition 4.** *Define the ball $\mathcal{B}_r$ associated with the function space $\mathcal{H}_K$ as*

$$\mathcal{B}_r = \{f \in \mathcal{H}_K : \|f\|_K \le r\} .$$

**Definition 5.** *Let $C^v$ be a $\nu$-times continuously differentiable function set. Then, for $K^{(j)} \in C^\nu\left(\mathcal{X}^{(j)} \times \mathcal{X}^{(j)}\right), j \in \{1, \ldots, p\}$, define*

$$\zeta = \begin{cases} \frac{2}{1+2v}, & v \in (0, 1] \\ \frac{2}{1+v}, & v \in (1, 3/2] \\ \frac{1}{v}, & v \in (3/2, \infty). \end{cases}$$

Now, we introduce the empirical covering number to measure the capacity of $\mathcal{B}_r$.

**Definition 6.** *Let $\mathcal{F}$ be a set of measurable functions on $\mathcal{X}$ and $\mathbf{x} = \{x_1, x_2, \ldots, x_n\} \subset \mathcal{X}$. The $\ell_2$-empirical metric for $f_1, f_2 \in \mathcal{F}$ is $d_{2,\mathbf{x}}(f_1, f_2) = \sqrt{\frac{1}{n} \sum_{i=1}^n (f_1(x_i) - f_2(x_i))^2}$. Then the $\ell_2$-empirical covering number of $\mathcal{F}$ is defined as*

$$\mathcal{N}_2(\mathcal{F}, \epsilon) = \sup_{n \in \mathbb{N}} \sup_{\mathbf{x}} \mathcal{N}_{2,\mathbf{x}}(\mathcal{F}, \epsilon), \forall \epsilon > 0,$$

*where*

$$\mathcal{N}_{2,\mathbf{x}}(\mathcal{F}, \epsilon) = \inf \left\{ m \in \mathbb{N} : \exists \left\{ f^{(j)} \right\}_{j=1}^m \subset \mathcal{F}, s.t., \mathcal{F} \subset \bigcup_{j=1}^m \left\{ f \in \mathcal{F} : d_{2,\mathbf{x}}\left(f, f^{(j)}\right) < \epsilon \right\} \right\} .$$

Indeed, the empirical covering number of $\mathcal{B}_r$ has been investigated extensively in learning theory literature (Steinwart & Christmann, 2008; Shi et al., 2011; Shi, 2013; Guo & Zhou, 2013; Chen et al., 2020).

The following concentration inequality established in (Wu et al., 2007) is used for our sample error estimation.

**Lemma 2.** *(Wu et al., 2007) Let $\mathcal{G}$ be a measurable function set on $\mathcal{Z}$. Assume that there are constants $B, c, a > 0$ and $\theta \in [0,1]$ such that $\|g\|_\infty \leqslant B$, $\mathrm{E}g^2 \leqslant c(\mathrm{E}g)^\theta$ for each $g \in \mathcal{G}$. If for $0 < \zeta < 2$, $\log \mathcal{N}_2(\mathcal{G}, \epsilon) \leqslant a\epsilon^{-\zeta}, \forall \epsilon > 0$, then for any $\delta \in (0,1)$ and i.i.d observations $\{z_i\}_{i=1}^n \subset \mathcal{Z}$, there holds*

$$\mathrm{E}g - \frac{1}{n}\sum_{i=1}^n g(z_i) \leqslant \frac{1}{2}\gamma^{1-\theta}(\mathrm{E}g)^\theta + C_\zeta\gamma + 2\left(\frac{c\log(1/\delta)}{n}\right)^{\frac{1}{2-\theta}} + \frac{18B\log(1/\delta)}{n}, \forall g \in \mathcal{G}$$

*with confidence at least $1 - \delta$, where $C_\zeta$ is a constant depending only on $\zeta$ and*

$$\gamma = \max\left\{c^{\frac{2-\zeta}{4-2\theta+\zeta\theta}}(a/n)^{\frac{2}{4-2\theta+\zeta\theta}}, B^{\frac{2-\zeta}{2+\zeta}}(a/n)^{\frac{2}{2+\zeta}}\right\}.$$

**Lemma 3.** *Let $\xi$ be a random variable on a probability space $\mathcal{Z}$ satisfying $|\xi(z) - E\xi| \leq M_\xi$ for some constant $M_\xi$ and variance $\sigma_\xi$. Then, for any $\delta \in (0,1)$, there holds*

$$\frac{1}{n}\sum_{i=1}^n \xi(z_i) - \mathrm{E}\xi \leq \frac{2M_\xi\log(1/\delta)}{3n} + \sqrt{\frac{2\sigma_\xi^2\log(1/\delta)}{n}}$$

*with confidence at least $1 - \delta$.*

### C.3.1 BOUNDING $\mathcal{S}_1(\mathbf{z}, \lambda)$

in equation 23.

**Proposition 2.** *If for $0 < \zeta < 2$, $\log \mathcal{N}_2(\mathcal{G}, \epsilon) \leqslant a\epsilon^{-\zeta}, \forall \epsilon > 0$, then for any $\delta \in (0,1)$ and i.i.d observations $\{z_i\}_{i=1}^{l+u} \subset \mathcal{Z}$, under Assumptions 2, 3 and 4, there holds*

$$\mathcal{S}_1(\mathbf{z}, \lambda) \leqslant \frac{1}{2}(\mathcal{E}(\pi(f_\mathbf{z})) - \mathcal{E}(f_\rho)) + C_\zeta\gamma + \frac{32M^2\log(4/\delta)}{l+u} + \frac{144M^2\log(4/\delta)}{l+u}, \forall g \in \mathcal{G}$$

*with confidence at least $1 - \delta/4$, where $C_\zeta$ is a constant depending only on $\zeta$ and*

$$\gamma = \max\left\{(16M^2)^{\frac{2-\zeta}{2+\zeta}}(C_\zeta p^{1+\zeta}(4Mr)^\zeta/(l+u))^{\frac{2}{2+\zeta}}, (8M^2)^{\frac{2-\zeta}{2+\zeta}}(C_\zeta p^{1+\zeta}(4Mr)^\zeta/(l+u))^{\frac{2}{2+\zeta}}\right\}.$$

*Proof.* Step 1: Bounding $f_\mathbf{z}$.

Since $f_\mathbf{z}$ is dependent on the training sample set $\mathbf{z}$, we first need to find a function set containing $f_\mathbf{z}$.

$$\lambda_1\sum_{j=1}^p \tau_j\|\alpha_\mathbf{z}^{(j)}\|_2 = \lambda_1\Omega_\mathbf{z}(f_\mathbf{z}) \leq \mathcal{E}_\mathbf{z}(f_\mathbf{z}) + \lambda_1\Omega_\mathbf{z}(f_\mathbf{z}) + \frac{\lambda_2}{(l+u)^2}\sum_{j=1}^p (f_\mathbf{z}^{(j)})^T L_j f_\mathbf{z}^{(j)} \leq \mathcal{E}_\mathbf{z}(0) \leq M^2.$$

Hence we have

$$\sum_{j=1}^p \|\alpha_\mathbf{z}^{(j)}\|_2 \leq \frac{M^2}{\lambda_1\min_j \tau_j}.$$

Furthermore, based on Cauchy inequality, we can obtain

$$\|f_\mathbf{z}\|_K = \left\|\sum_{j=1}^p\sum_{i=1}^{l+n}\alpha_{\mathbf{z},i}^{(j)}K^{(j)}\left(x_i^{(j)}, \cdot\right)\right\|_K \leq \kappa\sum_{j=1}^p\sum_{i=1}^{l+u}|\alpha_{\mathbf{z},i}^{(j)}| \leq \kappa\sum_{j=1}^p\sqrt{l+u}\sqrt{\sum_{i=1}^{l+u}\|\alpha_{\mathbf{z},i}^{(j)}\|^2}$$

$$= \kappa\sqrt{l+u}\sum_{j=1}^p \|\alpha_\mathbf{z}^{(j)}\|_2.$$

Therefore, $f_\mathbf{z}$ belongs to $B_r$ with $r = \kappa\sqrt{l+u}\sum_{j=1}^p \|\alpha_\mathbf{z}^{(j)}\|_2 \leq \frac{\kappa\sqrt{l+u}M^2}{\lambda_1\min_j \tau_j}$.

Step 2: Bounding $\mathcal{S}_1(\mathbf{z}, \lambda)$ in equation 23.

Consider the function set

$$\mathcal{G} = \left\{ g(z) = (y - \pi(f)(x))^2 - (y - f_p(x))^2 \, , f \in B_r, z = (x, y) \in \mathcal{Z} \right\}.$$

For any $f_1, f_2 \in \mathcal{B}_r$, we have

$$
\begin{aligned}
g(z_1) - g(z_2) &= (y - \pi(f_1)(x))^2 - (y - \pi(f_2)(x))^2 \\
&\leq |(2y - \pi(f_1)(x) - \pi(f_2)(x))(\pi(f_1)(x) - \pi(f_2)(x))| \\
&\leq 4M|\pi(f_1)(x) - \pi(f_2)(x)|.
\end{aligned}
$$

Hence for each $K^{(j)} \in C^v(x_j, x_j)$, $j = 1, \cdots, p$, we have

$$\log \mathcal{N}_2(\mathcal{G}, \epsilon) \leqslant \log \mathcal{N}_2 \left( \mathcal{B}_r, \frac{\epsilon}{4M} \right) \leqslant \log \mathcal{N}_2 \left( \mathcal{B}_1, \frac{\epsilon}{4Mr} \right) \leqslant C_s p^{1+\zeta} (4Mr)^\zeta \epsilon^{-\zeta}, \quad (25)$$

where $\zeta$ is defined in Definition 5, and the last inequality follows from the covering number bounds for $\mathcal{H}_{K^{(j)}}$ with $K^{(j)} \in C^v$ (see (Shi, 2013; Shi et al., 2011; Wang et al., 2021)).

Considering $0 \leq (y - \pi(f)(x))^2 \leq 4M^2$ and $0 \leq (y - f_\rho(x))^2 \leq 4M^2$, we have

$$|g(z)| \leq 8M^2, \quad |g(z) - \mathrm{E}(g)| \leq 16M^2,$$

and

$$\mathrm{E}g^2 = \int (2y - \pi(f)(x) - f_p(x))^2 \, (\pi(f)(x) - f_p(x))^2 \, d\rho \leqslant 16M^2 \mathrm{E}(g).$$

By applying Lemma 2 with $a = C_\zeta p^{1+\zeta}(4Mr)^\zeta$, $B = 8M^2$, $c = 16M^2$ and $\theta = 1$, $C_\zeta$ is the constant depending only on $\zeta$.

Therefore, we have the desired results for bounding $S_1$ with confidence of $1 - \delta/4$. $\qquad\square$

### C.3.2 Bounding $\mathcal{S}_2(\mathbf{z}, \lambda)$ in equation 24

**Proposition 3.** *Let Assumptions 2 and 3 hold, then for any $\delta > 0$, there holds*

$$
\begin{aligned}
\mathcal{S}_2(\mathbf{z}, \lambda) &\leq \frac{2M_\xi \log(4/\delta)}{3(l+u)} + \sqrt{\frac{2Var(\xi)^2 \log(4/\delta)d}{l+u}} \\
&\leq \frac{4 \left( 3M + \kappa \sqrt{\frac{\mathcal{D}(\lambda)}{\lambda_1 \min_j\{\tau_j\}}} \right)^2 \log(4/\delta)}{3(l+u)} + \sqrt{\frac{2\log(4/\delta)}{l+u}} \left( 3M + \kappa \sqrt{\frac{\mathcal{D}(\lambda)}{\lambda_1 \min_j\{\tau_j\}}} \right)^3 \mathcal{D}(\lambda)
\end{aligned}
$$

*with confidence at least $1 - \delta/4$.*

*Proof.* From the definition of $\mathcal{D}(\lambda)$ and $f_\lambda$, we can deduce that

$$\|f_\lambda\|_K^2 \leq \frac{\mathcal{D}(\lambda)}{\lambda_1 \min_j\{\tau_j\}},$$

and

$$\|f_\lambda\|_\infty \leq \kappa \|f_\lambda\|_K \leq \kappa \sqrt{\frac{\mathcal{D}(\lambda)}{\lambda_1 \min_j\{\tau_j\}}}.$$

Denote $\xi(z) = (y - f_\lambda(z))^2 - (y - f_\rho(x))^2$, we have

$$|\xi(z)| = |2y - f_\lambda(x) - f_\rho(x)| \cdot |f_\lambda(x) - f_\rho(x)| \leq \left( 3M + \kappa \sqrt{\frac{\mathcal{D}(\lambda)}{\lambda_1 \min_j\{\tau_j\}}} \right)^2 := d$$

Then

$$|\xi(z) - \mathrm{E}\xi| \leq 2d := M_\xi,$$

and

$$E\xi^2 = \int |2y - f_\lambda(x) - f_\rho(x)|^2 \cdot |f_\lambda(x) - f_\rho(x)|^2 d\rho_x$$

$$\leq \left(3M + \kappa\sqrt{\frac{\mathcal{D}(\lambda)}{\lambda_1 \min_j\{\tau_j\}}}\right)^2 \|f_\lambda(x) - f_\rho(x)\|_{\rho_x}^2$$

$$\leq d(\mathcal{E}(f_\lambda) - \mathcal{E}(f_\rho))$$

$$\leq d\mathcal{D}(\lambda).$$

Moreover,

$$\mathrm{Var}(\xi) \leq \mathrm{E}(\xi^2) \leq d\mathcal{D}(\lambda).$$

Applying the one side Bernstein inequality in Lemma 3 with $M_\xi = 2d$, $Var(\xi) \leq d\mathcal{D}(\lambda)$ and $d = \left(3M + \kappa\sqrt{\frac{\mathcal{D}(\lambda)}{\lambda_1 \min_j\{\tau_j\}}}\right)^2$, we get

$$\mathcal{S}_2(\mathbf{z}, \lambda) \leq \frac{2M_\xi \log(4/\delta)}{3(l+u)} + \sqrt{\frac{2Var(\xi)^2 \log(4/\delta)d}{l+u}}$$

$$\leq \frac{4\left(3M + \kappa\sqrt{\frac{\mathcal{D}(\lambda)}{\lambda_1 \min_j\{\tau_j\}}}\right)^2 \log(4/\delta)}{3(l+u)} + \sqrt{\frac{2\log(4/\delta)}{l+u}}\left(3M + \kappa\sqrt{\frac{\mathcal{D}(\lambda)}{\lambda_1 \min_j\{\tau_j\}}}\right)^3 \mathcal{D}(\lambda)$$

with confidence at least $1 - \delta/4$. $\qquad\qquad\qquad\qquad\qquad\qquad\qquad\qquad\qquad\square$

The desired upper bound of $S$ is obtained by combining the above estimations for $S_1$ and $S_2$.

C.4  BOUNDING HYPOTHESIS ERROR $\mathcal{H}(\mathbf{z}, \lambda)$

Before bounding $\mathcal{H}(\mathbf{z}, \lambda)$, we first introduce the auxiliary function

$$f_{\mathbf{z},\lambda} = \arg\min\left\{\frac{1}{l}\sum_{i=1}^{l}(y_i - f(x_i))^2 + \lambda_1 \sum_{j=1}^{p} \tau_j \|f^{(j)}\|_{K^{(j)}}^2 + \frac{\lambda_2}{(l+u)^2}\mathbf{f}^T L_w \mathbf{f}\right\}, \quad (26)$$

which enjoys the representation

$$f_{\mathbf{z},\lambda}(x_i) = \sum_{j=1}^{p}(\mathbf{K}_i^{(j)})^T \hat{\alpha}_{\mathbf{z}}^{(j)}.$$

Here $\mathbf{K}_i^{(j)} = (K^{(j)}(x_1^{(j)}, x_i^{(j)}), K^{(j)}(x_2^{(j)}, x_i^{(j)}), \cdots, K^{(j)}(x_{l+u}^{(j)}, x_i^{(j)})) \in \mathbb{R}^{l+u}$ and $\hat{\alpha}_{\mathbf{z}}^{(j)} = (\hat{\alpha}_{\mathbf{z},1}^{(j)}, \cdots, \hat{\alpha}_{\mathbf{z},l+u}^{(j)}) \in \mathbb{R}^{l+u}$.

**Remark 9.** *Based on the assumptions of boundedness (Assumption 2), we can naturally assume that the introduced function $\mathbf{f}_{\mathbf{z},\lambda}$ in (26) has a bounded output. That is, $\|\mathbf{f}_{\mathbf{z},\lambda}\|_\infty \leq \infty$ and $\|\mathbf{f}_{\mathbf{z},\lambda}^{(j)}\|_\infty \leq \infty$.*

Inspired by Lemma 4 of (Chen et al., 2020) and Lemma 5 of (Wang et al., 2023), we further build the following key lemma for deriving the upper bound of hypothesis error.

**Lemma 4.** *For $f_{\mathbf{z},\lambda}$ defined in (26), there exists*

$$\tau_j \|\hat{\alpha}_{\mathbf{z}}^{(j)}\|_2 \leq \frac{M + \|f_{\mathbf{z},\lambda}\|_\infty}{\lambda_1 \sqrt{l}} + \frac{\lambda_2 w \|\mathbf{f}_{\mathbf{z},\lambda}^{(j)}\|_\infty}{\lambda_1(l+u)}.$$

*Proof.* Based the definition of $f_{\mathbf{z},\lambda}$, we can deduce that

$$
\begin{aligned}
\frac{\partial f_{\mathbf{z},\lambda}}{\partial \alpha^{(j)}} =& \frac{2}{l} \sum_{i=1}^{l} (y_i - f_{\mathbf{z},\lambda}(x_i)(-(\mathbf{K}_i^{(j)})^T)) + 2\lambda_1 \tau_j (\hat{\alpha}_{\mathbf{z}}^{(j)})^T \mathbf{K}^{(j)} + \frac{\lambda_2 L_j \mathbf{f}_{\mathbf{z},\lambda}^{(j)} \mathbf{K}^{(j)}}{(l+u)^2} \\
=& \frac{2}{l} \left( \underbrace{y_1 - f_{\mathbf{z},\lambda}(x_1), \cdots, y_l - f_{\mathbf{z},\lambda}(x_l)}_{l \quad Items}, \underbrace{0, \quad \cdots, 0}_{u \quad Items} \right)^T (-\mathbf{K}^{(j)}) + 2\lambda_1 \tau_j (\hat{\alpha}_{\mathbf{z}}^{(j)})^T \mathbf{K}^{(j)} \\
& + \frac{2\lambda_2 L_j \mathbf{f}_{\mathbf{z},\lambda}^{(j)} \mathbf{K}^{(j)}}{(l+u)^2},
\end{aligned}
$$

where $\mathbf{K}^{(j)} = (K^{(j)}(x_a^{(j)}, x_b^{(j)}))_{a,b=1}^{l+u} \in \mathbb{R}^{(l+u) \times (l+u)}$.

When satisfying $\frac{\partial f_{\mathbf{z},\lambda}}{\partial \alpha^{(j)}} = 0$, we have

$$
\tau_j (\hat{\alpha}_{\mathbf{z}}^{(j)})^T = \frac{1}{l\lambda_1} (y_1 - f_{\mathbf{z},\lambda}(x_1), \cdots, y_l - f_{\mathbf{z},\lambda}(x_l), 0, \cdots, 0)^T - \frac{\lambda_2 L_j \mathbf{f}_{\mathbf{z},\lambda}^{(j)}}{\lambda_1 (l+u)^2}.
$$

Then it follows for any $j \in \{1, \cdots, p\}$,

$$
\begin{aligned}
\tau_j \|\hat{\alpha}_{\mathbf{z}}^{(j)}\|_2 &\le \frac{1}{l\lambda_1} \sqrt{\sum_{i=1}^{l} (y_i - f_{\mathbf{z},\lambda}(x_i))^2} + \frac{\lambda_2}{\lambda_1 (l+u)^2} \|L_j \mathbf{f}_{\mathbf{z},\lambda}^{(j)}\|_2 \\
&\le \frac{M + \|f_{\mathbf{z},\lambda}\|_\infty}{\lambda_1 \sqrt{l}} + \frac{\lambda_2 w}{\lambda_1 (l+u)^{3/2}} \|\mathbf{f}_{\mathbf{z},\lambda}^{(j)}\|_\infty,
\end{aligned}
$$

where $L_j \mathbf{f}_{\mathbf{z},\lambda}^{(j)}$ could also be rewritten as the sum of $l + u$ components. $\qquad \square$

Based on the above conclusions, we give the proof for bounding $\mathcal{H}(\mathbf{z}, \lambda)$.

**Proposition 4.** *The hypothesis error $\mathcal{H}(\mathbf{z}, \lambda)$ defined in Theorem 3 could be bounded by*

$$
\mathcal{H}(\mathbf{z}, \lambda) \le p \left( \frac{(M + \|f_{\mathbf{z},\lambda}\|_\infty)}{\sqrt{l}} + \frac{\lambda_2 w \|\mathbf{f}_{\mathbf{z},\lambda}\|_\infty}{(l+u)^{3/2}} \right),
$$

*where $f_{\mathbf{z},\lambda}$ is defined in equation 26.*

*Proof.* Recall the definitions of $f_{\mathbf{z}}$, $f_\lambda$ and $f_{\mathbf{z},\lambda}$, we have

$$
\begin{aligned}
\mathcal{E}_{\mathbf{z}}(f_{\mathbf{z}}) &\le \mathcal{E}_{\mathbf{z}}(f_{\mathbf{z}}) + \lambda_1 \Omega(f_{\mathbf{z}}) + \frac{\lambda_2}{(l+u)^2} \sum_{j=1}^{p} (\mathbf{f}_{\mathbf{z}}^{(j)})^T L_j \mathbf{f}_{\mathbf{z}}^{(j)} \\
&\le \mathcal{E}_{\mathbf{z}}(f_{\mathbf{z},\lambda}) + \lambda_1 \Omega(f_{\mathbf{z},\lambda}) + \frac{\lambda_2}{(l+u)^2} \sum_{j=1}^{p} (\mathbf{f}_{\mathbf{z},\lambda}^{(j)})^T L_j \mathbf{f}_{\mathbf{z},\lambda}^{(j)},
\end{aligned}
$$

and

$$
\begin{aligned}
\mathcal{E}_{\mathbf{z}}(f_{\mathbf{z},\lambda}) &+ \lambda_1 \sum_{j=1}^{p} \tau_j \|f_{\mathbf{z},\lambda}^{(j)}\|_{K^{(j)}}^2 + \frac{\lambda_2}{(l+u)^2} \sum_{j=1}^{p} (\mathbf{f}_{\mathbf{z},\lambda}^{(j)})^T L_j \mathbf{f}_{\mathbf{z},\lambda}^{(j)} \\
&\le \mathcal{E}_{\mathbf{z}}(f_\lambda) + \lambda_1 \sum_{j=1}^{p} \tau_j \|f_\lambda^{(j)}\|_{K^{(j)}}^2 + \frac{\lambda_2}{(l+u)^2} \sum_{j=1}^{p} (\mathbf{f}_\lambda^{(j)})^T L_j \mathbf{f}_\lambda^{(j)}.
\end{aligned}
$$

Then based on the definition of $\mathcal{H}(\mathbf{z}, \lambda)$, we can derive that

$$
\begin{aligned}
\mathcal{H}(\mathbf{z}, \lambda) =& \mathcal{E}_{\mathbf{z}}\left(\pi(f_{\mathbf{z}})\right) + \lambda_1 \Omega(f_{\mathbf{z}}) + \frac{\lambda_2}{(l+u)^2} \sum_{j=1}^{p} (\mathbf{f}_{\mathbf{z}}^{(j)})^T L_j \mathbf{f}_{\mathbf{z}}^{(j)} \\
& - \left\{ \mathcal{E}_{\mathbf{z}}\left(f_{\lambda}\right) + \lambda_1 \sum_{j=1}^{p} \tau_j \|f_{\lambda}^{(j)}\|_{K^{(j)}}^2 + \frac{\lambda_2}{(l+u)^2} \sum_{j=1}^{p} (\mathbf{f}_{\lambda}^{(j)})^T L_j \mathbf{f}_{\lambda}^{(j)} \right\} \\
\leq& \mathcal{E}_{\mathbf{z}}\left(f_{\mathbf{z}, \lambda}\right) + \lambda_1 \Omega(f_{\mathbf{z}, \lambda}) + \frac{\lambda_2}{(l+u)^2} \sum_{j=1}^{p} (\mathbf{f}_{\mathbf{z}, \lambda}^{(j)})^T L_j \mathbf{f}_{\mathbf{z}, \lambda}^{(j)} \\
& - \left\{ \mathcal{E}_{\mathbf{z}}\left(f_{\mathbf{z}, \lambda}\right) + \lambda_1 \sum_{j=1}^{p} \tau_j \|f_{\mathbf{z}, \lambda}^{(j)}\|_{K^{(j)}}^2 + \frac{\lambda_2}{(l+u)^2} \sum_{j=1}^{p} (\mathbf{f}_{\mathbf{z}, \lambda}^{(j)})^T L_j \mathbf{f}_{\mathbf{z}, \lambda}^{(j)} \right\} \\
\leq& \lambda_1 \Omega(f_{\mathbf{z}, \lambda}),
\end{aligned}
$$

and based on Lemma 4, we have

$$
\lambda_1 \Omega(f_{\mathbf{z}, \lambda}) = \lambda_1 \sum_{j=1}^{p} \tau_j \|\hat{\alpha}_{\mathbf{z}}^{(j)}\|_2 \leq p \left( \frac{M + \|f_{\mathbf{z}, \lambda}\|_{\infty}}{\sqrt{l}} + \frac{\lambda_2 w \max\limits_{j=1, \cdots, p} \|\mathbf{f}_{\mathbf{z}, \lambda}^{(j)}\|_{\infty}}{(l+u)^{3/2}} \right).
$$

The desired results can be obtained by combining the above inequalities. $\qquad\square$

## C.5 Bounding Manifold Error $\mathcal{M}(\mathbf{z}, \lambda)$

Recall the definition of $\mathcal{M}(\mathbf{z}, \lambda)$, we have

$$
\mathcal{M}(\mathbf{z}, \lambda) = \frac{\lambda_2}{(l+u)^2} \sum_{j=1}^{p} (\mathbf{f}_{\lambda}^{(j)})^T L_j \mathbf{f}_{\lambda}^{(j)} - \lambda_2 \sum_{j=1}^{p} \left\langle f_{\lambda}^{(j)}, L_{\omega} f_{\lambda}^{(j)} \right\rangle_2.
$$

The manifold error can be derived by bounding each of the terms with a reasonable assumption that the random variables on similarity measure $\mathcal{W}(\cdot, x)$, $f_{\lambda}(x)\mathcal{W}(\cdot, x)$ as well as $f_{\lambda}^2(x)\mathcal{W}(x, \cdot)$ lie in the additive space of RKHS. Thus we further divide the manifold error into the following 4 parts:

$$
\mathcal{M}(\mathbf{z}, \lambda) = \mathcal{M}_1(\mathbf{z}, \lambda) + \mathcal{M}_2(\mathbf{z}, \lambda) + \mathcal{M}_3(\mathbf{z}, \lambda) + \mathcal{M}_4(\mathbf{z}, \lambda),
$$

where

$$
\mathcal{M}_1(\mathbf{z}, \lambda) = \frac{\lambda_2}{l+u} \sum_{i=1}^{l+u} \left( \frac{1}{l+u} \sum_{k=1}^{l+u} f_{\lambda}^2(x_k) \mathcal{W}(x_k, x_i) - \int f_{\lambda}^2(x) \mathcal{W}(x, x_i) d\rho_{\mathcal{X}}(x) \right), \qquad (27)
$$

$$
\mathcal{M}_2(\mathbf{z}, \lambda) = \lambda_2 \int f_{\lambda}^2(x) \left( \frac{1}{l+u} \sum_{i=1}^{l+u} \mathcal{W}(x, x_i) - \int \mathcal{W}(x, x') d\rho_{\mathcal{X}}(x') \right) d\rho_{\mathcal{X}}(x), \qquad (28)
$$

$$
\mathcal{M}_3(\mathbf{z}, \lambda) = \frac{\lambda_2}{l+u} \sum_{i=1}^{l+u} f_{\lambda}(x_i) \left( \int f_{\lambda}(x) \mathcal{W}(x, x_i) d\rho_{\mathcal{X}}(x) - \frac{1}{l+u} \sum_{k=1}^{l+u} f_{\lambda}(x) \mathcal{W}(x_k, x_i) \right), \qquad (29)
$$

and

$$
\mathcal{M}_4(\mathbf{z}, \lambda) = \lambda_2 \int f_{\lambda}(x) \left( \int f_{\lambda}(x') \mathcal{W}(x, x') d\rho_{\mathcal{X}}(x') - \frac{1}{l+u} \sum_{i=1}^{l+u} f_{\lambda}(x_i) \mathcal{W}(x, x_i) \right) d\rho_{\mathcal{X}}(x). \qquad (30)
$$

To analyze the above 4 terms to bound the manifold error, we introduce the following techniques.

**Lemma 5.** *(Smale & Zhou, 2007) Let $\xi$ be a random variable on $\mathcal{Z}$ in a Hilbert space $\mathcal{H}$, which satisfies $\|\xi\| \leq M_\xi$. Denote $Var(\xi) = \sigma_\xi^2 = E(\|\xi\|^2)$. Then for any $\delta \in (0, 1)$, there holds*

$$\|\frac{1}{l+u}\sum_{i=1}^{l+u}[\xi_i - \mathrm{E}(\xi)]\| \leq \frac{2M_\xi \log(\frac{2}{\delta})}{l+u} + \left(\frac{2\sigma_\xi^2 \log(\frac{2}{\delta})}{l+u}\right)^{\frac{1}{2}}$$

*with confidence $1 - \delta$.*

**Proposition 5.** *For all $\delta \in (0, 1)$, with confidence at least $1 - \delta$, there holds*

$$\mathcal{M}(\mathbf{z}, \lambda) \leq \frac{8w\lambda_2\kappa^2\mathcal{D}(\lambda)\log(8/\delta)}{\lambda_1 \min_j\{\tau_j\}}(l+u)^{-\frac{1}{2}}.$$

*Proof.* Step 1: Bounding $\mathcal{M}_1(\mathbf{z}, \lambda)$ in equation 27. Based on the definition of $f_\lambda$, we have

$$\|f_\lambda^2(x)\mathcal{W}(x, \cdot)\| \leq w\|f_\lambda\|_\infty^2$$

since $\|f_\lambda\|_\infty \leq \kappa\|f_\lambda\|_K \leq \kappa\sqrt{\frac{\mathcal{D}(\lambda)}{\lambda_1 \min_j\{\tau_j\}}}$.

Thus we have

$$M_\xi = \|f_\lambda^2(x)\mathcal{W}(x, \cdot)\| \leq \frac{w\kappa^2\mathcal{D}(\lambda)}{\lambda_1 \min_j\{\tau_j\}}.$$

and

$$\sigma_\xi^2 = E[\|f_\lambda^2(x)\mathcal{W}(x, \cdot)\|^2] \leq \frac{w^2\kappa^4\mathcal{D}^2(\lambda)}{\lambda_1^2 \min_j\{\tau_j\}^2}.$$

Applying Lemma 5, we can derive that

$$\mathcal{M}_1(\mathbf{z}, \lambda) \leq \lambda_2\left(\frac{2\log(\frac{8}{\delta})}{l+u}\frac{w\kappa^2\mathcal{D}(\lambda)}{\lambda_1 \min_j \tau_j} + \sqrt{\frac{2\log(\frac{8}{\delta})}{l+u}}\frac{w\kappa^2\mathcal{D}(\lambda)}{\lambda_1 \min_j\{\tau_j\}}\right)$$

$$\leq \frac{\lambda_2 w\kappa^2\mathcal{D}(\lambda)}{\lambda_1 \min_j \tau_j}\left(\frac{2\log(\frac{8}{\delta})}{l+u} + \sqrt{\frac{2\log(\frac{8}{\delta})}{l+u}}\right)$$

$$\leq \frac{4\lambda_2 w\kappa^2\mathcal{D}(\lambda)\log(\frac{8}{\delta})}{\sqrt{l+u}\lambda_1 \min_j \tau_j}$$

with confidence of $1 - \delta/4$.

Step 2: Bounding $\mathcal{M}_2(\mathbf{z}, \lambda)$ in equation 28. Note that $\|\mathcal{W}(\cdot, x)\| \leq w$, $E[\|\mathcal{W}(\cdot, x)\|^2] \leq w^2$.

Then, with confidence of $1 - \frac{\delta}{4}$, we have

$$\mathcal{M}_2(\mathbf{z}, \lambda) \leq \lambda_2\int f_\lambda^2(x)w\left(\frac{2\log(8/\delta)}{l+u} + \sqrt{\frac{2\log(8/\delta)}{l+u}}\right)d\rho_\mathcal{X}(x)$$

$$\leq \lambda_2 w\left(\frac{2\log(8/\delta)}{l+u} + \sqrt{\frac{2\log(8/\delta)}{l+u}}\right)\int f_\lambda^2(x)d\rho_\mathcal{X}(x)$$

$$\leq \lambda_2 w\frac{4\log(8/\delta)}{\sqrt{l+u}}\frac{w\kappa^2\mathcal{D}(\lambda)}{\lambda_1 \min_j \tau_j}$$

$$\leq \frac{4\lambda_2 w\kappa^2\mathcal{D}(\lambda)}{\sqrt{l+u}\lambda_1 \min_j \tau_j}\log(\frac{8}{\delta}).$$

Step 3: Bounding $\mathcal{M}_3(\mathbf{z}, \lambda)$ in equation 29. It is easy to deduce that

$$\|f_\lambda(x)\mathcal{W}(\cdot, x)\| \leq w\kappa\sqrt{\frac{\mathcal{D}(\lambda)}{\lambda_1 \min_j \tau_j}},$$

and

$$E[\|f_\lambda(x)\mathcal{W}(\cdot,x)\|^2] \le w^2\kappa^2\frac{\mathcal{D}(\lambda)}{\lambda_1\min_j\tau_j}.$$

Then, with confidence of $1 - \frac{\delta}{4}$, we can derive that

$$\mathcal{M}_3(\mathbf{z},\lambda) = \frac{\lambda_2}{l+u}\sum_{i=1}^{l+u}f_\lambda(x_i)\left(\int f_\lambda(x)\mathcal{W}(x,x_i)d\rho_\mathcal{X}(x) - \frac{1}{l+u}\sum_{k=1}^{l+u}f_\lambda(x)\mathcal{W}(x_k,x_i)\right)$$

$$\le \frac{\lambda_2}{l+u}\sum_{i=1}^{l+u}f_\lambda(x_i)w\kappa\sqrt{\frac{\mathcal{D}(\lambda)}{\lambda_1\min_j\tau_j}}\left(\frac{2\log(\frac{8}{\delta})}{l+u} + \sqrt{\frac{2\log(\frac{8}{\delta})}{l+u}}\right)$$

$$\le \lambda_2 w\kappa^2\frac{\mathcal{D}(\lambda)}{\lambda_1\min_j\tau_j}\frac{4\log(\frac{8}{\delta})}{\sqrt{l+u}}$$

$$\le \frac{4\lambda_2 w\kappa^2\mathcal{D}(\lambda)}{\sqrt{l+u}\lambda_1\min_j\tau_j}\log(\frac{8}{\delta}).$$

Step 4: Bounding $\mathcal{M}_4(\mathbf{z},\lambda)$ in equation 30. Finally, we can deduce that with confidence of $1 - \delta/4$,

$$\mathcal{M}_4(\mathbf{z},\lambda) \le \lambda_2\int f_\lambda(x)w\kappa\sqrt{\frac{\mathcal{D}(\lambda)}{\lambda_1\min_j\tau_j}}\left(\frac{2\log(\frac{8}{\delta})}{l+u} + \sqrt{\frac{2\log(\frac{8}{\delta})}{l+u}}\right)d\rho_\mathcal{X}(x)$$

$$\le \lambda_2 w\kappa\sqrt{\frac{\mathcal{D}(\lambda)}{\lambda_1\min_j\tau_j}}2\frac{2\log(\frac{8}{\delta})}{\sqrt{l+u}}\int f_\lambda(x)d\rho_\mathcal{X}(x)$$

$$\le \frac{4\lambda_2 w\kappa^2\mathcal{D}(\lambda)}{\sqrt{l+u}\lambda_1\min_j\tau_j}\log(\frac{8}{\delta}).$$

The desired result follows by combining the above estimations.

$\square$

## C.6 PROOF OF THEOREM 2

Then we summarize the above conclusions and analyze the learning rate under mild assumptions.

**Proposition 6.** *Let Assumptions 2-4 be ture. For any $\delta \in (0,1/2)$, the following conclusion holds with confidence $1 - 2\delta$ there holds*

$$\mathcal{E}(\pi(f_\mathbf{z})) - \mathcal{E}(f_\rho)$$
$$\le \mathcal{D}(\lambda) + \mathcal{S}(\mathbf{z},\lambda) + \mathcal{H}(\mathbf{z},\lambda) + \mathcal{M}(\mathbf{z},\lambda)$$
$$\le C_r\lambda_1^r + \frac{1}{2}\left(\mathcal{E}(\pi(f_\mathbf{z})) - \mathcal{E}(f_\rho)\right) + C_\zeta\gamma + \frac{32M^2\log(4/\delta)}{l+u} + \frac{144M^2\log(4/\delta)}{l+u}$$
$$+ \frac{4\left(3M + \kappa\sqrt{\frac{\mathcal{D}(\lambda)}{\lambda_1\min_j\{\tau_j\}}}\right)^2\log(4/\delta)}{3l} + \sqrt{\frac{2\log(4/\delta)d}{l}}\left(3M + \kappa\sqrt{\frac{\mathcal{D}(\lambda)}{\lambda_1\min_j\{\tau_j\}}}\right)^2\mathcal{D}(\lambda)$$
$$+ p\left(\frac{(M + \|f_{\mathbf{z},\lambda}\|_\infty)}{\sqrt{l}} + \frac{\lambda_2 w\|\mathbf{f}_{\mathbf{z},\lambda}^{(j)}\|_\infty}{(l+u)^{3/2}}\right) + \frac{16\lambda_2 w\kappa^2\mathcal{D}(\lambda)}{\sqrt{l+u}\lambda_1\min_j\tau_j}\log(\frac{8}{\delta}),$$

*where*

$$C_r = \sum_{j=1}^{p}\left(L\|g_j^*\|_{L_2\left(\rho_{\mathcal{X}^{(j)}}\right)} + \left(2\omega\kappa^2 + \max_j\{\tau_j\}\right)\|g_j^*\|_{L_2\left(\rho_{\mathcal{X}^{(j)}}\right)}^2\right),$$

$$\gamma = \max\left\{(16M^2)^{\frac{2-\zeta}{2+\zeta}}(C_\zeta p^{1+\zeta}(4Mr)^\zeta/(l+u))^{\frac{2}{2+\zeta}}, (8M^2)^{\frac{2-\zeta}{2+\zeta}}(C_\zeta p^{1+\zeta}(4Mr)^\zeta/(l+u))^{\frac{2}{2+\zeta}}\right\},$$

*$C_\zeta$ is a constant, $0 < r \le 1/2$, $0 < \zeta < 2$ and $f_{\mathbf{z},\lambda}$ is defined in equation 26.*

*Proof.* The above results can be obtained by directly combining the results of Theorem 3 and Propositions 1-5. □

Now, we present the proof of Theorem 2.

*Proof.* Let $\lambda_1 = (l+u)^{-\Delta}$ and $\lambda_2 = \lambda_1^{1-r} = (l+u)^{-\Delta(1-r)}$, where $0 < r \leq 1/2$. According to Proposition 6, we have

$$
\mathcal{E}\left(\pi\left(f_{\mathbf{z}}\right)\right) - \mathcal{E}\left(f_{\rho}\right)
$$
$$
\leq C_1(l+u)^{-\Delta r} + C_2(l+u)^{-2/(2+\xi)} + C_3 \log(4/\delta)(l+u)^{-1}
$$
$$
+ C_4 \log(4/\delta)(l+u)^{\Delta(1-r)-r} + C_5 \sqrt{\log(4/\delta)}(l+u)^{\Delta(1-2r)-1/2} + C_6(l+u)^{-1/2}
$$
$$
+ C_7(l+u)^{-\Delta(1-r)-3/2} + C_8 \log(8/\delta)(l+u)^{-2\Delta-1/2}
$$
$$
\leq C_9 \log(8/\delta)\left((l+u)^{-\Delta r} + (l+u)^{-2/(2+\xi)} + (l+u)^{-1} + (l+u)^{\Delta(1-r)-r}\right.
$$
$$
\left.+ (l+u)^{\Delta(1-2r)-1/2} + (l+u)^{-1/2} + (l+u)^{-\Delta(1-r)-3/2} + (l+u)^{-2\Delta-1/2}\right)
$$
$$
\leq C_{10} \log(8/\delta)(l+u)^{-\Theta}
$$

where

$$
\Theta = \min\{\Delta r, 2/(2+\zeta), 1, r+\Delta(r-1), \Delta(2r-1)+1/2, 1/2\} = \min\{\Delta r, 2/(2+\zeta), r+\Delta(r-1)\},
$$

and $\Delta > 0$, $0 < r \leq 1/2$, $0 < \zeta < 2$. And $C_1, \cdots, C_{10}$ are positive constants independently of $l, u, \delta$ and $r$.

□

## D CONVERGENCE ANALYSIS (PROOF OF THEOREM 1)

As described in the main paper, the masks on all features are learned at the upper level of S²MAM, where a project operation for limiting informative variables is employed. Thus we mainly focus on the corresponding convergence performance of the upper level of S²MAM.

Notice that the update rule for variable $s$ in practice can be formulated by

$$
s^{t+1} = \mathcal{P}_{\mathcal{C}}\left(s^t - \eta^t \mathcal{L}_{\mathcal{B}}\left(\alpha^*(m)\right) \nabla_s \ln p\left(m \mid s^t\right)\right), \tag{31}
$$

where $\mathcal{L}_{\mathcal{B}}$ is the loss on selected sample batch $\mathcal{B}$.

Furthermore, denote the update rules with stochastic and deterministic gradient mappings as

$$
s^{t+1} = s^t - \eta^t \hat{\mathcal{G}}^t = \mathcal{P}_{\mathcal{C}}\left(s^t - \eta^t \mathcal{L}_{\mathcal{B}}\left(\alpha^*(m)\right) \nabla_s \ln p\left(m \mid s^t\right)\right),
$$
$$
s^{t+1} = s^t - \eta^t \mathcal{G}^t = \mathcal{P}_{\mathcal{C}}\left(s^t - \eta^t \nabla_s \Phi\left(s^t\right)\right).
$$

That is to say,

$$
\hat{\mathcal{G}}^t = \frac{1}{\eta^t}\left(s^t - \mathcal{P}_{\mathcal{C}}\left(s^t - \eta^t \mathcal{L}_{\mathcal{B}}\left(\alpha^*(m)\right) \nabla_s \ln p\left(m \mid s^t\right)\right)\right) = \frac{1}{\eta^t}\left(s^t - s^{t+1}\right),
$$
$$
\mathcal{G}^t = \frac{1}{\eta^t}\left(s^t - \mathcal{P}_{\mathcal{C}}\left(s^t - \eta^t \nabla_s \Phi\left(s^t\right)\right)\right).
$$

Firstly, we recall some necessary assumptions and definitions for projection operation, which have been used in existing works on algorithmic convergence analysis on projection optimization for single-level problems (Bauschke et al., 2012) and bilevel ones (Pedregosa, 2016).

Inspired by some research on bilevel optimization problems (Pedregosa, 2016; Shu et al., 2023; Zhao et al., 2023) with mini-batch settings, this paper adopts the independently and identically distributed (i.i.d.) random variables induced by the mini-batch. Notice that $\xi^{(t)} = \mathcal{L}_{\mathcal{B}}\left(\alpha^*(m)\right) \nabla_s \ln p(m \mid s^t) - \nabla_s \Phi(s^t)$ for $t \in [1, 2, \cdots, T]$ are i.i.d random variables with finite variance $\sigma^2$, since the mini-batch are drawn i.i.d with a finite number of samples. Furthermore, $\mathbb{E}\left[\xi^{(t)}\right] = 0$ since samples are drawn uniformly at random.

**Lemma 6.** *Given a compact convex set $\mathcal{C} \subset \mathbb{R}^d$ and let $\mathcal{P}_{\mathcal{C}}(\cdot)$ be the projection operator on $\mathcal{C}$, then for any $\boldsymbol{u} \in \mathbb{R}^d$ and $\boldsymbol{v} \in \mathbb{R}^d$, we have*

$$\|\mathcal{P}_{\mathcal{C}}(\boldsymbol{u}) - \mathcal{P}_{\mathcal{C}}(\boldsymbol{v})\|^2 \le (\boldsymbol{u} - \boldsymbol{v})^\top (\mathcal{P}_{\mathcal{C}}(\boldsymbol{u}) - \mathcal{P}_{\mathcal{C}}(\boldsymbol{v}))$$

**Lemma 7.** *Given a compact convex set $\mathcal{C} \subset \mathbb{R}^d$ and let $\mathcal{P}_{\mathcal{C}}(\cdot)$ be the projection operator on $\mathcal{C}$, then for any $\boldsymbol{c} \in \mathcal{C}$ and $\boldsymbol{u} \in \mathbb{R}^d, \boldsymbol{v} \in \mathbb{R}^d$, we have*

$$\|\mathcal{P}_{\mathcal{C}}(\boldsymbol{c} + \boldsymbol{u}) - \mathcal{P}_{\mathcal{C}}(\boldsymbol{c} + \boldsymbol{v})\| \le \|\boldsymbol{u} - \boldsymbol{v}\|.$$

**Remark 10.** *Considering $\boldsymbol{c} = s^t$, $\boldsymbol{u} = \eta^t \mathcal{L}_{\mathcal{B}}(\boldsymbol{\alpha}^*(\boldsymbol{m})) \nabla_{\boldsymbol{s}} \ln p(\boldsymbol{m} \mid s^t)$ and $\boldsymbol{v} = \nabla_{\boldsymbol{s}} \Phi(s^t)$, we can easily obtain that*

$$\|\hat{\mathcal{G}}^t - \mathcal{G}^t\| \le \|\mathcal{L}_{\mathcal{B}}(\boldsymbol{\alpha}^*(\boldsymbol{m})) \nabla_{\boldsymbol{s}} \ln p(\boldsymbol{m} \mid s^t) - \nabla_{\boldsymbol{s}} \Phi(s^t)\| := \|\xi^{(t)}\|.$$

In the following, we present the corresponding proof for Theorem 1.

*Proof.* Inspired from Theorem 2 in (Pedregosa, 2016), the following holds with Lemma 6 by setting $\boldsymbol{u} = s^t$ and $\boldsymbol{v} = s^t - \eta^t g^t$,

$$\|s^t - s^{t+1}\|^2 \le \eta^t (\mathcal{L}_{\mathcal{B}}(\boldsymbol{\alpha}^*(\boldsymbol{m})) \nabla_{\boldsymbol{s}} \ln p(\boldsymbol{m} \mid s^t))^T (s^t - s^{t+1}) = \eta^t (\mathcal{L}_{\mathcal{B}}(\boldsymbol{\alpha}^*(\boldsymbol{m})) \nabla_{\boldsymbol{s}} \ln p(\boldsymbol{m} \mid s^t))^T \hat{\mathcal{G}}^t.$$

Thus we have

$$\|\hat{\mathcal{G}}^t\|^2 \le \left\langle \mathcal{L}_{\mathcal{B}}(\boldsymbol{\alpha}^*(\boldsymbol{m})) \nabla_{\boldsymbol{s}} \ln p(\boldsymbol{m} \mid s^t), \hat{\mathcal{G}}^t \right\rangle.$$

Recall the random variable $\xi^{(t)} = \mathcal{L}_{\mathcal{B}}(\boldsymbol{\alpha}^*(\boldsymbol{m})) \nabla_{\boldsymbol{s}} \ln p(\boldsymbol{m} \mid s^t) - \nabla_{\boldsymbol{s}} \Phi(s^t)$ for $t \in [1, 2, \cdots, T]$. Based on the definitions of the stochastic gradient mapping $\hat{\mathcal{G}}^t$ and the $L$ smoothness of $\Phi$, we have

$$
\begin{aligned}
\Phi\left(s^{t+1}\right) - \Phi\left(s^t\right) &\le \frac{L}{2} \left\|s^{t+1} - s^t\right\|^2 - \left\langle \nabla_{\boldsymbol{s}} \Phi\left(s^t\right), s^t - s^{t+1} \right\rangle \\
&= \frac{L(\eta^t)^2}{2} \left\|\hat{\mathcal{G}}^t\right\|^2 - \eta^t \left\langle \mathcal{L}_{\mathcal{B}}(\boldsymbol{\alpha}^*(\boldsymbol{m})) \nabla_{\boldsymbol{s}} \ln p(\boldsymbol{m} \mid s^t) - \xi^{(t)}, \hat{\mathcal{G}}^t \right\rangle \\
&= \frac{L(\eta^t)^2}{2} \left\|\hat{\mathcal{G}}^t\right\|^2 - \eta^t \left\langle \mathcal{L}_{\mathcal{B}}(\boldsymbol{\alpha}^*(\boldsymbol{m})) \nabla_{\boldsymbol{s}} \ln p(\boldsymbol{m} \mid s^t), \hat{\mathcal{G}}^t \right\rangle + \eta^t \left\langle \xi^{(t)}, \hat{\mathcal{G}}^t \right\rangle \\
&\le (\frac{L(\eta^t)^2}{2} - \eta^t) \left\|\hat{\mathcal{G}}^t\right\|^2 + \eta^t \left\langle \xi^{(t)}, \mathcal{G}^t \right\rangle + \eta^t \left\langle \xi^{(t)}, \hat{\mathcal{G}}^t - \mathcal{G}^t \right\rangle \\
&\le (\frac{L(\eta^t)^2}{2} - \eta^t) \left\|\hat{\mathcal{G}}^t\right\|^2 + \eta^t \left\langle \xi^{(t)}, \mathcal{G}^t \right\rangle + \eta^t \|\xi^{(t)}\|^2 \\
&\le (L(\eta^t)^2 - 2\eta^t)(\|\mathcal{G}^t\|^2 + \|\xi^{(t)}\|^2) + \eta^t \left\langle \xi^{(t)}, \mathcal{G}^t \right\rangle + \eta^t \|\xi^{(t)}\|^2
\end{aligned}
$$

where the last line is obtained with Lemma 7 and $\left\|\hat{\mathcal{G}}^t\right\|^2 \le 2(\|\mathcal{G}^t\|^2 + \left\|\xi^{(t)}\right\|^2)$.

By summing up from $t = 1$ to $T$, we derive that

$$\sum_{t=1}^T \left(2\eta^t - L(\eta^t)^2\right) \left\|\mathcal{G}^t\right\|^2 \le \Phi\left(s^1\right) - \Phi\left(s^{T+1}\right) + \sum_{t=1}^T \left(\eta^t \left\langle \xi^{(t)}, \mathcal{G}^t \right\rangle + (L(\eta^t)^2 - \eta^t) \left\|\xi^{(t)}\right\|^2\right).$$

Since $\eta^t = \frac{c}{\sqrt{t}} \le \frac{1}{L}$, we have $2\eta^t - L\eta^t \ge \eta^t \ge 0$. Denote $(\eta^t)' = \min\{\eta^t, t = 1, \cdots, T\} = \frac{c}{\sqrt{T}}$. Then we can derive

$$\sum_{t=1}^T \left(2\eta^t - L(\eta^t)^2\right) \ge \sum_{t=1}^T \eta^t,$$

and

$$\frac{1}{\sum_{t=1}^T \left(2\eta^t - L(\eta^t)^2\right)} \le \frac{1}{\sum_{t=1}^T \eta^t} \le \frac{1}{T(\eta^t)'} = \frac{1}{c\sqrt{T}}.$$

Under the assumptions on $\mathbb{E}[\xi^{(t)}] = 0$ and $\mathbb{E}\|\xi^{(t)}\|^2 \leq \sigma^2$, we have

$$\min_{1 \leq t \leq T} \mathbb{E}\left\|\mathcal{G}^t\right\|^2 \leq \frac{\sum_{t=1}^T \left(2\eta^t - L(\eta^t)^2\right)\|\mathcal{G}^t\|^2}{\sum_{t=1}^T \left(2\eta^t - L(\eta^t)^2\right)} \leq \frac{\Phi\left(s^1\right) - \Phi\left(s^{T+1}\right) + \sum_{t=1}^T (L(\eta^t)^2 - \eta^t)\sigma^2}{c\sqrt{T}}$$

$$\leq \frac{\Phi\left(s^1\right) - \Phi\left(s^{T+1}\right)}{c\sqrt{T}},$$

where last inequality is obtained by $\eta^t \leq 1/L$ and $L(\eta^t)^2 - \eta^t \leq 0$.

Finally, it can be obtained that

$$\min_{1 \leq t \leq T} \mathbb{E}\left\|\mathcal{G}^t\right\|^2 \lesssim \mathcal{O}\left(\frac{1}{\sqrt{T}}\right).$$

$\square$

**Remark 11.** *Zhou et.al. Zhou et al. (2022) demonstrate that with assumed variance $\sigma$, smoothness parameter $\ell$ and learning rate $\eta \leq \frac{2}{\ell}$, the average gradient $\frac{1}{T}\sum_{t=1}^T \mathbb{E}\left\|\mathcal{G}^t\right\|^2$ converges to a small constant $\frac{8-2\ell\eta}{2-\ell\eta}\sigma^2$, when $T \to \infty$.*

*Differently, we further adopt the learning rate $\eta = \frac{c}{t} \leq \frac{1}{L}$ ($c > 0$), and new inequalities to further derive an improved convergence rate, $\mathcal{O}(\frac{1}{\sqrt{T}})$, which converges to zero with $T \to \infty$.*

# E    OPTIMIZATION DETAILS

## E.1    DISCRETE MASKS $\boldsymbol{m}$ TO CONTINUOUS PROBABILITY $\boldsymbol{s}$

As introduced in (Zhou et al., 2022), the probabilistic bilevel problem indeed is a tight relaxation (although not equivalent) of the original discrete problem. For completeness, we summarize the reasons for such transformation:

- The discrete masks $m = 0/1$ can be represented as a particular stochastic one by letting $s_i = 0/1$, thus we have $\min_{\boldsymbol{s}\in\mathcal{C}} \Phi(\boldsymbol{s}) \leq \min_{\boldsymbol{m}\in\tilde{\mathcal{C}}} \tilde{\Phi}(\boldsymbol{m})$;

- The constraint on $\boldsymbol{s}$ with $\ell$-1 regularization within $[0,1]$ guides the most components of the optimal $\boldsymbol{s}$ either 0 or 1, which has already been empirically validated in (Zhou et al., 2022);

- The new probabilistic form can be optimized directly with the gradient-based method as follows,

$$\nabla_{\boldsymbol{s}}\Phi(\boldsymbol{s}) = \nabla_{\boldsymbol{s}}\mathbb{E}_{p(\boldsymbol{m}|\boldsymbol{s})}\mathcal{L}\left(\boldsymbol{\alpha}^*(\boldsymbol{m})\right)$$

$$= \nabla_{\boldsymbol{s}} \int \mathcal{L}\left(\boldsymbol{\alpha}^*(\boldsymbol{m})\right) p(\boldsymbol{m} \mid \boldsymbol{s})d\boldsymbol{m}$$

$$= \int \mathcal{L}\left(\boldsymbol{\alpha}^*(\boldsymbol{m})\right) \frac{\nabla_{\boldsymbol{s}}p(\boldsymbol{m} \mid \boldsymbol{s})}{p(\boldsymbol{m} \mid \boldsymbol{s})}p(\boldsymbol{m} \mid \boldsymbol{s})d\boldsymbol{m}$$

$$= \int \mathcal{L}\left(\boldsymbol{\alpha}^*(\boldsymbol{m})\right) \nabla_{\boldsymbol{s}}\ln p(\boldsymbol{m} \mid \boldsymbol{s})p(\boldsymbol{m} \mid \boldsymbol{s})d\boldsymbol{m}$$

$$= \mathbb{E}_{p(\boldsymbol{m}|\boldsymbol{s})}\mathcal{L}\left(\boldsymbol{\alpha}^*(\boldsymbol{m})\right) \nabla_{\boldsymbol{s}}\ln p(\boldsymbol{m} \mid \boldsymbol{s}),$$

which obviously reduced the computation cost of bilevel problems.

## E.2    PROJECT OPTIMIZATION FROM PROBABILITY $\boldsymbol{s}$ TO DOMAIN $\boldsymbol{C}$

Inspired from existing works (Zhao et al., 2023; Zhou et al., 2022), the algorithm for project operation from probability $\boldsymbol{s}$ to domain $\mathcal{C}$ is realized with projection operation $\mathcal{P}_{\mathcal{C}}(\boldsymbol{s})$, which is summarized in Algorithm 2. Indeed, the Lagrangian multiplier as well as the bisection method are employed for designing this algorithm with closed form solution. The theoretical guarantee for learning masks on all samples $\boldsymbol{m} \in \mathbb{R}^N$ can be found at (Zhou et al., 2022). Moreover, this paper focuses on the masks on all variables $\boldsymbol{m} \in \mathbb{R}^p$. For completeness, we present the corresponding theoretical proof as follows.

*Proof.* Given variable $\boldsymbol{a} \in \mathbb{R}^p$, in order to project $\boldsymbol{a}$ to set $\mathcal{C}$, we introduce the following problem with constraints:

$$\min_{\boldsymbol{s} \in \mathbb{R}^p} \frac{1}{2} \|\boldsymbol{s} - \boldsymbol{a}\|^2, \text{ s.t.} \mathbf{1}^T \boldsymbol{s} \leq C \text{ and } 0 \leq s_i \leq 1,$$

where $\mathbf{1} = (1, 1, \cdots, 1) \in \mathbb{R}^p$ and $\boldsymbol{s}$ is the ideal output after projection.

The above problem can be resolved by the commonly used Lagrangian multiplier method formulated with:

$$L(\boldsymbol{s}, b) = \frac{1}{2} \|\boldsymbol{s} - \boldsymbol{a}\|^2 + b \left(\mathbf{1}^\top \boldsymbol{s} - C\right) = \frac{1}{2} \|\boldsymbol{s} - (\boldsymbol{a} - b\mathbf{1})\|^2 + b \left(\mathbf{1}^\top \boldsymbol{a} - C\right) - \frac{n}{2} b^2. \quad (32)$$

where the auxiliary variable $b \geq 0$ and $0 \leq s_i \leq 1$.

To minimize above problem equation 32 with respect to $\boldsymbol{s}$, we can derive that $\tilde{s} = \mathbf{1}_{\boldsymbol{a} - b\mathbf{1} \geq 1} + (\boldsymbol{a} - b\mathbf{1})_{1 > \boldsymbol{a} - b\mathbf{1} > 0}$.

Then we can develop two auxiliary functions as follows:

$$g(b) = L(\tilde{\boldsymbol{s}}, b) = \frac{1}{2} \left\| [\boldsymbol{a} - b\mathbf{1}]_- + [\boldsymbol{a} - (b+1)\mathbf{1}]_+ \right\|^2 + b \left(\mathbf{1}^\top \boldsymbol{a} - s\right) - \frac{n}{2} b^2$$

$$= \frac{1}{2} \left\| [\boldsymbol{a} - b\mathbf{1}]_- \right\|^2 + \frac{1}{2} \left\| [\boldsymbol{a} - (b+1)\mathbf{1}]_+ \right\|^2 + b \left(\mathbf{1}^\top \boldsymbol{a} - s\right) - \frac{n}{2} b^2, \text{ for } \quad b \geq 0,$$

and

$$g'(b) = \mathbf{1}^\top [b\mathbf{1} - \boldsymbol{a}]_+ + \mathbf{1}^\top [(b+1)\mathbf{1} - \boldsymbol{a}]_- + \left(\mathbf{1}^T \boldsymbol{a} - s\right) - nb = \mathbf{1}^\top \min(1, \max(0, \boldsymbol{a} - b\mathbf{1})) - C, \text{ for } \quad b \geq 0.$$

Finally, with the monotone decreasing property of $g'(b)$, a bisection method is exploited to solve the equation $g'(b) = 0$ with solution $b^*$. Because $g(b)$ increases in $(-\infty, b^*]$ and decreases in $[b^*, +\infty)$, we can conclude that the maximum of g($b$) is obtained at 0 if $b^* \leq 0$ and $b^*$ if $b^* > 0$.

Finally, by setting $c^* = \max(0, b^*)$, we have the output

$$\boldsymbol{s}^* = \mathbf{1}_{\boldsymbol{a} - c^*\mathbf{1} \geq 1} + (\boldsymbol{a} - c^*\mathbf{1})_{1 > \boldsymbol{a} - c^*\mathbf{1} > 0} = \min\left(1, \max\left(0, \boldsymbol{a} - c^*\mathbf{1}\right)\right).$$

$\square$

### E.3 OPTIMIZATION FOR UPPER-LEVEL PROBLEM

The detailed optimization steps for probabilistic S$^2$MAM have been already introduced in Section 2.4, which has been further summarized in Algorithm 1. Notably, this policy gradient estimation approach obviously improves the algorithmic efficiency by reducing the computation process on the hypergradient of bilevel optimization problems.

### E.4 OPTIMIZATION FOR LOWER-LEVEL PROBLEM

Based on the principle of the Alternating Direction Method of Multipliers (ADMM), an optimization algorithm is designed for solving the manifold regularized sparse additive problem at the lower level. For simplicity, merely the regression task with squared loss is present here.

Here we generate the Gram matrix over labeled and unlabeled points $\mathbf{K} = \left(\mathbf{K}^{(1)}, \ldots, \mathbf{K}^{(p)}\right) \in \mathbb{R}^{(l+u) \times (l+u)p}$ with masked input $\boldsymbol{m} \odot x_i$ where $i \in [1, 2, \cdots, l + u]$, the model coefficient $\boldsymbol{\alpha} = \left(\alpha^{(1)^T}, \ldots \alpha^{(p)^T}\right)^T \in \mathbb{R}^{(l+u)p}$, and the label vector $Y = (y_1, \ldots, y_l, 0, \ldots, 0)^T \in \mathbb{R}^{l+u}$. Then, the lower-level problem can be reformulated as

$$\boldsymbol{\alpha}^* = \arg \min_{\boldsymbol{\alpha} \in \mathbb{R}^{(l+u)p}} \frac{1}{l} (Y - J\mathbf{K}\boldsymbol{\alpha})^T (Y - J\mathbf{K}\boldsymbol{\alpha}) + \lambda_1 \sum_{j=1}^{p} \tau_j \left\| \boldsymbol{\alpha}^{(j)} \right\|_2 + \frac{\lambda_2}{(l+u)^2} \boldsymbol{\alpha}^T \mathbf{K} L \mathbf{K} \boldsymbol{\alpha}, \quad (33)$$

where the matrix $J = \text{diag}(1, \ldots, 1, 0, \ldots, 0)$ is an $(l+u) \times (l+u)$ diagonal matrix with the first $l$ diagonal entries as 1 and the rest as 0 (Belkin et al., 2006).

By introducing the auxiliary variable $\vartheta = \left( \vartheta^{(1)^T}, \ldots, \vartheta^{(p)^T} \right)^T \in \mathbb{R}^{(l+u)p}, \vartheta^{(j)} = \left( \vartheta_1^{(j)}, \ldots, \vartheta_{l+u}^{(j)} \right) \in \mathbb{R}^{l+u}$, equation 33 can be rewritten as:

$$\min_{\alpha, \vartheta} \frac{1}{l}(Y - J\mathbf{K}\alpha)^T (Y - J\mathbf{K}\alpha) + \lambda_1 \sum_{j=1}^{p} \tau_j \left\| \vartheta^{(j)} \right\|_2 + \frac{\lambda_2}{(l+u)^2} \alpha^T \mathbf{K}L\mathbf{K}\alpha, \quad \text{s.t.} \quad \alpha - \vartheta = 0. \quad (34)$$

Hence, by introducing the auxiliary variable $\vartheta \in \mathbb{R}^{(l+u)p}$ and the Lagrange multiplier $\Lambda \in \mathbb{R}^{(l+u)p}$, the scaled augmented Lagrangian function of the primal problem equation 33 is

$$L(\alpha, \vartheta, \Lambda) = \frac{1}{l}(Y - J\mathbf{K}\alpha)^T (Y - J\mathbf{K}\alpha) + \lambda_1 \sum_{j=1}^{p} \tau_j \left\| \vartheta^{(j)} \right\|_2$$
$$+ \frac{\lambda_2}{(l+u)^2} \alpha^T \mathbf{K}L\mathbf{K}\alpha + \frac{\varrho}{2} \|\alpha - \vartheta - \Lambda\|_2^2 - \frac{\varrho}{2} \|\Lambda\|_2^2, \quad (35)$$

where $\varrho > 0$ is a positive penalty coefficient.

Given initialized parameters $(\alpha^0, \vartheta^0, \Lambda^0)$ and convergence criterion $\epsilon$, the manifold regularized additive regression problem with squared loss can be solved by the following iterative steps:

(1) Fix $\vartheta^t$ and $\Lambda^t$, and update the model coefficient $\alpha^{t+1}$:

$$\alpha^{t+1} = \arg\min_{\alpha} \frac{1}{l}(Y - J\mathbf{K}\alpha)^T (Y - J\mathbf{K}\alpha) + \frac{\lambda_2}{(l+u)^2} \alpha^T \mathbf{K}L\mathbf{K}\alpha + \frac{\varrho}{2} \|\alpha - \vartheta^t - \Lambda^t\|_2^2.$$

$\alpha^{t+1}$ can be calculated by the derivative of the objective function, which vanishes at the minimizer:

$$\frac{1}{l}(Y - J\mathbf{K}\alpha)^T (-J\mathbf{K}) + \left( \frac{\lambda_2}{(l+u)^2} \mathbf{K}L\mathbf{K} + \varrho(\alpha - \vartheta^t - \Lambda^t)^T \right) \alpha = 0.$$

(2) Fix $\alpha^{t+1}$ and $\Lambda^t$, and update the auxiliary variable $\vartheta^{t+1}$:

$$\vartheta^{t+1} = \arg\min_{\vartheta} \frac{1}{2} \|\alpha^{t+1} - \vartheta + \Lambda^t\|_2^2 + \frac{\lambda_1}{\varrho} \sum_{j=1}^{p} \tau_j \left\| \vartheta^{(j)} \right\|_2. \quad (36)$$

With fixed $\alpha^{t+1}$ and $\Lambda^t$, equation 36 is equivalent to the following $p$ subproblems:

$$(\vartheta^{(j)})^{t+1} = \arg\min_{\vartheta^{(j)}} \frac{1}{2} \left\| (\alpha^{(j)})^{t+1} - \vartheta^{(j)} + (\Lambda^{(j)})^t \right\|_2^2 + \frac{\lambda_1 \tau_j}{\varrho} \left\| \vartheta^{(j)} \right\|_2.$$

Thanks to the soft thresholding operators (Boyd et al., 2011; Chen et al., 2020), we have

$$(\vartheta^{(j)})^{t+1} = S_{\lambda_1 \tau_j / \varrho} \left( (\alpha^{(j)})^{t+1} + (\Lambda^{(j)})^t \right), \quad j = 1, \ldots, p,$$

where the soft thresholding operator $S$ stands for

$$S_k(a) = (a - k/\|a\|_2)_+ a.$$

(3) Fix $\alpha^{t+1}$ and $\vartheta^{t+1}$, and update the Lagrange multiplier $\Lambda^{t+1}$:

$$\Lambda^{t+1} = \Lambda^t + \alpha^{t+1} - \vartheta^{t+1}.$$

Denote the objective function of lower level problem as $\mathcal{R}(\alpha)$ (standing for $\mathcal{R}(\boldsymbol{\alpha}; \boldsymbol{m}; \boldsymbol{L})$) parameterized by model coefficient $\alpha$ (and mask $\boldsymbol{m}$ learned by upper level problem, the Laplacian matrix $\boldsymbol{L}$). The above three iterative steps form a loop until the following convergence conditions are met at $(t + 1)$-th iteration:

$$|\mathcal{R}(\alpha^{t+1}) - \mathcal{R}(\alpha^t)| \leq \epsilon. \quad (37)$$

Then the updating process stops and the output $\alpha^{t+1}$ can be considered as the desired model coefficient. Moreover, inspired by (Chen et al., 2020; Yuan et al., 2023), the early-stop condition in equation 37 could also be set as

$$\|\alpha^{t+1} - \alpha^t\|_\infty \leq \epsilon \quad \text{and} \quad \|\alpha^{t+1} - \vartheta^{t+1}\|_\infty \leq \epsilon.$$

### E.5 COMPUTATION COMPLEXITY ANALYSIS

With the $\epsilon$-stationary point defined in (Ji et al., 2021; Chu et al., 2024; Zhang et al., 2024), we conclude that the optimization for the upper problem requires at most $T = \mathcal{O}(\epsilon_1^{-2})$ iterations before reaching $\epsilon_1$-stationary based on Theorem 1. The lower level requires $\mathcal{O}(K(l + u))$ steps on gradient computations and $\mathcal{O}(p(l + u))$ assigning masks per outer iteration. Notice that K is the inner iteration and $p$ is the input dimension. The lower problem optimized by ADMM (Culp, 2011; Culp & Michailidis, 2008) enjoys the sublinear convergence rate $\mathcal{O}(1/K)$ w.r.t. Nash Point with threshold $1/K \lesssim \epsilon_2$ when the lower problem satisfies the convexity condition. Please refer to (Wang & Zhao, 2022) for the corresponding proof of general ADMM optimization.

In summary, the computation complexity of S$^2$MAM reaches $\mathcal{O}\left(\frac{p(l+u)}{\epsilon_1^2 \epsilon_2}\right)$, which is competitive with some latest bilevel algorithms(Liu et al., 2022; Xiao et al., 2023). Empirically, please refer to *Appendix B.4* for convergence analysis and *Appendix B.7* for some experimental comparisons on training time cost.

## F LIMITATIONS AND DISCUSSIONS

This paper proposes a new bilevel manifold regularization for semi-supervised learning tasks with an automatic feature masking mechanism. Theoretically, we establish its foundations of learning theory including the computing convergence and the generalization error analysis. As far as we know, this is the first work for bounding the excess risk of semi-supervised additive model. And our results show better convergence performance than (Zhou et al., 2022). Empirically, we verify the effectiveness of the proposed approach on synthetic datasets and real-world datasets. We designed the novel optimization algorithm for the proposed manifold regularized sparse additive model (see Appendix E.4). In the implemented codes, we further provide the settings of spline-based additive models.

However, there still exist some limitations including the computational difficulties on large-scale datasets and the assumption of bounded output. Fortunately, as introduced in Appendix B.7, S$^2$MAM can also deal with high-dimensional data with data preprocessing. An interesting approach for dealing with high-dimensional data like images is to extract the feature vectors first, which has been widely employed in some supervised (Su et al., 2023) and semi-supervised works (Qiu et al., 2018; Nie et al., 2019; Kang et al., 2020; Nie et al., 2021). And the random Fourier technique (Rahimi & Recht, 2007; Wang et al., 2023) could be further considered to accelerate the computation process. Theoretically, the bounded condition of the response can be relaxed to include the unbounded output, e.g., replacing it by the $1 + \epsilon$ moment bounded assumptions (Feng, 2021; Feng & Wu, 2022)). The neural additive modeling strategy (Agarwal et al., 2021; Yang et al., 2020) is also another interesting and effective direction to better improve the non-linear approximation ability and prediction performance of S$^2$MAM. In addition, the current generalization analysis just focuses on the basic model of S$^2$MAM, which can be further improved to match the bilevel manifold regularization tightly.

