# OpenReview forum: "S$^2$MAM: Semi-supervised Meta Additive Model for Robust Estimation and Variable Selection"
_ICLR.cc/2025/Conference — Submitted to ICLR 2025_

### Official Review · Reviewer_5g1j · 2024-10-21

**Soundness:** 2
**Presentation:** 2
**Contribution:** 2
**Rating:** 5
**Confidence:** 2

**Summary:**

This paper proposes a new semi-supervised meta additive model (S2MAM) under a bilevel optimization scheme to automatically identify the informative variables, update the similarity matrix, and achieve the interpretable prediction simultaneously. Theoretical guarantees are provided for S2MAM including the computing convergence and the statistical generalization bound. Experimental assessments on synthetic and real-world datasets validate the robustness and interpretability of the proposed approach.

**Strengths:**

This paper inject the meta learning strategy and sparse additive models into manifold regularized SSL framework, and formulate a new semi-supervised meta additive model (S2MAM) to realize automatic variable masking and sparse approximation for high-dimensional inputs even with noisy variables.

There are no technical errors, and the presentation and writing are clear.

An efficient implementation is employed here to solve the bilevel optimization problem, which avoids the heavy computing burden on the implicit hypergradient calculation.

**Weaknesses:**

I am absolutely not in this field and the comments from me are not relatively professional. The comments in the following are just raised from the presentation or organization.

The authors propose the manifold regularized semi-supervised additive model. However, the authors do not analyze the important part "manifold regularized" in deepth for the presentation of the whole paper, making the paper inevitably lacks the novelty, i.e., the formulation of the manifold regularized sparse additive model is built by the combination of the existing works and the rationality behind this combination and the related theoretical analysis are not clearly given.

I wonder why the authors give many parts in presenting the semi-supervised additive model, i.e., Manifold Regularized Sparse Additive Model, Discrete Bilevel Framework for S2MAM, and Probabilistic Bilevel Framework for S2MAM. These parts seem to be all needed to be presented. However, the limited space in the paper seem to be the combination of these parts and the novelty of each part is dispersed to some degree. The definition of upper level and the lower level of discrete bilevel framework lack clear definition and the realted analysis in Eq. (3) and Eq. (4). The rationality of just using Bernoulli random variable to denote mi should be given, i.e., why it it simple and wheter the other types of random variables can be better in probalisitic bilevel framework.

The authors should add more recent methods for comparison in Table 3 and Table 5 to better show the effectiveness of the proposed method.

The authors just perform evaluations of S2MAM on eight real-world datasets from UCI repository. I think the authors can add other datasets for comparison in the experiment, which is not limited to UCI repository.

**Questions:**

1. Can the authors analysis the extreme cases when the fixed label percentages are high, i.e., r=50% and the noisy variables are high in the experiment, i.e., 100 noisy variables.

2. The authors can further analyze why the average accuracy on Breast Cancer achieve such high performance on Breast Cancer, i.e., the proposed method on unlabeled case is about 88 and the second best performance is just 77.197.

**Details Of Ethics Concerns:**

I have no details of ethics concerns for this work.

---

> ### Author Response · Authors · 2024-11-21
> **Responses to Weakness 1,2,3**
>
> Thanks for your meaningful comments and suggestions. We believe the quality of the paper has been improved following your suggestions.
>
> ### **To Weakness 1**
>
> **However, the authors do not analyze the important part "manifold regularized" in deepth for the presentation of the whole paper, making the paper inevitably lacks the novelty, i.e., the formulation of the manifold regularized sparse additive model is built by the combination of the existing works and the rationality behind this combination and the related theoretical analysis are not clearly given.**
>
>
> As stated in the Abstract and Introduction, this paper focuses on Laplace-Beltrami operator-based manifold regularization [1-3]. It is further stated in Line 225 (Page 5), where $\mathbf{f}^{T} \boldsymbol{L} \mathbf{f}$ in Eq.(2) stands for the manifold regularization.
>
> **Besides, "Appendix B.3 Ablation Analysis" has empirically analyzed the roles of:**
> **1) manifold regularization for utilizing unlabeled samples,**
>
> **2) additive modeling for nonlinear approximation,**
>
> **3) and the bilevel scheme for feature selection.**
>
> Corresponding discussions are also summarized in "Appendix B.3".
>
> > [1] Semi-supervised learning literature survey, 2006
> >
> > [2] Semi-Supervised Learning With GANs: Revisiting Manifold Regularization, ICLR 2018
> >
> > [3] Federated Learning with Manifold Regularization and Normalized Update Reaggregation, NeurIPS 2023
>
>
>
> ### **To Weakness 2**
>
> **I wonder why the authors give many parts in presenting the semi-supervised additive model, i.e., Manifold Regularized Sparse Additive Model, Discrete Bilevel Framework for S2MAM, and Probabilistic Bilevel Framework for S2MAM. These parts seem to be all needed to be presented.**
>
> Thanks for your comments. And we've **renamed Section 2.1 as "Revisiting Manifold Regularized Sparse Additive Model" to enhance readability**.
>
> Indeed, **the connections between these parts have been illustrated** as follows:
>
> **1) In Section 2, descriptions in Line 175 (Page 4) and Line 269 (Page 5) involve:**
>
> ​	**(1)** The first model is our basic one, whose theoretical generalization behavior is firstly analyzed with the polynomial decay rate in this paper.
>
> ​	**(2)** The second discrete bilevel model is formulated by equipping the basic model with the 0/1 mask learning technique at the upper level, to remove noisy features.
>
> ​	**(3)** Due to the computing difficulty of 0/1 problem of the second model, the third probabilistic model is proposed with accelerated and efficient optimization.
>
> **2) In Appendix B.3 (Page 20), Figure 2 visualizes their connections.**
>
> **3) In Appendix B.3 Ablation Analysis (Table 7 in Page 20):**
>
> These ablation experiments validate the contributions of each module of Probabilistic S2MAM as follows:
>
> ​	**(1)** the **manifold regularization** ($\mathbf{f}^{T} \boldsymbol{L} \mathbf{f}$) helps to **use the unlabeled samples** to learn better prediction functions;
>
> ​	**(2)** the **additive modeling** strategy can improve the **non-linear approximation** ability.
>
> ​	**(3)** the employed **probabilistic bilevel scheme** for automatically assigning variable masks is vital to **deal with noisy variables**.
>
>
>
> ### **To Weakness 3**
>
> **The definition of upper level and the lower level of discrete bilevel framework lack clear definition and the realted analysis in Eq. (3) and Eq. (4).**
>
> The two objective functions are presented with detailed illustrations and mathematical definitions.
> As present **in Line 250 to 252 (Page 5),** we note that **the upper level is designed to learn the mask variable m, and the lower level learns the prediction function with parameter alpha**. Besides, the lower level in Eq.(4) is the manifold regularized sparse additive model, which enjoys similar representations in Eq.(2).
>
> **Secondly, all these relevant notations and symbols are defined in detail.** Except for the newly introduced mask, upper risk L, and lower risk R, the rest of the notations come from the basic model of manifold regularized model in Section 2.1.
>
> **Ablation Analysis in Appendix B.3** empirically validate the connections among the upper problem (3) for assigning mask and the additive strategy in the lower problem (3) for nonlinear fitting and interpretability (See Figure 9 in Page 25).
>
> Thus, we think the definitions for both levels of the discrete model are clear with sufficient explanations, which shows how we equip the basic model in Eq.(2) with a bilevel learning framework.

---

> ### Author Response · Authors · 2024-11-21
> **Responses to Weakness 4,5,6 and Questions**
>
> ### **To Weakness 4**
>
> **The rationality of just using Bernoulli random variable to denote mi should be given, i.e., why it it simple and whether the other types of random variables can be better in probabilistic bilevel framework.**
>
> **Bernoulli distribution is a simple and natural choice** for a binary outcome with a probability value, with simple theoretical gradient derivation and empirical validations. Notably, **it is commonly used** to assign random masks on MAE or when pre-training BERT or transformers [1], improving the robustness of Diffusion models [2], finetuning Dropout layer and Bayesian deep modeling [3] for semantic segmentation and image classification.
>
> **Some other commonly used distributions**, like geometric distribution, Poisson distribution, negative binomial distribution, Laplace distribution and uniform distribution, **are not applicable**. The logistic distribution takes complex forms, which fails to simplify gradient computation.
>
> Thanks again for your constructive comment, and we've added relevant comments in Appendix B***.
>
> > [1] Compression method and platform of pre-training language model based on knowledge distillation
> >
> > [2] Binary Noise for Binary Tasks: Masked Bernoulli Diffusion for Unsupervised Anomaly Detection
> >
> > [3] Learnable Bernoulli Dropout for Bayesian Deep Learning
>
>
>
> ### **To Weakness 5**
>
> **The authors should add more recent methods for comparison in Table 3 and Table 5 to better show the effectiveness of the proposed method.**
>
> Thanks for your meaningful suggestions.
>
> Please refer to Common Concerns (3) and resubmitted pdf for new competitors [1,2].
>
> New results on more baselines are reported in the revised paper, highlighted in red color.
>
> > [1] Consistent feature selection for analytic deep neural networks. NeurIPS 2020.
> >
> > [2] NP-Match: When Neural Processes meet Semi-Supervised Learning. ICML 2022.
>
>
>
> ### **To Weakness 6**
>
> **The authors just perform evaluations of S2MAM on eight real-world datasets from UCI repository. I think the authors can add other datasets for comparison in the experiment, which is not limited to UCI repository.**
>
> **Indeed, this work has used a total of 4 synthetic datasets, 8 UCI datasets, and one high-dimensional image data (COIL).**
> Please refer to Common Concerns (2) and Appendix B.1 for data descriptions.
>
>
>
> ### **To Question 1**
>
> **Can the authors analyze the extreme cases when the fixed label percentages are high, i.e., r=50\% and the noisy variables are high in the experiment, i.e., 100 noisy variables?**
>
> Thanks for your constructive suggestions on a high labeled percentage.
> **The results under the settings that you've mentioned have already been presented in Table 10 Page 24.** The empirical analysis in Line 1400 includes:
>
> ​	**(1)** At a 5\% labeling rate, S2MAM is capable of assigning suitable masks, effectively utilizing the input from 95\% unlabeled data to boost the model's predictive accuracy.
>
> ​	**(2)** **At a 50\% labeling rate**, these supervised baselines usually maintain better sparse regression estimators than S2MAM. The empirical observations are natural since the labeled data under this setting is often enough to find the predictor, and **supervised sparse methods should be suggested.**
>
>
>
> ### **To Question 2**
>
> **The authors can further analyze why the average accuracy on Breast Cancer achieves such high performance on Breast Cancer, i.e., the proposed method on unlabeled cases is about 88 and the second-best performance is just 77.197.**
>
> Thanks for your detailed review of our empirical results in the Breast Cancer dataset.
> Four potential reasons for the superior performance of S2MAM over the second-best AWSSL include:
>
> **(1) Feature selection policy**
>
> AWSSL: continuous weighing on all features, where some noisy dimensions may still cause impacts
>
> S2MAM: validation set-induced auto 0/1 masks on all features
>
> **(2) Mechanism for assigning labels**
>
> AWSSL: similarity-based label propagation
>
> S2MAM: representation of additive kernels, usually with stronger approximation capability
>
> **(3) Algorithms**
>
> AWSSL: iterative optimization for weight updation, may not meet global convergence as bilevel optimization does [1]
>
> S2MAM: optimized by bilevel technique with convergence guarantee in Theorem 1
>
> **(4) Parameters**
>
> AWSSL is sensitive to the specified neighbor numbers, where S2MAM can tolerate a slightly larger number of selected features than the actual ones.
>
>
>
> > [1] A review on bilevel optimization: from classical to evolutionary approaches and applications

---

### Official Review · Reviewer_iyKb · 2024-10-25

**Soundness:** 3
**Presentation:** 2
**Contribution:** 2
**Rating:** 3
**Confidence:** 4

**Summary:**

This paper presents a model named S2MAM, which aims to improve classical graph-based semi-supervised learning (GSSL) by combining bilevel optimization with manifold regularization to select relevant features and adapt similarity matrices in noisy and redundant data settings. S2MAM addresses issues in traditional manifold regularization, where fixed similarity metrics can perform poorly with uninformative features. The authors propose a probabilistic bilevel optimization framework to apply adaptive variable masking and provide theoretical proofs of convergence and generalization bounds. Empirical evaluations on synthetic and small real-world data suggest that S2MAM performs well in both regression and classification tasks, particularly under noisy conditions.

**Strengths:**

* The method partially solves the issue of poor similarity matrix in the graph-based SSL methods
* The authors provided convergence and generalization bounds, which enhances the model's credibility
* The proposed method outperforms the other GSSL methods of the same kind

**Weaknesses:**

The method still has many limitations of the graph-based GSSL methods. Therefore the impact of the method is very limited.
* It is expensive. The W or L matrix is infeasible when l is large. The method involves multiple kernel matrices, which is even more expensive.
* The assumption of a single bandwidth mu is unrealistic
* The construction of W is problematic for many data types, e.g., images, text documents, and biology sequences.
* The experiments are not convincing because all tested datasets are very small.

**Questions:**

* The computational complexity is not ananlyzed.
* The prediction function requires all training samples. This is rather infeasible in practice. If changed to other prediction functions, does the theoretical guarantee still hold?

---

> ### Author Response · Authors · 2024-11-21
> **Responses to the Weakness**
>
> ### **To Weakness 1**
>
> **The method still has many limitations of the graph-based GSSL methods. Therefore the impact of the method is very limited. It is expensive. The W or L matrix is infeasible when l is large. The method involves multiple kernel matrices, which is even more expensive.**
>
> Thanks for your constructive comments and suggestions!
>
> **1) We agree with the limitations on computing with an extremely large labeled set.** Indeed, this work aims to learn from noisy data with few labeled points. As stated in the first sentence at the Introduction, this paper focuses on the scenario of "utilizing a large amount of unlabeled data with limited labeled data jointly".
>
> **2) Moreover, it has been discussed and empirically validated in Table 10** under the settings of 5\% or 50\% labeled points, which are further summarized in Line 1294 (Page 24) as follows.
>
> The empirical results in Table 10 show that:
>
> ​	(1) At a 5\% labeling rate, S2MAM is capable of assigning suitable masks, effectively utilizing the input from 95\% unlabeled data to boost the model's predictive accuracy.
>
> ​	(2) **At a 50\% labeling rate**, these supervised baselines usually maintain better sparse regression estimators. Such observations are natural since the labeled data is often enough to find the supervised predictor, and **supervised sparse methods should be suggested under this setting**.
>
> **3) Besides, experiments on both Buzz-regression and Buzz-classification datasets containing $N=38393$ samples** also show that S2MAM is capable of dealing with such scenarios, where probabilistic modeling and projected gradient estimation are adopted for acceleration.
>
>
>
> ### **To Weakness 2**
>
> **The assumption of a single bandwidth mu is unrealistic.**
>
>
> We agree that using a single bandwidth parameter mu may not fully capture the complexity of all data.
>
> But after considering the following points, we observe this is a reasonable setting:
>
> **1) Noise data limitations:** Although some adaptive bandwidth methods (e.g., K-means) can adapt to complex data distributions, noisy dimensions considered in this work lead to the \textbf{failure of adaptive bandwidth estimation} based on density or distance measures.
>
> **2) Parameter selection flexibility:** In practice, the bandwidth parameter mu is selected through cross-validation to accommodate different data distributions and task requirements.
>
> **3) Theoretical guarantee:** In kernel density estimation, a Gaussian kernel with a single bandwidth parameter mu can provide a smooth and continuous density estimation [1,2,3], which is crucial for the construction of the similarity matrix.
>
> **4) Practical effectiveness:** Similar setting with a single bandwidth has been proven to be effective in a variety of tasks, including similarity measurements in image processing [4].
>
> Thanks again for this meaningful comment. We have added the following comments on Page 20 Remark 6:
>
> After filtering out effective features using S2MAM, the extracted data can be applied to downstream tasks under an adaptive bandwidth strategy, which can adapt to complex data distributions like imbalanced categories.
>
> > [1] On estimation of a probability density function and mode. Annals of Mathematical Statistics 1962.
> >
> > [2] Uniform convergence rates for kernel density estimation. ICML 2017.
> >
> > [3] On the consistency of quick shift. NeurIPS 2017.
> >
> > [4] Robust embedding regression for semi-supervised learning. Pattern Recognition 2024.
>
>
>
> ### **To Weakness 3**
>
> **The construction of W is problematic for many data types, e.g., images, text documents, and biology sequences.**
>
> **Please refer to Common Concerns (2) for non-numeric applications.**
> The applications to other types of data, e.g., images and texts, have been realized in [1,2,3,4]. Their data processing strategies are further adopted in this paper for image classification, see "Appendix B.7 Extension to Image Data".
>
> Moreover, for some large datasets that may contain contextual or sequential semantic information, e.g., text documents and biology sequences, we recommend using supervised technologies like transformers to collect deeper information.
>
>
>
> ### **To Weakness 4**
>
> **The experiments are not convincing because all tested datasets are very small.**
>
> Indeed, **this work has used a total of 4 synthetic datasets, 8 UCI datasets, and one high-dimensional image data (COIL)**.
>
> Please refer to Common Concerns (2) and Appendix B.1 for detailed descriptions of used datasets, e.g.,
>
> 1) **Large Data Sets**: Buzz-Regression and Buzz-Classification have 38,393 samples, each with 77 features.
>
> 2) **High-Dimensional Data**: An image from COIL20 originally has 16,384 features (128*128 grayscale).

---

> ### Author Response · Authors · 2024-11-21
> **Responses to the Questions**
>
> ### **To Question 1**
>
> **The computational complexity is not analyzed.**
>
> Please refer to Common Concerns (4).
>
> Indeed, **as stated in Remark 4 (Page 7, Line 359), the computation complexity has been theoretically and empirically analyzed.**
>
> **Theoretically,** the computational complexity has been derived in "Appendix E.5" (Page 45) with the order of $\mathcal{O}\left(\frac{p(l+u)}{\epsilon_1^2 \epsilon_2} \right)$.
>
> **Empirically,** please refer to \emph{Appendix B.7} for some experimental comparisons and discussions on training time cost.
>
>
>
> ### **To Question 2**
>
> **The prediction function requires all training samples. This is rather infeasible in practice. If changed to other prediction functions, does the theoretical guarantee still hold?**
>
> **1) The predictive function based on additive kernel representation**
> $$
> f=\sum_{j=1}^p \sum_{i=1}^l\alpha_i^{(j)}{K}_i^{(j)}(x_i^{(j)},\cdot),
> $$
> **requires the labeled samples, instead of all training samples.** Thus the computational cost involved is empirically limited. Such representation based on reproducing kernel Hilbert space enjoys solid theoretical guarantees [1,2]. And our additive modeling helps to alleviate the curse of dimensionality [3].
>
>
>
> In response to Weakness 1, this work aims to learn from noisy data with few labeled points. As stated in the first sentence of the Introduction, this paper focuses on the scenario of "utilizing a large amount of unlabeled data with limited labeled data jointly".
> Experiments on two large-size Buzz datasets involving 38393 samples also validate the practical effectiveness (See Tables 4,5,8,9).
>
>
>
> **2) Usually, the additive models could be implemented based on the kernels[1,2], smooth splines [4,5] and neural networks [6,7].**
>
> When replacing the kernel strategy with the smooth splines or feature-wise MLP [6,7,8]:
>
> **(1) Theorem 1 still holds**, as relevant assumptions do not depend on the kernels.
>
> **(2) Theorem 2 requires specific assumptions on splines or networks.** The current results are obtained based on the commonly used assumptions about the kernels (Assumptions 2 \& 4). By replacing with specific assumptions as in spline-based [4,5] or neural [8] additive research, the generalization guarantees similar to those in Theorem 2 may still hold, which is another interesting learning topic.
>
> To the best of our knowledge, Theorem 2 is the first attempt to characterize the generalization performance of semi-supervised additive models.
>
>
>
> > [1] High-dimensional additive modeling. The Annals of Statistics 2009.
> >
> > [2] SpAM: sparse additive models. Journal of the Royal Statistical Society: Series B 2009.
> >
> > [3] Additive approximations in high dimensional nonparametric regression via the SALSA. ICML 2016.
> >
> > [4] Huber Additive Models for Non-stationary Time Series Analysis. ICLR 2022.
> >
> > [5] Minimax-optimal rates for sparse additive models over kernel classes via convex programming. Journal of Machine Learning Research 2012.
> >
> > [6] Neural additive models: Interpretable machine learning with neural nets. NeurIPS 2021.
> >
> > [7] CAT: Interpretable Concept-based Taylor Additive Models. KDD 2024 Best Paper Award.
> >
> > [8] Neural Partially Linear Additive Model. Frontiers of Computer Science 2023.

---

### Official Review · Reviewer_HK4x · 2024-10-27

**Soundness:** 3
**Presentation:** 3
**Contribution:** 3
**Rating:** 6
**Confidence:** 5

**Summary:**

This paper proposes a novel semi-supervised meta additive model where a bilevel optimization scheme is employed to identify the informative features. The authors analyze the properties of the model including the convergence and statistical generalization   bound. Experimental results on toy and real-world data sets demonstrate promising improvements over previous methods.

**Strengths:**

1. The paper is well-organized.
2. This paper proposes a bilevel optimization scheme to select informative features.
3. The authors provided theoretical guarantees for the proposed model.

**Weaknesses:**

1. The authors employed a sampling strategy from Bernoulli distributions to select informative features. Do selected features contain noisy features by using this strategy? If this occurs, how to remove noisy features?
2. Hyperparameter C affects the selection of  informative features. It would be much better to discuss the effect of this hyperparameter or give the specific values in the experiments.
3. The authors used the batch in each iteration. In practical implementations,  what is the proportion of the labeled samples in the batch since the unlabeled samples dominate the training set?
4. The authors assume  that $\Phi(s)$ is L smooth. For the model in this paper,  how to compute this constant L?
5.  The authors employed dimension reduction techniques to reduce the dimensions of high-dimensional data.  Is the method effective for directly selecting features of high-dimensional data such as gene expression data?

**Questions:**

Please refer to the weakness.

---

> ### Author Response · Authors · 2024-11-21
> **Responses to Weakness 1,2,3,5**
>
> ### **To Weakness 1**
>
> **The authors employed a sampling strategy from Bernoulli distributions to select informative features. Do selected features contain noisy features by using this strategy? If this occurs, how to remove noisy features?**
>
>
> Thanks for your constructive comments.
> The Bernoulli distribution is used for probabilistic modeling, where its probability is considered as the learning target to select the informative features.
>
> Theorem 1 states the convergence of the upper problem, which aims to update the proper mask (probability) m (s) under the constraint on coresize C. As illustrated in Remark 6 (Page 20), setting too large C may select noisy features.
>
> Such failure cases are empirically addressed by the grid search on selecting C. And S2MAM is still capable of identifying the informative features with C slightly larger than the ground truth.
>
>
>
> ### **To Weakness 2**
>
> **Hyperparameter C affects the selection of informative features. It would be much better to discuss the effect of this hyperparameter or give the specific values in the experiments.**
>
>
> Following your meaningful suggestions, we've added a section "Appendix B.4.3 Impact of Selected Core Size C", for sensitive analysis and empirical selection strategy on C.
>
> The visualized curves on varying C are plotted in Figure 7 (Page 22). The results validate our findings that S2MAM is capable of removing irrelevant features even with a slightly larger C than the ground truth.
>
>
>
> ### **To Weakness 3**
>
> **what is the proportion of the labeled samples in the batch?**
>
> Thanks for pointing out the issues on batch settings. To enhance the readability, we’ve added the following content on "Appendix B.2.1", Page 18, Line 968:
>
> In practice, the proportion of labeled points in a single batch is consistent with the settings in the whole training set to avoid empty labeled sets or inconsistency among each batch.
>
> ### **To Weakness 5**
>
> **The authors employed dimension reduction techniques to reduce the dimensions of high-dimensional data. Is the method effective for directly selecting features of high-dimensional data such as gene expression data?**
>
> The dimension reduction technique is commonly adopted for semi-supervised learning [1,2], which has been empirically validated to be efficient in reducing the computational cost of complex data with extremely high dimensions like images. For fairness, the extracted data is empirically shared for all baselines on high-dimensional scenarios, like COIL-20 images.
>
> It may not be applicable to directly operate on extremely high-dimensional data for S2MAM and some other feature selection approaches [1,2,3,4]. Please refer to Common Concerns (1) and (3) for the analysis of scalability and computation costs.
>
> Besides, for some large datasets that may contain contextual or sequential semantic information, such as text (genes), audio, and video, we recommend using technologies like supervised transformers to mine deeper information. Additionally, the masked autoencoder [5] based on random masks on high-dimensional texts, has also been widely verified to be effective for similar problems you've raised.
>
> > [1] Adaptive local embedding learning for semi-supervised dimensionality reduction. IEEE TKDE 2021.
> >
> > [2] Robust embedding regression for semi-supervised learning. Pattern Recognition 2024.
> >
> > [3] Semi-supervised learning with auto-weighting feature and adaptive graph. IEEE TKDE 2019.
> >
> > [4] Robust graph learning from noisy data. IEEE Transactions on Cybernetics 2020.
> >
> > [5] Masked autoencoders are scalable vision learners. CVPR 2022.

---

> ### Author Response · Authors · 2024-11-21
> **Responses to Weakness 4**
>
> ### **To Weakness 4**
>
> **The authors assume that $\phi$(s) is L smooth. For the model in this paper, how to compute this constant L?**
>
> The smooth assumption on function $\phi$(s) for convergence analysis is originally adopted from [1].
>
> Indeed, the probabilistic formulation brings challenges to determine the smoothness constant L w.r.t. probability variable s. The following content summarizes the informal proof.
>
> **Proof:**
>
> Assume that the prediction w.r.t. j-th dimension
>
> $$
> f^{(j)}(\cdot; \alpha)=\sum_{i=1}^l \alpha_i^{(j)}{K}_i^{(j)}(x_i^{(j)},\cdot)
> $$
>
> is bounded by $\|f^{(j)}\|_2 \leq a$. Notably, similar bounded assumptions are commonly required for generalization analysis [2,3,4].
>
> Given independent variable $\boldsymbol{m} = [m_1, m_2 , ..., m_p]$ following Bernoulli distribution with probability of $\boldsymbol{s} = [s_1,...,s_p]$ taking **1**,
>
> $$
> \Phi(\boldsymbol{s})=\mathbb{E}_{p(\boldsymbol{m} \mid \boldsymbol{s})} \mathcal{L}\left(\boldsymbol{\alpha}^*(\boldsymbol{m})\right)
> $$
>
> $$
> \mathcal{L}\left(\boldsymbol{\alpha}^*(\boldsymbol{m})\right) =\frac{1}{l} \| y - f(x ; \boldsymbol{\alpha}^*(\boldsymbol{m})) \|_2^2,
> $$
>
> and
>
> $$
> f(x ; \boldsymbol{\alpha}^*(\boldsymbol{m})) = \sum_{j=1}^p m_j \sum_{i=1}^l \alpha_i^{(j)}{K}_i^{(j)}(x_i^{(j)}, x^{(j)}).
> $$
>
>
> With
>
> $$
> \mathbb{E}_{p(\boldsymbol{m}\mid \boldsymbol{s})} \left[ \frac{\partial m_j}{\partial s_j} \right]  = 1
> $$
>
> and
>
> $$
> \frac{\partial f(x ; \boldsymbol{\alpha}^*(\boldsymbol{m})) }{\partial s} = \left[ \sum_{i=1}^l \alpha_i^{(1)} K_i^{(1)}(x_i^{(1)}, x^{(1)}), ..., \sum_{i=1}^l \alpha_i^{(p)} K_i^{(p)}(x_i^{(p)}, x^{(p)})  \right]
> $$
>
> we can derive that
>
> $$
> \nabla_{\boldsymbol{s}} \Phi(\boldsymbol{s})
> =\frac{2}{l} \mathbb{E}_{p(\boldsymbol{m}\mid \boldsymbol{s})} \left[ (f(x ; \boldsymbol{\alpha}^*(\boldsymbol{m}))-y) \frac{\partial f(x ; \boldsymbol{\alpha}^*(\boldsymbol{m})) }{\partial s}\right],
> $$
>
> and
>
> $$
> \nabla_{\boldsymbol{s}}^2 \Phi(\boldsymbol{s})
> = \frac{2}{l} \mathbb{E}_{p(\boldsymbol{m}\mid \boldsymbol{s})} \left[ (\frac{\partial f(x ; \boldsymbol{\alpha}^*(\boldsymbol{m})) }{\partial s})^T \cdot (\frac{\partial f(x ; \boldsymbol{\alpha}^*(\boldsymbol{m})) }{\partial s}) + (f(x ; \boldsymbol{\alpha}^*(\boldsymbol{m}))-y) \frac{\partial^2 f(x ; \boldsymbol{\alpha}^*(\boldsymbol{m})) }{\partial s^2} \right].
> $$
>
>
>
>
> That is,
> $$
> \|\nabla_{\boldsymbol{s}}^2 \Phi(\boldsymbol{s}) \|_2 \leq \frac{2}{l} \| \frac{\partial f(x ; \boldsymbol{\alpha}^*(\boldsymbol{m})) }{\partial s}\|_F^2 \leq  2 p a^2/l \leq L.
> $$
>
>
> > [1] Probabilistic bilevel coreset selection. ICML 2022.
> >
> > [2] Learning theory estimates for coefficient-based regularized regression. Applied and Computational Harmonic Analysis 2013.
> >
> > [3] Learning rates for the risk of kernel-based quantile regression estimators in additive models. Analysis and Applications 2016.
> >
> > [4] Sparse kernel regression with coefficient-based L1 regularization. Journal of Machine Learning Research 2019.

---

### Official Review · Reviewer_GvdS · 2024-11-04

**Soundness:** 2
**Presentation:** 3
**Contribution:** 2
**Rating:** 5
**Confidence:** 5

**Summary:**

The paper introduces a new Semi-Supervised Meta Additive Model (S2MAM) that aims to improve the robustness and interpretability of semi-supervised learning through bilevel optimization. The proposed model combines manifold regularization, meta-learning, and sparse additive modeling to automatically identify informative features while simultaneously performing semi-supervised learning tasks. Unlike existing methods that rely on pre-specified similarity matrices, S2MAM leverages probabilistic bilevel optimization to adaptively update both the similarity matrix and decision functions. The authors provide theoretical guarantees regarding convergence and generalization, and they validate the proposed model using synthetic and real-world datasets, demonstrating its robustness and interpretability against noisy and redundant variables.

**Strengths:**

(1) The paper introduces an innovative bilevel optimization framework for manifold regularization that addresses key limitations of existing semi-supervised learning models, particularly those involving noisy and redundant input variables.
(2) The paper offers a rigorous theoretical foundation, including proofs for optimization convergence and bounds for generalization error. These theoretical results provide assurance about the robustness and applicability of S2MAM in different learning scenarios.
(3) Extensive experiments on both synthetic and real-world datasets show that S2MAM outperforms other state-of-the-art semi-supervised learning models, particularly in scenarios with noisy and redundant features. The robustness and interpretability claims are supported by competitive performance results.

**Weaknesses:**

(1) While the paper provides a novel framework, the bilevel optimization procedure can be computationally expensive, especially for large-scale datasets. Despite the probabilistic bilevel framework simplifying the optimization process, the overall complexity might still be a limitation in practical applications.
(2) The proposed method involves both Laplacian matrix computation and optimization over probabilistic masks, which may not scale efficiently to very high-dimensional datasets or very large sample sizes. Although theoretical guarantees are provided, the lack of a discussion on practical scalability leaves questions about how S2MAM would handle real-world big data scenarios.
(3) While the paper compares S2MAM against a wide variety of baselines, some comparisons (especially against modern deep learning-based semi-supervised models) are limited. This is particularly important since deep learning approaches have become dominant in semi-supervised learning tasks.

**Questions:**

(1) Could the authors clarify the learning dynamics of the probabilistic mask variables? Specifically, how does the model ensure convergence of these masks to truly informative features, and is there a risk of instability in mask selection during the optimization?
(2) How does S2MAM's computational complexity compare to models like LapSVM or AWSSL? Specifically, can the authors clarify its scalability for high-dimensional datasets and large sample sizes?
(3) The paper compares S2MAM with several established baselines, including LapSVM, AWSSL. However, the comparisons with recent deep learning-based semi-supervised learning methods are limited. Could the authors provide additional experimental comparisons with state-of-the-art deep learning-based SSL methods to better evaluate S2MAM's competitive performance?
(4) The datasets used in the experiments appear relatively small. Could the authors evaluate S2MAM on larger, more complex datasets to demonstrate its scalability and robustness in real-world scenarios?

---

> ### Author Response · Authors · 2024-11-21
> **Responses to the Weakness**
>
> Thanks for your selfless efforts on reviewing our work!
>
> ### **To Weakness 1 \& 2**
>
> **computationally expensive for large-scale datasets, like very high-dimensional datasets or very large sample sizes. lack of a discussion on practical scalability.**
>
> Thanks for your constructive comments! Please refer to Common Concerns (1) (2) for discussions on large-scale scenarios, and Common Concerns (4) for computation cost analysis.
>
> Yes, we agree that the proposed method may not perform efficiently when handling very large-scale datasets, which has been discussed in "Appendix B.7 Extension to Image Data". Naturally, the bilevel scheme for selecting informative features and the additive strategy for interpretable modeling require more computation costs.
> Following your suggestions, we've summarized the limitations and solutions on scalability in "Appendix F Limitations and Discussions".
>
> **Directly handling extremely large datasets is still a challenging task** for both traditional [1,2] and deep learning-based SSL approaches like VAE and SSDKL. And it's suggested to adopt some strategies like corset selection [3,4], dimensional reduction [5,6], random Fourier acceleration [7,8] and the low-rank representation technique [9]. **Following your suggestions, relevant discussions are added in \emph{Appendix F Limitations and Discussions}.**
>
> > [1] Joint Feature Selection and Extraction With Sparse Unsupervised Projection. TNNLS 2023.
> >
> > [2] Fast Unsupervised Projection for Large-Scale Data. TNNLS 2022.
> >
> > [3] Coresets via bilevel optimization for continual learning and streaming. NeurIPS 2020.
> >
> > [4] Probabilistic bilevel coreset selection. ICML 2022.
> >
> > [5] Adaptive local embedding learning for semi-supervised dimensionality reduction. IEEE TKDE 2021.
> >
> > [6] Robust embedding regression for semi-supervised learning. Pattern Recognition 2024.
> >
> > [7] Towards a unified analysis of random Fourier features. JMLR 2021.
> >
> > [8] Tilted Sparse Additive Models, ICML 2023.
> >
> > [9] Robust embedding regression for semi-supervised learning. Pattern Recognition 2024.
>
> ### **To Weakness 3**
>
> **While the paper compares S2MAM against a wide variety of baselines, some comparisons (especially against modern deep learning-based semi-supervised models) are limited.**
>
>
> Following your suggestions, we've further conducted experiments with both supervised [1] and semi-supervised [2] deep learning-based approaches.
>
> Please refer to the revised pdf (red content in tables) and Common Concerns (3) for new results and descriptions of these new competitors.
>
> > [1] Consistent feature selection for analytic deep neural networks. NeurIPS 2020.
> >
> > [2] NP-Match: When Neural Processes meet Semi-Supervised Learning. ICML 2022.

---

> ### Author Response · Authors · 2024-11-21
> **Responses to the Questions**
>
> ### **To Question 1**
>
> **1) learning dynamics  ... 2) convergence ... 3) is there a risk of instability ... during the optimization?**
>
> **1) Learning dynamics.** To better show the updating process, the example of image classification among the training process visualized by tSNE technique has been drawn in Figure 9 (Page 25), which implies the dynamics and correctness of learned masks.
>
> Following your suggestions, we’ve further added a section, "Appendix D.4.3 Impact of selected core size C" and plotted the curves on varying C in Figure 7, which presents the sensitivity of parameter C.
>
> **2) Convergence guarantee.** Theoretically, the upper problem for selecting informative features with proper masks enjoys the convergence guarantee (Theorem 1) under the constraint on empirically configured core size C. Notably, S2MAM could still identify the truly informative features with a slightly larger C than the ground truth.
>
> Empirically, the convergence curves in Figure 6 (Page 22) also imply the convergence of the proposal in the optimization process.
>
> **3) Instability risk.** As discussed in Remark 6 (Page 20, Line 1068), the instability results could be observed when setting extremely larger or smaller C than the ground truth, where some redundant features would be included or informative features are ignored.
>
> > [1] Accelerating flexible manifold embedding for scalable semi-supervised learning. TKDE 2018.
> >
> > [2] Semi-supervised learning with auto-weighting feature and
> > adaptive graph. TKDE 2019.
> >
> > [3] Adaptive local embedding learning for semi-supervised dimensionality reduction. IEEE TKDE 2021.
> >
> > [4] Robust embedding regression for semi-supervised learning. Pattern Recognition 2024.
>
> ### **To Question 2**
>
> **computational complexity compare to models like LapSVM or AWSSL? ... clarify its scalability for high-dimensional datasets and large sample sizes?**
>
> **1) Please refer to Common Concerns (4) for complexity analysis.**
>
> In Appendix E.5 (Page 45), the computational complexity is theoretically analyzed. Unsurprisingly, the bilevel scheme and additive modeling strategy bring a higher computation burden than single-level models like LapSVM and AWSSL.
>
> Besides, the empirical comparisons of training time cost and prediction on high-dimensional COIL images are summarized in Tables 11\&12 (in Page 26). The results show that S2MAM brings more training costs but enjoys competitive prediction accuracy than LapSVM and AWSSL, especially under noisy scenarios. Besides, S2MAM could provide feature-wise explanations for decisions (see Figure 9 in Page 25).
>
> **2) Please refer to Common Concerns (1) \& (2) for discussions and validations on scalability** to high-dimensional or large-size datasets, as well as some available preprocessing techniques for acceleration.
>
> ### **To Question 3**
>
> **provide additional experimental comparisons with state-of-the-art deep learning-based SSL methods ... ?**
>
> Thanks for pointing out the issues on more deep learning-based baselines.
> Following your suggestions, we’ve further conducted experiments with new competitors [1,2]. Please refer to the revised manuscript (see Tables 2,3,4,5 in Pages 9,10) and Common Concern (3).
>
> The empirical results show that, the proposed S2MAM could handle the noisy features and provide interpretable explanations for prediction. The competitive prediction performance and theoretical guarantees on convergence and generalization show the superiority of S2MAM over some state-of-the-art SSL networks.
>
> For completeness, the limitations and available solutions are further summarized in \emph{Appendix F: Limitations and Discussions}.
>
> > [1] Consistent feature selection for analytic deep neural networks. NeurIPS 2020.
> >
> > [2] NP-Match: When Neural Processes meet Semi-Supervised Learning. ICML 2022.
>
> ### **To Question 4**
>
> **The datasets used in the experiments appear relatively small. ...**
>
> Please refer to Common Concerns (1) for the summary of the existing high-dimensional and large-size datasets. Following your suggestions, we've highlighted the image experiments in Page 10 Line 531.
>
> There exist experiments on the high-dimensional COIL dataset in Appendix B.7. Some realistic UCI datasets (Buzz data) used in this work contain $38393$ samples.
>
> The existing results on large-size Buzz data and high-dimensional COIL data imply that
>
> 1) Under a clean scenario, the deep learning-based methods heavily relying on data quality could provide higher accuracy and variance than S2MAM while ignoring the interpretability.
>
> 2) Under noisy settings, the prediction performance of S2MAM is superior to most baselines.
>
> 3) S2MAM usually requires more training time, which could be controlled in this work by preprocessing like dimensional reduction [1,2].
>
> > [1] Adaptive local embedding learning for semi-supervised dimensionality reduction. IEEE TKDE 2021.
> >
> > [2] Robust embedding regression for semi-supervised learning. Pattern Recognition 2024.

---

### Author Response · Authors · 2024-11-21
**Common concerns from reviewers**

We thank all reviewers for their valuable and constructive comments. We address the common
concerns here and post a point-to-point response to each reviewer as well. Notice that some new results are reported in the resubmitted PDF, highlighted in red color.

### **Common Concerns (1): Experiments on Complex Data**

Indeed, this paper conducts experiments on 4 synthetic datasets, 8 UCI datasets and **one image-type COIL dataset**. And the following complex data scenarios have already been empirically considered:

1. **Large-size Data.** Both Buzz-Regression and Buzz-Classification datasets contain 38393 samples with 77 features. Please refer to Appendix B.1 for detailed descriptions of the selected datasets.
2. **High-dimensional Data.** The used images are selected from the COIL20 image library with 16384 original features (128x128). Corresponding results are shown in Tables 11\&12.
3. **More Labeled Training Samples.** In Table 10 (Appendix B.5, Page 25), the training set contains 5% or 50% labeled samples.
4. **Various Noisy Corruption.** The irrelevant and extremely noisy features are drawn from the distributions like Chi-square distribution and Gaussian distribution. The number of noisy dimensions varies in [0,10,100].



### **Common Concerns (2): Extensions to high-dimensional or non-numeric data**

1) **Existing high-dimensional applications**

As stated in Line 530 (Page 10), results of high-dimensional applications (e.g., images) are left in Appendix B.7 (Tables 11\&12).


2) **Methodology for processing high-dimensional or non-numeric data**

Inspired by some supervised [1] and semi-supervised works [2], an interesting approach for dealing with high-dimensional data like images and texts is to extract the feature vectors first. Following [2] and AWSSL, we first use a CNN to learn the vectors with a fixed number of features, which realizes rough dimensional reduction. However, this step may not remove those irrelevant or even noisy features.

Similar preprocessing methods for feature extraction and dimensional reduction also apply to larger or non-numeric datasets (image, text or biological data).

> [1] Stab-GKnock: Controlled variable selection for partially linear models using generalized knockoffs.
>
> [2] Robust embedding regression for semi-supervised learning.
>
> [3] Predicting effects of noncoding variants with deep learning–based sequence model.

### **Common Concerns (3): More deep learning-based competitors**

Following your constructive suggestions, we've added discussions and empirical comparisons on some recent deep learning baselines [1,2].

1) New results are reported in the revised pdf highlighted in red color.

Unsurprisingly, the supervised neural network [1] fails to select the informative features or predict with few labeled samples.
S2MAM still shows better robustness against noisy dimensions than those semi-supervised networks including [2], VAE and SSDKL, especially under noisy circumstances.

2) Model descriptions for [1,2] are added in Appendix B.2.

> [1] Consistent feature selection for analytic deep neural networks.
>
> [2] NP-Match: When Neural Processes meet Semi-Supervised Learning.



### **Common Concerns (4): Computation cost analysis**

**Indeed, the computational complexity has been theoretically analyzed in Appendix E.5.** The computation complexity of S2MAM reaches $\mathcal{O}\left(\frac{p(l+u)}{\epsilon_1^2 \epsilon_2} \right)$, where $\epsilon_1,\epsilon_2$ are the stationary parameters, $p,l,u$ stand for the sample dimensions, labeled and unlabeled sizes.

And the **empirical comparisons in training time cost on complex image data are left in Appendix B.7.**

In the following, we further restate the reasons at Appendix F for the higher computation costs as well as some techniques for acceleration.

**1) Why higher cost?**
As pointed out in Appendix B.7 and Appendix F, we admit the computation limitations of S2MAM.
This is mainly caused by:

1) The bilevel optimization requires more iterations to learn the additional masks.
2) The additive scheme expands the data dimensions to provide interpretable feature-wise contributions.

**2)  How to accelerate?**

Indeed, the probabilistic formulation and policy gradient estimation have been used in this work to reduce the computation burden of bilevel optimization.

To further accelerate the computation process, the random Fourier acceleration technique [1,2] and some dimensional reduction strategies [3] can be exploited to approximate the additive Gram matrix, which has been previously validated to be effective for additive models [4].

> [1] Random features for large-scale kernel machines.
>
> [2] Towards a unified analysis of random Fourier features.
>
> [3] Robust embedding regression for semi-supervised learning.
>
> [4] Tilted Sparse Additive Models.

---

### Author Response · Authors · 2024-12-01
**Dear Reviewers,**

Once again, we sincerely appreciate your comments, which has been extremely helpful for our work. We hope this message does not cause any inconvenience, but as we only have a few days left for discussions, we would like to ask whether our responses have sufficiently addressed your concerns.

---

### Meta-Review · Area_Chair_1E6f · 2024-12-19

**Metareview:**

The paper proposes a Semi-Supervised Meta Additive Model (S2MAM) that aims to improve the robustness and interpretability of semi-supervised learning through bilevel optimization. The proposed model combines manifold regularization, meta-learning, and sparse additive modeling to automatically identify informative features while simultaneously performing semi-supervised learning tasks. Theoretical guarantees are presented along with numerical experiments.

While reviewers appreciate the novelty of the method and the theoretical analysis, they also have major concerns about its high computational cost, especially for large data sets.

**Additional Comments On Reviewer Discussion:**

The concerns raised by reviewers include the following

- Computational complexity (Reviewers GvdS, iyKb): the proposed bilevel optimization scheme is computationally expensive and difficult to scale to large data sets. The authors acknowledged this limitation in their rebuttal. A thorough analysis of the computational costs in comparison with other competing methods (along the line in B.7) should be provided.

- Further real-world experiments, especially beyond the UCI datasets (Reviewer  GvdS, 5g1j): at the moment only COIL20, which is very small, is not part of UCI.

These points, among others, lead to the reject decision of the paper.

---

### Decision · Program_Chairs · 2025-01-22

Reject